# Exponentially Convergent Algorithms for Supervised Matrix Factorization

**Joowon Lee**
Department of Statistics
University of Wisconsin - Madison, WI, USA
jlee2256@wisc.edu

**Hanbaek Lyu**
Department of Mathematics
University of Wisconsin - Madison, WI, USA
hlyu@math.wisc.edu

**Weixin Yao**
Department of Statistics
University of California, Riverside, CA, USA
weixiny@ucr.edu

## Abstract

Supervised matrix factorization (SMF) is a classical machine learning method that simultaneously seeks feature extraction and classification tasks, which are not necessarily a priori aligned objectives. Our goal is to use SMF to learn low-rank latent factors that offer interpretable, data-reconstructive, and class-discriminative features, addressing challenges posed by high-dimensional data. Training SMF model involves solving a nonconvex and possibly constrained optimization with at least three blocks of parameters. Known algorithms are either heuristic or provide weak convergence guarantees for special cases. In this paper, we provide a novel framework that 'lifts' SMF as a low-rank matrix estimation problem in a combined factor space and propose an efficient algorithm that provably converges exponentially fast to a global minimizer of the objective with arbitrary initialization under mild assumptions. Our framework applies to a wide range of SMF-type problems for multi-class classification with auxiliary features. To showcase an application, we demonstrate that our algorithm successfully identified well-known cancer-associated gene groups for various cancers.

## 1   Introduction

In classical classification models, such as logistic regression, a conditional class-generating probability distribution is modeled as a simple function of the observed features with unknown parameters to be trained. However, the raw observed features may be high-dimensional, and most of them might be uninformative and hard to interpret (e.g., pixel values of an image). Therefore, it would be desirable to extract more informative and interpretable low-dimensional features prior to the classification task. For instance, the multi-layer perceptron or deep neural networks (DNN) in general [8, 9] use additional feature extraction layers prior to the logistic regression layer. This allows the model itself to learn the most effective (supervised) feature extraction mechanism and the association of the extracted features with class labels simultaneously.

Matrix factorization (MF) is a classical unsupervised feature extraction framework, which learns latent structures of complex datasets and is regularly applied in the analysis of text and images [13, 32, 44]. Various matrix factorization models such as singular value decomposition (SVD), principal component analysis (PCA), and nonnegative matrix factorization (NMF) provide fundamental tools for unsupervised feature extraction tasks [17, 55, 2, 25]. Extensive research has been conducted to adapt matrix factorization models to perform classification tasks by supervising the matrix factorization

37th Conference on Neural Information Processing Systems (NeurIPS 2023).

process using additional class labels. Note that matrix factorization and classification are not necessarily aligned objectives, so some degree of trade-off is necessary when seeking to achieve both goals simultaneously. *Supervised matrix factorization* (SMF) provides systematic approaches for such multi-objective tasks. Our goal is to use SMF to learn low-rank latent factors that offer interpretable, data-reconstructive, and class-discriminative features, addressing challenges posed by high-dimensional data. The general framework of SMF was introduced in [33]. A similar SMF-type framework of discriminative K-SVD was proposed for face recognition [61]. A stochastic formulation of SMF was proposed in [30]. SMF has also found numerous applications in various other problem domains, including speech and emotion recognition [16], music genre classification [62], concurrent brain network inference [62], structure-aware clustering [59], and object recognition [27]. Recently, supervised variants of NMF, as well as PCA, were proposed in [5, 26, 47]. See also the survey work of [15] on SMF.

Various SMF-type models have been proposed in the past two decades. We divide them into two categories depending on whether the extracted low-dimensional feature or the feature extraction mechanism itself is supervised. We refer to them as feature-based and filter-based SMF, respectively. Feature-based SMF models include the classical ones by Mairal et al. (see, e.g., [33, 30]) as well as the more recent model of convolutional matrix factorization by [22] for a contextual text recommendation system. Filter-based SMF models have been studied more recently in the supervised matrix factorization literature, most notably from supervised nonnegative matrix factorization [5, 26] and supervised PCA [47].

**Contributions**  In spite of vast literature on SMF, due to the high non-convexity of the associated optimization problem (see (4)), algorithms for SMF mostly lack rigorous convergence analysis and there has not been any algorithm that provably converges to a global minimizer of the objective at an exponential rate. We summarize our contributions below.

- We formulate a general class of SMF-type models (including both the feature- and the filter-based ones) with high-dimensional features as well as low-dimensional auxiliary features (see (4)).
- We provide a novel framework that 'lifts' SMF as a low-rank matrix estimation problem in a combined factor space and propose an efficient algorithm that converges exponentially fast to a global minimizer of the objective with an arbitrary initialization (Theorem 3.5) . We numerically validate our theoretical results (see Fig. 2).
- We theoretically compare the robustness of filter-based and feature-based SMF, establishing that the former is computationally more robust (see Theorem 3.5) while the latter is statistically more robust (see Theorem 4.1).
- Applying our method to microarray datasets for cancer classification, we show that not only it is competitive against benchmark methods, but it is able to identify groups of genes including well-known cancer-associated genes (see Fig. 3).

### 1.1 Notations

Throughout this paper, we denote by $\mathbb{R}^p$ the ambient space for data equipped with standard inner project $\langle \cdot, \cdot \rangle$ that induces the Euclidean norm $\|\cdot\|$. We denote by $\{0, 1, \ldots, \kappa\}$ the space of class labels with $\kappa + 1$ classes. For a convex subset $\Theta$ in an Euclidean space, we denote $\Pi_{\Theta}$ the projection operator onto $\Theta$. For an integer $r \geq 1$, we denote by $\Pi_r$ the rank-$r$ projection operator for matrices. For a matrix $\mathbf{A} = (a_{ij})_{ij} \in \mathbb{R}^{m \times n}$, we denote its Frobenius, operator (2-), and supremum norm by $\|\mathbf{A}\|_F^2 := \sum_{i,j} a_{ij}^2, \|\mathbf{A}\|_2 := \sup_{\mathbf{x} \in \mathbb{R}^n, \|\mathbf{x}\|=1} \|\mathbf{A}\mathbf{x}\|, \|\mathbf{A}\|_{\infty} := \max_{i,j} |a_{ij}|$, respectively. For each $1 \leq i \leq m$ and $1 \leq j \leq n$, we denote $\mathbf{A}[i, :]$ and $\mathbf{A}[:, j]$ for the $i$th row and the $j$th column of $\mathbf{A}$, respectively. For each integer $n \geq 1$, $\mathbf{I}_n$ denotes the $n \times n$ identity matrix. For square symmetric matrices $\mathbf{A}, \mathbf{B} \in \mathbb{R}^{n \times n}$, we denote $\mathbf{A} \preceq \mathbf{B}$ if $\mathbf{v}^T \mathbf{A} \mathbf{v} \leq \mathbf{v}^T \mathbf{B} \mathbf{v}$ for all unit vectors $\mathbf{v} \in \mathbb{R}^n$. For two matrices $\mathbf{A}$ and $\mathbf{B}$, we denote $[\mathbf{A}, \mathbf{B}]$ and $[\mathbf{A} \parallel \mathbf{B}]$ the matrices obtained by concatenating (stacking) them by horizontally and vertically, respectively, assuming matching dimensions.

### 1.2 Model setup

Suppose we are given with $n$ labeled signals $(y_i, \mathbf{x}_i, \mathbf{x}_i')$ for $i = 1, \ldots, n$, where $y_i \in \{0, 1, \ldots, \kappa\}$ is the label, $\mathbf{x}_i \in \mathbb{R}^p$ is a high-dimensional feature of $i$, and $\mathbf{x}_i' \in \mathbb{R}^q$ is a low-dimensional auxiliary feature of $i$ ($p \gg q$). For a vivid context, think of $\mathbf{x}_i$ as the X-ray image of a patient $i$ and $\mathbf{x}_i'$ denoting

some biological measurements, such as gender, smoking status, and body mass index. When making predictions of $y_i$, we use a suitable $r$ $(\ll p)$ dimensional compression of the high-dimensional feature $\mathbf{x}_i$ as well as the low-dimensional feature $\mathbf{x}_i'$ as-is. We assume such compression is done by some matrix of *(latent) factors* $\mathbf{W} = [\mathbf{w}_1, \ldots, \mathbf{w}_r] \in \mathbb{R}^{p \times r}$ that is *reconstructive* in the sense that the observed signals $\mathbf{x}_i$ can be reconstructed as (or approximated by) the linear transform of the 'atoms' $\mathbf{w}_1, \ldots, \mathbf{w}_r \in \mathbb{R}^p$ for some suitable 'code' $\mathbf{h}_i \in \mathbb{R}^r$. More concisely, $\mathbf{X}_{\text{data}} = [\mathbf{x}_1, \ldots, \mathbf{x}_n] \approx \mathbf{WH}$, where $\mathbf{H} = [\mathbf{h}_1, \ldots, \mathbf{h}_n] \in \mathbb{R}^{r \times n}$. In practice, we can choose $r$ to be the approximate rank of data matrix $\mathbf{X}_{\text{data}}$ (e.g., by finding the elbow of the scree plot).

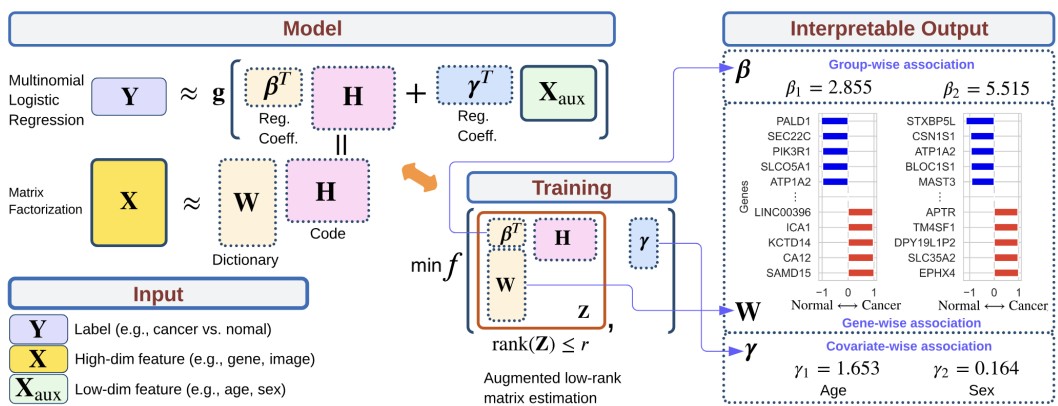

Figure 1: Overall scheme of the proposed method for SMF-$\mathbf{H}$.

Now, we state our probabilistic modeling assumption. Fix parameters $\mathbf{W} \in \mathbb{R}^{p \times r}$, $\mathbf{h}_i \in \mathbb{R}^r$, $\boldsymbol{\beta} \in \mathbb{R}^{r \times \kappa}$, and $\boldsymbol{\gamma} \in \mathbb{R}^{q \times \kappa}$. Let $h : \mathbb{R} \to [0, \infty)$ be a *score function* (e.g., $h(\cdot) = \exp(\cdot)$ for multinomial logistic regression). We assume $y_i$ is a realization of a random variable whose conditional distribution is specified as

$$[\mathbb{P}(y_i = 0 \mid \mathbf{x}_i, \mathbf{x}_i'), \ldots, \mathbb{P}(y_i = \kappa \mid \mathbf{x}_i, \mathbf{x}_i')] = \mathbf{g}(\mathbf{a}_i) := C[1, h(\mathbf{a}_{i,1}), \ldots, h(\mathbf{a}_{i,\kappa})], \quad (1)$$

where $C$ is the normalization constant and $\mathbf{a}_i = (\mathbf{a}_{i,1}, \ldots, \mathbf{a}_{i,\kappa}) \in \mathbb{R}^{\kappa}$ is the *activation* for $y_i$ defined in two ways, depending on whether we use a 'feature-based' or 'filter-based' SMF model:

$$\mathbf{a}_i = \begin{cases} \boldsymbol{\beta}^T \mathbf{h}_i + \boldsymbol{\gamma}^T \mathbf{x}_i' & \text{for feature-based (SMF-}\mathbf{H}\text{)}, \\ \boldsymbol{\beta}^T \mathbf{W}^T \mathbf{x}_i + \boldsymbol{\gamma}^T \mathbf{x}_i' & \text{for filter-based (SMF-}\mathbf{W}\text{)}. \end{cases} \quad (2)$$

One may regard $(\boldsymbol{\beta}, \boldsymbol{\gamma})$ as the 'multinomial regression coefficients' with input feature $(\mathbf{h}_i, \mathbf{x}_i')$ or $(\mathbf{W}^T \mathbf{x}_i, \mathbf{x}_i')$. In (2), we may regard the code $\mathbf{h}_i$ (coming from $\mathbf{x}_i \approx \mathbf{W}\mathbf{h}_i$) or the 'filtered signal' $\mathbf{W}^T \mathbf{x}_i$ as the $r$-dimensional compression of $\mathbf{x}_i$. Note that these two coincide if we have perfect factorization $\mathbf{x}_i = \mathbf{W}\mathbf{h}_i$ and the factor matrix $\mathbf{W}$ are orthonormal, i.e., $\mathbf{W}^T \mathbf{W} = \mathbf{I}_r$, but we do not necessarily make such an assumption.

There are some notable differences between SMF-$\mathbf{H}$ and SMF-$\mathbf{W}$ when predicting the unknown label of a test point. If we are given a test point $(\mathbf{x}_{\text{test}}, \mathbf{x}_{\text{test}}')$, the predictive probabilities for its unknown label $y_{\text{test}}$ is given by (1) with activation $\mathbf{a}$ computed as in (2). This only involves straightforward matrix multiplications for SMF-$\mathbf{W}$, which can also be viewed as a forward propagation in a multilayer perceptron [35] with $\mathbf{W}$ acting as the first layer weight matrix (hence named 'filter'). However, for SMF-$\mathbf{H}$, one needs to solve additional optimization problems for testing. Namely, for every single test signal $(\mathbf{x}_{\text{test}}, \mathbf{x}_{\text{test}}')$, its correct code representation $\mathbf{h}_{\text{test}}$ needs to be learned by solving the following 'supervised sparse coding' problem (see [33]):

$$\min_{y \in \{0,1,\ldots,\kappa\}} \min_{\mathbf{h}} \ell(y, \boldsymbol{\beta}^T \mathbf{h} + \boldsymbol{\gamma}^T \mathbf{x}_{\text{test}}') + \xi \|\mathbf{x}_{\text{test}} - \mathbf{W}\mathbf{h}\|_F^2. \quad (3)$$

A more efficient heuristic testing method for SMF-$\mathbf{H}$ is by approximately computing $\mathbf{h}_{\text{test}}$ by only minimizing the second term in (3).

In order to estimate the model parameters $(\mathbf{W}, \mathbf{H}, \boldsymbol{\beta}, \boldsymbol{\gamma})$ from observed training data $(\mathbf{x}_i, y_i)$ for $i = 1, \ldots, n$, we consider the following multi-objective optimization problem:

$$\min_{\mathbf{W}, \mathbf{H}, \boldsymbol{\beta}, \boldsymbol{\gamma}} \sum_{i=1}^{n} \ell(y_i, \mathbf{a}_i) + \xi \|\mathbf{X}_{\text{data}} - \mathbf{WH}\|_F^2, \quad (4)$$

where $\mathbf{X}_{\mathrm{data}} = [\mathbf{x}_1, \ldots, \mathbf{x}_n] \in \mathbb{R}^{p \times n}$, $\mathbf{a}_i$ is as in (2), and $\ell(\cdot)$ is the classification loss measured by the negative log-likelihood:

$$\ell(y, \mathbf{a}) := \log \left( 1 + \sum_{c=1}^{\kappa} h(a_c) \right) - \sum_{c=1}^{\kappa} \mathbf{1}_{\{y=c\}} \log h(a_c). \tag{5}$$

In (4), the *tuning parameter* $\xi$ controls the trade-off between the two objectives of classification and matrix factorization. The above is a nonconvex problem involving four blocks of parameters that could have additional constraints (e.g., bounded norm). This problem entails many classical models as special cases. When $\xi \gg 1$ so that effectively only the second term in (4) is being minimized with respect to $\mathbf{W}$ and $\mathbf{H}$, it becomes the classical matrix factorization problem [31, 28, 29], where one seeks to find factor matrix $\mathbf{W}$ that can best reconstruct the feature vectors $\mathbf{X}_{\mathrm{data}}$ via the factorization $\mathbf{X}_{\mathrm{data}} \approx \mathbf{WH}$. In Figure 3, we will demonstrate that the best reconstructive factor matrix could be significantly different from the supervised factor matrix learned by SMF and may not be very effective for classification tasks.

## 1.3 Related works

The SMF training problem (4) is a nonconvex and possibly constrained optimization problem, generally with non-unique minimizers. Since it is difficult to solve exactly, approximate procedures such as block coordinate descent (BCD) (see, e.g., [57]) are often used. Such methods utilize the fact that the objective function in (4) is convex in each of the four (matrix) variables. Such an algorithm proceeds by iteratively optimizing for only one block while fixing the others (see [33, 5, 26, 47]). However, convergence analysis or statistical estimation bounds of such algorithms are quite limited. Appealing to general convergence results for BCD methods (e.g., [18, 58]), one can at most guarantee asymptotic convergence to the stationary points or polynomial convergence to Nash equilibria or of the objective (4), modulo carefully verifying the assumptions of these general results. We also remark that [30] provided a rigorous justification of the differentiability of a feature-based SMF model.

The main finding of our work is that the non-convexity of the SMF problem (4) is 'benign', in the sense that *there exists an algorithm globally convergent to a global optimum at an exponential rate.* We use a 'double-lifting' technique that converts the nonconvex SMF problem (4) into a low-rank factored estimation with a convex objective. This is reminiscent of the tight relation between a low-rank matrix estimation and a nonconvex factored estimation problem, which has been actively employed in a body of works in statistics and optimization [3, 45, 36, 64, 53, 56, 42, 41, 51]. Our exponentially convergent SMF algorithms are versions of low-rank projected gradient descent in the algorithm (44) that operate in the double-lifted space.

## 2 Methods

### 2.1 Sketch of key idea

Our key idea to solve (4) is to transform it into a variant of the low-rank matrix estimation problem (6) and then use a *Low-rank Projected Gradient Descent* (LPGD) algorithm (7):

$$\min_{\mathbf{Z} = [\boldsymbol{\theta}, \boldsymbol{\gamma}] \in \boldsymbol{\Theta}, \, \mathrm{rank}(\boldsymbol{\theta}) \leq r} f(\mathbf{Z}) \quad (6) \qquad \mathbf{Z}_t \leftarrow \Pi_r \left( \Pi_{\boldsymbol{\Theta}} \left( \mathbf{Z}_{t-1} - \tau \nabla f(\mathbf{Z}_{t-1}) \right) \right), \; \tau > 0 \text{ fixed.} \quad (7)$$

In (6), one seeks to minimize an objective $f$ w.r.t. a paired matrix parameter $\mathbf{Z} = [\boldsymbol{\theta}, \boldsymbol{\gamma}]$ within a convex constraint set $\boldsymbol{\Theta}$ and an additional rank constraint $\mathrm{rank}(\boldsymbol{\theta}) \leq r$. In (7), $\Pi_r$ denotes applying rank-$r$ projection on the first factor $\boldsymbol{\theta}$ while keeping $\boldsymbol{\gamma}$ the same. Problem (6) encompasses a variety of significant problems, such as matrix regression [11, 38] and matrix completion [48, 37]. For these problems, algorithms of type (7) have been studied in [41, 56].

To illustrate how the SMF problem (4) transforms into a low-rank matrix estimation (6), we consider a much simpler version of SMF-$\mathbf{H}$. That is, instead of combining matrix factorization with multinomial logistic regression for multi-class classification problems, we combine it with **linear regression**. Thus, the response variable $y$ in this discussion assumes all values in the real line. For additional simplicity, we assume there are no auxiliary features $\mathbf{X}_{\mathrm{aux}}$. We seek to solve matrix factorization and linear regression problems simultaneously for data matrix $\mathbf{X}_{\mathrm{data}} \in \mathbb{R}^{p \times n}$ and response variable

$\mathbf{Y} \in \mathbb{R}^{1 \times n}$:

$$\min_{\mathbf{W}, \mathbf{H}, \boldsymbol{\beta}} \|\mathbf{Y} - \boldsymbol{\beta}^T \mathbf{H}\|_F^2 + \xi \|\mathbf{X}_{\text{data}} - \mathbf{W}\mathbf{H}\|_F^2. \tag{8}$$

This is a three-block optimization problem involving three factors $\mathbf{W} \in \mathbb{R}^{p \times r}, \mathbf{H} \in \mathbb{R}^{r \times n}$ and $\boldsymbol{\beta} \in \mathbb{R}^{r \times 1}$, which is nonconvex and computationally challenging to solve exactly. Instead, consider reformulating this nonconvex problem as the following matrix factorization problem:

$$\min_{\mathbf{W}, \mathbf{H}, \boldsymbol{\beta}} f \left( \begin{bmatrix} \boldsymbol{\beta}^T \\ \mathbf{W} \end{bmatrix} \mathbf{H} \right) := \left\| \begin{bmatrix} \mathbf{Y} \\ \sqrt{\xi} \mathbf{X}_{\text{data}} \end{bmatrix} - \begin{bmatrix} \boldsymbol{\beta}^T \\ \sqrt{\xi} \mathbf{W} \end{bmatrix} \mathbf{H} \right\|_F^2. \tag{9}$$

Indeed, we now seek to find *two* decoupled matrices (instead of three), one for $\boldsymbol{\beta}^T$ and $\mathbf{W}$ stacked vertically, and the other for $\mathbf{H}$. A similar idea of matrix stacking was used in [61] for discriminative K-SVD. Proceeding one step further, another important observation we make is that it is also equivalent to finding a *single* matrix $\boldsymbol{\theta} := \begin{bmatrix} \boldsymbol{\beta}^T \mathbf{H} \, \| \, \mathbf{W}\mathbf{H} \end{bmatrix} \in \mathbb{R}^{(1+p) \times n}$ of rank at most $r$ that minimizes the function $f$ in (9), which is convex (specifically, quadratic) in $\boldsymbol{\theta}$: (See Fig. 1 Training).

For SMF-$\mathbf{W}$, consider the following analogous linear regression model:

$$\min_{\mathbf{W}, \mathbf{H}, \boldsymbol{\beta}} f \left( \mathbf{W}[\boldsymbol{\beta}, \mathbf{H}] \right) := \|\mathbf{Y} - \boldsymbol{\beta}^T \mathbf{W}^T \mathbf{X}_{\text{data}}\|_F^2 + \xi \|\mathbf{X}_{\text{data}} - \mathbf{W}\mathbf{H}\|_F^2, \tag{10}$$

where the right-hand side above is obtained by replacing $\mathbf{H}$ with $\mathbf{W}^T \mathbf{X}_{\text{data}}$ in (9). Note that the objective function depends only on the product of the two matrices $\mathbf{W}$ and $[\boldsymbol{\beta}, \mathbf{H}]$. Then, we may further lift it as the low-rank matrix estimation problem by seeking a single matrix $\boldsymbol{\theta} := [\mathbf{W}\boldsymbol{\beta}, \, \mathbf{W}\mathbf{H}] \in \mathbb{R}^{p \times (1+n)}$ of rank at most $r$ that solves (6) with $f$ being the function in (10).

## 2.2 Algorithm

Motivated by the observation we made before, we rewrite SMF-$\mathbf{H}$ in (4) as

$$\min_{\substack{[\boldsymbol{\theta}, \boldsymbol{\gamma}] \in \boldsymbol{\Theta} \\ \text{rank}(\boldsymbol{\theta}) \leq r}} F(\boldsymbol{\theta}, \boldsymbol{\gamma}) := \sum_{i=1}^n \ell(y_i, \mathbf{A}[:, i] + \boldsymbol{\gamma}^T \mathbf{x}_i') + \xi \|\mathbf{X}_{\text{data}} - \mathbf{B}\|_F^2 + \lambda \left( \|\mathbf{A}\|_F^2 + \|\boldsymbol{\gamma}\|_F^2 \right), \tag{11}$$

where $\mathbf{A} = \boldsymbol{\beta}^T \mathbf{H}, \mathbf{B} = \mathbf{W}\mathbf{H}, \boldsymbol{\theta} = [\mathbf{A} \, \| \, \mathbf{B}] \in \mathbb{R}^{(\kappa+p) \times n}$, and $\boldsymbol{\Theta}$ is a convex subset of $\mathbb{R}^{(\kappa+p) \times n} \times \mathbb{R}^{q \times \kappa}$. We have added a $L_2$-regularization term for $\mathbf{A}$ and $\boldsymbol{\gamma}$ with coefficient $\lambda \geq 0$, which will play a crucial role in well-conditioning (11).

For solving (11), we propose to use the LGPD algorithm (7): *We iterate gradient descent followed by projecting onto the convex constraint set $\boldsymbol{\Theta}$ of the combined factor $[\boldsymbol{\theta}, \boldsymbol{\gamma}]$ and then perform rank-$r$ projection of the first factor $\boldsymbol{\theta} = [\mathbf{A} \, \| \, \mathbf{B}]$ via truncated SVD until convergence.* Once we have a solution $[\boldsymbol{\theta}^\star, \boldsymbol{\gamma}^\star]$ to (11), we can use SVD of $\boldsymbol{\theta}^\star$ to obtain a solution to (4). Let $\boldsymbol{\theta}^\star = \mathbf{U}\boldsymbol{\Sigma}\mathbf{V}^T$ denote the SVD of $\boldsymbol{\theta}$. Since $\text{rank}(\boldsymbol{\theta}^\star) \leq r$, $\boldsymbol{\Sigma}$ is an $r \times r$ diagonal matrix of singular values of $\boldsymbol{\theta}$. Then $\mathbf{U} \in \mathbb{R}^{(\kappa+p) \times r}$ and $\mathbf{V} \in \mathbb{R}^{n \times r}$ are semi-orthonormal matrices, that is, $\mathbf{U}^T \mathbf{U} = \mathbf{V}^T \mathbf{V} = \mathbf{I}_r$. Then since $\boldsymbol{\theta}^\star = [(\boldsymbol{\beta}^\star)^T \, \| \, \mathbf{W}^\star]\mathbf{H}^\star$, we can take $\mathbf{H}^\star = \boldsymbol{\Sigma}^{1/2}\mathbf{V}^T$ and $[(\boldsymbol{\beta}^\star)^T \, \| \, \mathbf{W}^\star] = \mathbf{U}\boldsymbol{\Sigma}^{1/2}$.

We summarize this approach of solving (4) for SMF-$\mathbf{H}$ in Algorithm 1. Here, $\text{SVD}_r$ denotes rank-$r$ truncated SVD and the projection operators $\Pi_{\boldsymbol{\Theta}}$ and $\Pi_r$ are defined in Subsection 1.1.

As for SMF-$\mathbf{W}$, we can rewrite (4) with additional $L_2$-regularizer for $\mathbf{A} = \mathbf{W}\boldsymbol{\beta}$ and $\boldsymbol{\gamma}$ as

$$\min_{\substack{[\boldsymbol{\theta}, \boldsymbol{\gamma}] \in \boldsymbol{\Theta} \\ \text{rank}(\boldsymbol{\theta}) \leq r}} F(\boldsymbol{\theta}, \boldsymbol{\gamma}) = \sum_{i=1}^n \ell(y_i, \mathbf{A}^T \mathbf{x}_i + \boldsymbol{\gamma}^T \mathbf{x}_i') + \xi \|\mathbf{X}_{\text{data}} - \mathbf{B}\|_F^2 + \lambda \left( \|\mathbf{A}\|_F^2 + \|\boldsymbol{\gamma}\|_F^2 \right), \tag{12}$$

where $\boldsymbol{\theta} = [\mathbf{A}, \mathbf{B}] = \mathbf{W}[\boldsymbol{\beta}, \mathbf{H}] \in \mathbb{R}^{p \times (\kappa+n)}$ and $\boldsymbol{\Theta} \in \mathbb{R}^{p \times (\kappa+n)} \times \mathbb{R}^{q \times \kappa}$ is a convex set. Algorithm 1 for SMF-$\mathbf{W}$ follows similar reasoning as before with the reformulation above.

By using randomized truncated SVD for the efficient low-rank projection in Algorithm 1, the per-iteration complexity is $O(pn \min(n, p))$, while that for the nonconvex algorithm is $O((pr + q)n)$. While the LPGD algorithm is in general more expensive per iteration than the nonconvex method, the iteration complexity is only $O(\log \epsilon^{-1})$ thanks to the exponential convergence to the global optimum (will be discussed in Theorem 3.5). To our best knowledge, the nonconvex algorithm for SMF does not have any guarantee to converge to a global optimum, and the iteration complexity of the nonconvex SMF method to reach an $\epsilon$-stationary point is at best $O(\epsilon^{-1})$ using standard analysis. Hence for $\epsilon$ small enough, Algorithm 1 achieves an $\epsilon$-accurate global optimum for SMF with a total computational cost comparable to a nonconvex SMF algorithm to achieve an $\epsilon$-stationary point.

**Algorithm 1** Lifted PGD for SMF
***

**Input:** $\mathbf{X}_{\text{data}} \in \mathbb{R}^{p \times n}$; $\mathbf{X}'_{\text{aux}} \in \mathbb{R}^{q \times n}$ (auxiliary features); $\mathbf{Y}_{\text{label}} \in \{0, 1, \ldots, \kappa\}^n$
**Parameters:** $\tau > 0$ (stepsize); $N \in \mathbb{N}$ (iterations); $r \geq 1$ (rank); $\lambda \geq 0$ ($L_2$-reg. param.)
**Constraints:** Convex $\boldsymbol{\Theta} \subseteq \mathbb{R}^{(\kappa+p) \times n} \times \mathbb{R}^{q \times \kappa}$ for SMF-**H**, $\boldsymbol{\Theta} \subseteq \mathbb{R}^{p \times (\kappa+n)} \times \mathbb{R}^{q \times \kappa}$ for SMF-**W**;
Initialize $\mathbf{W}_0 \in \mathbb{R}^{p \times r}, \mathbf{H}_0 \in \mathbb{R}^{r \times n}, \boldsymbol{\beta}_0 \in \mathbb{R}^{r \times \kappa}, \boldsymbol{\gamma}_0 \in \mathbb{R}^{q \times \kappa}$

$\begin{cases} \boldsymbol{\theta}_0 \leftarrow [\boldsymbol{\beta}_0^T \mathbf{H}_0 \;\|\; \mathbf{W}_0 \mathbf{H}_0] \in \mathbb{R}^{(\kappa+p) \times n} & (\triangleright \text{ for SMF-}\mathbf{H}) \\ \boldsymbol{\theta}_0 \leftarrow [\mathbf{W}_0 \boldsymbol{\beta}_0, \mathbf{W}_0 \mathbf{H}_0] \in \mathbb{R}^{p \times (\kappa+n)} & (\triangleright \text{ for SMF-}\mathbf{W}) \end{cases}$

**for** $k = 1$ **to** $N$ **do**
    $\boldsymbol{\theta}_k \leftarrow \Pi_r \left( \Pi_{\boldsymbol{\Theta}} \left( \boldsymbol{\theta}_{k-1} - \tau \nabla_{\boldsymbol{\theta}} F(\boldsymbol{\theta}_{k-1}, \boldsymbol{\gamma}_{k-1}) \right) \right)$         $(\triangleright \text{ See Appendix A for computation})$
    $\boldsymbol{\gamma}_k \leftarrow \boldsymbol{\gamma}_{k-1} - \tau \nabla_{\boldsymbol{\gamma}} F(\boldsymbol{\theta}_{k-1}, \boldsymbol{\gamma}_{k-1})$
$\boldsymbol{\theta}_N = \mathbf{U} \boldsymbol{\Sigma} \mathbf{V}^T$       $(\triangleright \text{ rank-}r \text{ SVD})$
$\begin{cases} [\boldsymbol{\beta}_N^T \;\|\; \mathbf{W}_N] \leftarrow \mathbf{U} \boldsymbol{\Sigma}^{1/2}, \mathbf{H}_N \leftarrow (\boldsymbol{\Sigma})^{1/2} \mathbf{V}^T & (\triangleright \text{SMF-}\mathbf{H}) \\ \mathbf{W}_N \leftarrow \mathbf{U}, [\boldsymbol{\beta}_N, \; \mathbf{H}_N] \leftarrow \boldsymbol{\Sigma} \mathbf{V}^T & (\triangleright \text{SMF-}\mathbf{W}) \end{cases}$
**Output:** $(\mathbf{W}_N, \mathbf{H}_N, \boldsymbol{\beta}_N, \boldsymbol{\gamma}_N)$

***

# 3 Global convergence guarantee

We have discussed that one can cast the SMF problem (4) as the following 'factored estimation problem' $\min_{\mathbf{T}, \mathbf{S}, \boldsymbol{\gamma}} f(\mathbf{T} \mathbf{S}^T, \boldsymbol{\gamma})$. Note that such problems generally do not have a unique minimizer due to the 'rotation invariance'. Namely, let $\mathbf{R}$ be any $r \times r$ orthonormal (rotation) matrix (i.e., $\mathbf{R}^T \mathbf{R} = \mathbf{R} \mathbf{R}^T = \mathbf{I}_r$). Then $f((\mathbf{T}\mathbf{R})(\mathbf{S}\mathbf{R})^T, \boldsymbol{\gamma}) = f(\mathbf{T} \mathbf{R} \mathbf{R}^T \mathbf{S}^T, \boldsymbol{\gamma}) = f(\mathbf{T} \mathbf{S}^T, \boldsymbol{\gamma})$. Hence the best one is to obtain parameters up to rotation that globally minimize the objective value. Our main result, Theorem 3.5, establishes that this can be achieved by Algorithm 1 at an exponential rate. First, we introduce the following technical assumptions (3.1-3.3).

**Assumption 3.1.** (Bounded activation) The activation $\mathbf{a} \in \mathbb{R}^\kappa$ defined in (2) assumes bounded norm, i.e., $\|\mathbf{a}\| \leq M$ for some constant $M \in (0, \infty)$.

**Assumption 3.2.** (Bounded eigenvalues of covariance matrix) Denote $\boldsymbol{\Phi} = [\boldsymbol{\phi}_1, \ldots, \boldsymbol{\phi}_n] \in \mathbb{R}^{(p+q) \times n}$, where $\boldsymbol{\phi}_i = [\mathbf{x}_i \;\|\; \mathbf{x}'_i] \in \mathbb{R}^{p+q}$ (so $\boldsymbol{\Phi} = [\mathbf{X}_{\text{data}} \;\|\; \mathbf{X}_{\text{aux}}]$), where $\mathbf{X}_{\text{aux}} = [\mathbf{x}'_1, \ldots, \mathbf{x}'_n]$. Then, there exist constants $\delta^-, \delta^+ > 0$ such that for all $n \geq 1$,

$$\delta^- \leq \lambda_{\min}(n^{-1} \boldsymbol{\Phi} \boldsymbol{\Phi}^T) \leq \lambda_{\max}(n^{-1} \boldsymbol{\Phi} \boldsymbol{\Phi}^T) \leq \delta^+. \tag{13}$$

**Assumption 3.3.** (Bounded stiffness and eigenvalues of observed information) The score function $h : \mathbb{R} \to [0, \infty)$ is twice continuously differentiable. Further, let observed information $\ddot{\mathbf{H}}(y, \mathbf{a}) := \nabla_{\mathbf{a}} \nabla_{\mathbf{a}^T} \ell(y, \mathbf{a})$ for $y$ and $\mathbf{a}$. Then, for the constant $M > 0$ in Assumption 3.1, there are constants $\gamma_{\max}, \alpha^-, \alpha^+ > 0$ s.t. $\gamma_{\max} := \sup_{\|\mathbf{a}\| \leq M} \max_{1 \leq s \leq n} \|\nabla_{\mathbf{a}} \ell(y_s, \mathbf{a})\|_\infty$ and

$$\alpha^- := \inf_{\|\mathbf{a}\| \leq M} \min_{1 \leq s \leq n} \lambda_{\min}(\ddot{\mathbf{H}}(y_s, \mathbf{a})), \quad \alpha^+ := \sup_{\|\mathbf{a}\| \leq M} \max_{1 \leq s \leq n} \lambda_{\max}(\ddot{\mathbf{H}}(y_s, \mathbf{a})). \tag{14}$$

Assumption 3.1 limits the norm of the activation $\mathbf{a}$ as an input for the classification model in (4) is bounded. This is standard in the literature (see, e.g., [36, 60, 23]) in order to uniformly bound the eigenvalues of the Hessian of the (multinomial) logistic regression model. Assumption 3.2 introduces uniform bounds on the eigenvalues of the covariance matrix. Assumption 3.3 introduces uniform bounds on the eigenvalues of the $\kappa \times \kappa$ observed information as well as the first derivative of the predictive probability distribution (see [10] and Sec. D in Appendix for more details). Under Assumption 3.1 and the multinomial logistic regression model $h(\cdot) = \exp(\cdot)$, one can derive Assumption 3.3 with a simple expression for the bounds $\alpha^\pm$, as discussed in the following remark.

**Remark 3.4** (Multinomial Logistic Classifier). Let $\ell$ denote the negative log-likelihood function in (5), where we take the multinomial logistic model with the score function $h(\cdot) = \exp(\cdot)$. Denote $(\dot{h}_1, \ldots, \dot{h}_\kappa) := \nabla_{\mathbf{a}} \ell(y, \mathbf{a})$ and $\ddot{\mathbf{H}}(y, \mathbf{a}) := \nabla_{\mathbf{a}} \nabla_{\mathbf{a}^T} \ell(y, \mathbf{a})$. Then in this special case, we have $\dot{h}_j(y, \mathbf{a}) = g_j(\mathbf{a}) - \mathbf{1}(y = j)$ and $\ddot{H}(y, \mathbf{a})_{i,j} = g_i(\mathbf{a}) (\mathbf{1}(i = j) - g_j(\mathbf{a}))$ (See (28) and (30) in Appendix). Under Assumption 3.1, according to Lemma B.1, we can take $\gamma_{\max} = 1 + \frac{e^M}{1+e^M+(\kappa-1)e^{-M}} \leq 2$, $\alpha^- = \frac{e^{-M}}{1+e^{-M}+(\kappa-1)e^M}$, and $\alpha^+ = \frac{e^M(1+2(\kappa-1)e^M)}{(1+e^M+(\kappa-1)e^{-M})^2}$. For binary classification, $\alpha^+ \leq 1/4$.

Now define the following quantities:

$$\mu := \begin{cases} \min(2\xi, \, 2\lambda + n\delta^{-}\alpha^{-}) \\ \min(2\xi, \, 2\lambda) \end{cases}, \quad L := \begin{cases} \max(2\xi, \, 2\lambda + n\delta^{+}\alpha^{+}) & \text{for SMF-}\mathbf{W} \\ \max(2\xi, \, 2\lambda + \alpha^{+}) & \text{for SMF-}\mathbf{H} \end{cases}. \tag{15}$$

Now, we state a special case of our first main result, specifically when the model is 'correctly specified', allowing the rank-$r$ SMF model to effectively account for the observed data. This implies the existence of a 'low-rank stationary point' of $F$, as also demonstrated in [56]. However, we also handle the general case in Appendix (see Theorem D.1).

**Theorem 3.5.** *(Exponential convergence) Let $\mathbf{Z}_t := [\boldsymbol{\theta}_t, \boldsymbol{\gamma}_t]$ denote the iterates of Algorithm 1. Assume 3.1-3.3 hold. Let $\mu$ and $L$ be as in (15), fix $\tau \in (\frac{1}{2\mu}, \frac{3}{2L})$, and let $\rho := 2(1 - \tau\mu) \in (0, 1)$. Suppose $L/\mu < 3$ and let $\mathbf{Z}^* = [\boldsymbol{\theta}^*, \boldsymbol{\gamma}^*]$ be any stationary point of $F$ over $\Theta$ s.t. $\mathrm{rank}(\boldsymbol{\theta}^*) \leq r$. Then $\mathbf{Z}^*$ is the unique global minimizer of $F$ among all $\mathbf{Z} = [\boldsymbol{\theta}, \boldsymbol{\gamma}]$ with $\mathrm{rank}(\boldsymbol{\theta}) \leq r$. Moreover, $\|\mathbf{Z}_t - \mathbf{Z}^*\|_F \leq \rho^t \|\mathbf{Z}_0 - \mathbf{Z}^*\|_F$ for $t \geq 1$.*

In the statement above, we write $\|\mathbf{Z}\|_F^2 = \|[\boldsymbol{\theta}, \boldsymbol{\gamma}]\|_F^2 := \|\boldsymbol{\theta}\|_F^2 + \|\boldsymbol{\gamma}\|_F^2$. Note that we may view the ratio $L/\mu$ that appears in Theorem 3.5 as the condition number of the SMF problem in (4), whereas the ratio $L^*/\mu^*$ for $\mu^* := \delta^-\alpha^-$ and $L^* := \delta^+\alpha^+$ as the condition number for the multinomial classification problem. These two condition numbers are closely related. First, note that for any given $\mu^*, L^*$ and sample size $n$, we can always make $L/\mu < 3$ by choosing sufficiently large $\xi$ and $\lambda$ so that Theorem 3.5 holds. However, using large $L_2$-regularization parameter $\lambda$ may perturb the original objective in (4) too much that the converged solution may not be close to the optimal solution. Hence, we may want to take $\lambda$ as small as possible. Setting $\lambda = 0$ leads to

$$\frac{L}{\mu} < 3, \, \lambda = 0 \iff \begin{cases} 0 < \frac{L^*}{\mu^*} < 3, \, \frac{L^*}{6} < \frac{\xi}{n} < \frac{3\mu^*}{2} & \text{for SMF-}\mathbf{W} \\ \frac{\max(2\xi, \alpha^+)}{\min(2\xi, 0)} < 3 & \text{for SMF-}\mathbf{H}. \end{cases} \tag{16}$$

For SMF-$\mathbf{W}$, if the multinomial classification problem is well-conditioned ($L^*/\mu^* < 3$) and the ratio $\xi/n$ is in the above interval, then SMF-$\mathbf{W}$ enjoys exponential convergence in Theorem 3.5. However, the condition for SMF-$\mathbf{H}$ in (16) is violated, so $L_2$-regularization is necessary for guaranteeing exponential convergence of SMF-$\mathbf{H}$.

The proof of Theorem 3.5 involves two steps: (1) We establish a general exponential convergence result for the general LPGD algorithm (7) in Theorem C.2 in Appendix. (2) We compute the Hessian eigenvalues of the SMF objectives (11)-(12) and apply the result to obtain Theorem 3.5. The proof contains two challenges: first, the low-rank projection in (7) is not non-expansive in general. To overcome this, we show that the iterates closely approximate certain 'auxiliary iterates' which exhibit exponential convergence towards the global optimum. Secondly, the second-order analysis is highly non-trivial since the SMF problem (4) has a total of four unknown matrix factors that are intertwined through the joint multi-class classification and matrix factorization tasks. See Appendix D for the details.

## 4 Statistical estimation guarantee

In this section, we formulate a generative model for SMF (4) and state statistical parameter estimation guarantee. Fix dimensions $p \gg q$, and let $n \geq 1$ be possibly growing sample size, and fix unknown true parameters $\mathbf{B}^\star \in \mathbb{R}^{p \times n}$, $\mathbf{C}^\star \in \mathbb{R}^{q \times n}$, $\boldsymbol{\gamma}^\star \in \mathbb{R}^{q \times \kappa}$. In addition, fix $\mathbf{A}^\star \in \mathbb{R}^{\kappa \times n}$ for SMF-$\mathbf{H}$ and $\mathbf{A}^\star \in \mathbb{R}^{p \times \kappa}$ for SMF-$\mathbf{W}$. Now suppose that class label, data, and auxiliary features are drawn i.i.d. according to the following joint distribution:

$$\begin{cases} \mathbf{x}_i = \mathbf{B}^\star[:, i] + \boldsymbol{\varepsilon}_i, & \mathbf{x}_i' = \mathbf{C}^\star[:, i] + \boldsymbol{\varepsilon}_i', \\ y_i \mid \mathbf{x}_i, \mathbf{x}_i' \sim \text{Multinomial}\big(1, \mathbf{g}\,(\mathbf{a}_i)\big), & \begin{cases} \mathrm{rank}([\mathbf{A}^\star \parallel \mathbf{B}^\star]) \leq r & \textit{for SMF-}\mathbf{H}, \\ \mathrm{rank}([\mathbf{A}^\star, \mathbf{B}^\star]) \leq r & \textit{for SMF-}\mathbf{W}. \end{cases} \\ \mathbf{a}_i = \begin{cases} \mathbf{A}^\star[:, i] + (\boldsymbol{\gamma}^\star)^T \mathbf{x}_i' & \textit{SMF-}\mathbf{H}, \\ (\mathbf{A}^\star)^T \mathbf{x}_i + (\boldsymbol{\gamma}^\star)^T \mathbf{x}_i' & \textit{SMF-}\mathbf{W}, \end{cases} \end{cases} \tag{17}$$

where each $\boldsymbol{\varepsilon}_i$ (resp., $\boldsymbol{\varepsilon}_i'$) are $p \times 1$ (resp., $q \times 1$) vector of i.i.d. mean zero Gaussian entries with variance $\sigma^2$ (resp., $(\sigma')^2$). We call the above the *generative SMF model*. In what follows, we will assume that the noise levels $\sigma$ and $\sigma'$ are known and focus on estimating the four-parameter matrices.

The ($L_2$-regularized) negative log-likelihood of observing triples $(y_i, \mathbf{x}_i, \mathbf{x}_i')$ for $i = 1, \ldots, n$ is given as $\mathcal{L}_n := F(\mathbf{A}, \mathbf{B}, \boldsymbol{\gamma}) + \frac{1}{2(\sigma')^2} \|\mathbf{X}_{\text{aux}} - \mathbf{C}\|_F^2 + c$, where $c$ is a constant and $F$ is as in (11) or (12) depending on the activation type with tuning parameter $\xi = \frac{1}{2\sigma^2}$. The $L_2$ regularizer in $F$ can be understood as Gaussian prior for the parameters and interpreting the right-hand side above as the negative logarithm of the posterior distribution function (up to a constant) in a Bayesian framework. Note that the problem of estimating $\mathbf{A}$ and $\mathbf{B}$ are coupled due to the low-rank model assumption in (17), while the problem of estimating $\mathbf{C}$ is standard and separable, so it is not of our interest. The joint estimation problem for $[\mathbf{A}, \mathbf{B}, \boldsymbol{\gamma}]$ is equivalent to the corresponding SMF problem (4) with tuning parameter $\xi = (2\sigma^2)^{-1}$. This and Theorem 3.5 motivate us to estimate the true parameters $\mathbf{A}^\star, \mathbf{B}^\star$, and $\boldsymbol{\gamma}^\star$ by the output of Algorithm 1 with $\xi = (2\sigma^2)^{-1}$ for $O(\log n)$ iterations.

Now we give the second main result. Roughly speaking, it states that the estimated parameter $\mathbf{Z}_t$ is within the true parameter $\mathbf{Z}^\star = [\mathbf{A}^\star, \mathbf{B}^\star, \boldsymbol{\gamma}^\star]$ within $O(\log n/\sqrt{n})$ with high probability, provided that the noise variance $\sigma^2$ is small enough and the SMF objective (11)-(12) is well-conditioned.

**Theorem 4.1.** *(Statistical estimation for SMF) Assume the model* (17) *with fixed $p$. Suppose Assumptions 3.1-3.3 hold. Let $\mu, L$ be as in* (15), $\rho := 2(1 - \tau\mu)$ *and $c = O(1)$ if $\mathbf{Z}^\star - \tau\nabla_\mathbf{Z} F(\mathbf{Z}^\star) \in \boldsymbol{\Theta}$ and $c = O(\sqrt{\min(p, n)})$ otherwise. Let $\mathbf{Z}_t$ denote the iterates of Algorithm 1 with the tuning parameter $\xi = (2\sigma^2)^{-1}$, $L_2$-regularization parameter $\lambda > 0$, and stepsize $\tau \in (\frac{1}{2\mu}, \frac{3}{2L})$. Then following holds with probability at least $1 - \frac{1}{n}$: For all $t \geq 1$ and $n \geq 1$, $\|\mathbf{Z}_t - \mathbf{Z}^\star\|_F - \rho^t \|\mathbf{Z}_0 - \mathbf{Z}^\star\|_F \leq c\frac{(\sqrt{n}\log n + \lambda)}{\mu}$, provided $L/\mu < 3$. Furthermore, $c\frac{(\sqrt{n}\log n + \lambda)}{\mu}$ is $O(\log n/\sqrt{n})$ if $\mathbf{Z}^\star - \tau\nabla_\mathbf{Z} F(\mathbf{Z}^\star) \in \boldsymbol{\Theta}$ and $\sigma^2 = O(1/n)$.*

We remark that Theorem 4.1 implies that *SMF-$\mathbf{H}$ is statistically more robust than SMF-$\mathbf{W}$*. Namely, in order to have an arbitrarily accurate estimate with high probability, one needs to have $\mu \gg \sqrt{n}\log n$. Combining with the expression in (15) and the well-conditioning assumption $L/\mu < 3$, one needs to require $\xi = \Omega(n)$, hence small noise variance $\sigma^2 = O(1/n)$ for SMF-$\mathbf{W}$. However, for SMF-$\mathbf{H}$, this is guaranteed whenever $\sigma^2 = o(1/(\sqrt{n}\log n))$ and $\lambda \approx \mu$.

## 5   Simulation and Numerical Validation

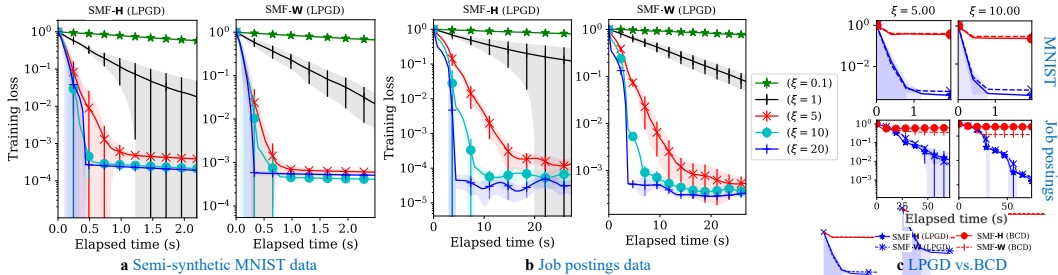

a Semi-synthetic MNIST data    b Job postings data    c LPGD vs.BCD

Figure 2: (**a-b**) Training loss vs. elapsed CPU time for Algorithm 1 (with binary logistic classifier) on the semi-synthetic MNIST and Job postings datasets for several values of $\xi$ in log scale. Average training loss over ten runs and the shades representing the standard deviation shown. (**c**) Comparison between LPGD (Algorithm 1) and BCD algorithms for SMF.

We numerically verify Theorem 3.5 on a semi-synthetic dataset generated by using MNIST image dataset [24] ($p = 28^2 = 784$, $q = 0$, $n = 500$, $\kappa = 1$) and a text dataset named 'Real / Fake Job Posting Prediction' [1] ($p = 2840$, $q = 72$, $n = 17880$, $\kappa = 1$). Details about these datasets are in Sec. G in Appendix.[1] We used Algorithm 1 with rank $r = 2$ for MNIST and $r = 20$ for job postings datasets. For all experiments, $\lambda = 2$ and stepsize $\tau = 0.01$ were used.

We validate the theoretical exponential convergence results of our LPGD algorithm (Algorithm 1) in Figure 2. Note that the convexity and smoothness parameters $\mu$ and $L$ in Theorem 3.5 are difficult to compute exactly. In practice, cross-validation of hyperparameters is usually employed. For

---

[1]We provide our implementation of Algorithm 1 in our code repository https://github.com/ljw9510/SMF/tree/main.

$\xi \in \{0.1, 1, 5, 10, 20\}$ in Figure 2, we indeed observe exponential decay of training loss as dictated by our theoretical results for Algorithm 1. We also observe that the exponential rate of decay in training loss increases as $\xi$ increases. According to Theorem 3.5, the contraction coefficient is $\rho = (1 - \tau\mu)$, which decreases in $\xi$ since $\mu$ increases in $\xi$ (see (15)). The decay for large $\xi \in \{10, 20\}$ seems even superexponential. Furthermore, 2c shows that our LPGD algorithm converges significantly faster than BCD for training both SMF-**H** and SMF-**W** models at $\xi \in \{5, 10\}$ (other values of $\xi$ omitted).

## 6   Application: Microarray Analysis for Cancer Classification

We apply the proposed methods to two datasets from the Curated Microarray Database (CuMiDa) [14]. CuMiDa provides well-preprocessed microarray data for various cancer types for various machine-learning approaches. One consists of 54,676 gene expressions from 51 subjects with binary labels indicating pancreatic cancer; Another we use has 35,982 gene expressions from 289 subjects with binary labels indicating breast cancer. The primary purpose of the analysis is to classify cancer patients solely based on their gene expression. We compare the accuracies of the proposed methods – SMF-**W** and SMF-**H** with a binary logistic classifier trained using Algorithm 1 – against the following benchmark algorithms: SMF-**W** and SMF-**H** trained using BCD; 1-dimensional seven-layer Convolutional Neural Networks (CNN); three-layer Feed-Forward Neural Networks (FFNN); Naive Bayes (NB); Support Vector Machine (SVM); Random Forest (RF); Logistic Regression with Matrix Factorization by truncated SVD (MF-LR). For the last benchmark method MF-LR, we use rank-$r$ SVD to factorize $\mathbf{X}_{\text{train}} \approx \mathbf{U}\Sigma\mathbf{V}^T$ and take $\mathbf{W} = \mathbf{U}\Sigma$ and $\mathbf{H} = \mathbf{V}^T$. For testing, we use $\mathbf{W}^T\mathbf{X}_{\text{test}}$ as input to logistic regression for both filter and feature methods since $\|\mathbf{X}_{\text{test}} - \mathbf{W}\mathbf{H}_{\text{test}}\|_F$ is minimized when $\mathbf{H}_{\text{test}} = (\mathbf{W}^T\mathbf{W})^{-1}\mathbf{W}^T\mathbf{X}_{\text{test}} = \mathbf{W}^T\mathbf{X}_{\text{test}}$ with orthogonal $\mathbf{W}$.

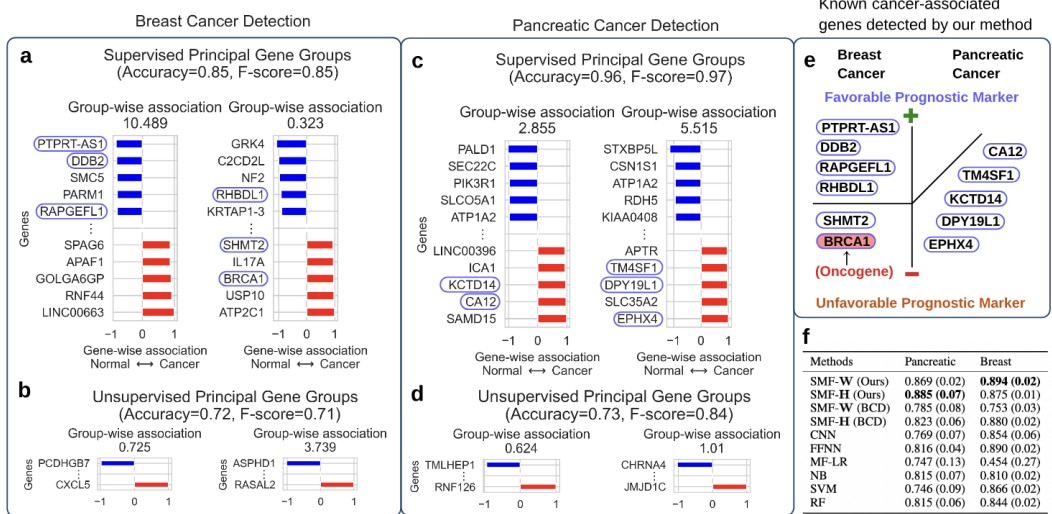

Figure 3: (**a-b**) Two selected supervised/unsupervised principal gene groups (low-dimensional compression of genes) learned by rank-16 SMF-**W**/SVD and their associated logistic regression coefficients for breast cancer detection. (**c-d**) Similar to **a-b** learned by rank-2 SMF-**W**/SVD for pancreatic cancer detection. (**e**) Blue-circled genes within each gene group of extreme coefficients coincide with known prognostic markers (for pancreatic cancer) and oncogene (for breast cancer). (**f**) Average classification accuracies and their standard deviations (in parenthesis) for various methods on two cancer microarray datasets over five-fold cross-validation. The highest-performing instances are marked in bold.

We normalize gene expression for stable matrix factorization and interpretability of regression coefficients. We split each data into 50% of the training set and 50% of the test set and repeat the comparison procedure 5 times. A scree plot is used to determine the rank $r$. Other parameters are chosen through 5-fold cross-validation ($\xi \in \{0.1, 1, 10\}$ and $\lambda \in \{0.1, 1, 10\}$), and the algorithms are repeated in 1,000 iterations or until convergence. As can be seen in the table in Figure 3**a**, the proposed methods show the best performance for both types of cancers.

An important advantage of SMF methods is that they provide interpretable results in the form of 'supervised factors'. Each supervised factor consists of a latent factor and the associated regression coefficient. That is, once we train the SMF model (for $\kappa = 1$) and learn factor matrix $\mathbf{W} = [\mathbf{w}_1, \ldots, \mathbf{w}_r] \in \mathbb{R}^{p \times r}$ and vector of regression coefficients $\boldsymbol{\beta} = [\beta_1, \ldots, \beta_r] \in \mathbb{R}^{1 \times r}$, each column $\mathbf{w}_j$ of $\mathbf{W}$ describes a latent factor and the corresponding regression coefficient $\beta_j$ tells us how $\mathbf{w}_j$ is associated with class labels. The pairs $(\mathbf{w}_j, \beta_j)$, which form supervised latent factors, provide insights into how the trained SMF model perceives the classification task. See Fig. 1 for illustration.

In the context of microarray analysis for cancer research, each $\mathbf{w}_j$ corresponds to a weighted group of genes (which we call a 'principal gene group') and $\beta_j$ represents the strength of its association with cancer. SMF learns supervised gene groups (Fig. 3**a, c**) with significantly higher classification accuracy than the unsupervised gene groups (Fig. 3**b, d**). In Fig. 3**a, c**, both gene groups (consisting of $p$ genes) have positive regression coefficients, so they are positively associated with the log odds of the predictive probability of the corresponding cancer. Remarkably, our method detected the well-known oncogene BRCA1 of breast cancer and other various genes (in Fig. 3**e**) that are known to be prognostic markers of breast/pancreatic cancer (see Human Protein Atlas [49]) in these groups of extreme coefficients (top five). The high classification accuracy suggests that the identified supervised principal gene groups may be associated with the occurrence of breast/pancreatic cancer.

## 7 Conclusion and Limitations

We propose an exponentially convergent algorithm for nonconvex SMF training using new lifting techniques. Our analysis demonstrates strong convergence and estimation guarantee. We compare the robustness of filter-based and feature-based SMF, finding that the former is computationally more robust while the latter is statistically more robust. The algorithm's exponential convergence is numerically verified. In cancer classification using microarray data analysis, our algorithm successfully identifies discriminative gene groups for various cancers and shows potential for identifying important gene groups as protein complexes or pathways in biomedical research. Our analysis framework can be extended to more complex classification models, such as combining a feed-forward deep neural network with a matrix factorization objective. While our convergence analysis holds in certain parameter regimes, we discuss them in detail. We have tested our method and convergence analysis on various real-world datasets but recommend further numerical verification on a wider range of datasets.

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
