## A Gradient computation for executing the main algorithm

A straightforward computation shows (recall that $\boldsymbol{\theta} = [\mathbf{A} \parallel \mathbf{B}]$ for SMF-$\mathbf{H}$ and $\boldsymbol{\theta} = [\mathbf{A}, \mathbf{B}]$ for SMF-$\mathbf{W}$)

$$\nabla_{\mathrm{vec}(\mathbf{A})} F - 2\lambda \, \mathrm{vec}(\mathbf{A}) = \begin{cases} \sum_{s=1}^{n} \nabla_{\mathbf{a}} \ell(y_s, \mathbf{a}_s) \otimes \mathbf{I}_n[:, s] & \text{for SMF-}\mathbf{H} \\ \sum_{s=1}^{n} \nabla_{\mathbf{a}} \ell(y_s, \mathbf{a}_s) \otimes \mathbf{x}_s & \text{for SMF-}\mathbf{W}, \end{cases} \tag{18}$$

$$\nabla_{\mathbf{B}} F = 2\xi(\mathbf{B} - \mathbf{X}_{\mathrm{data}}), \qquad \nabla_{\mathrm{vec}(\boldsymbol{\gamma})} F = \left( \sum_{s=1}^{n} \nabla_{\mathbf{a}} \ell(y_s, \mathbf{a}_s) \otimes \mathbf{x}_s' \right) + 2\lambda \, \mathrm{vec}(\boldsymbol{\gamma}), \tag{19}$$

where $\otimes$ denotes the Kronecker product. Here, we have $\nabla_{\mathbf{a}} \ell(y, \mathbf{a}) = (\dot{h}_1, \ldots, \dot{h}_\kappa)$, where

$$\dot{h}_j := \frac{h'(a_j)}{1 + \sum_{c=1}^{\kappa} h(a_c)} - \mathbf{1}_{\{y=j\}} \frac{h'(a_j)}{h(a_j)}. \tag{20}$$

## B Generalized multinomial logistic regression

In this section, we provide some background on a generalized multinomial logistic regression and record some useful computations. (See [10] for backgrounds on multinomial logistic regression.) Without loss of generality, we can assume that the $\kappa + 1$ classes are the integers in $\{0, 1, \ldots, \kappa\}$. Say we have training examples $(\boldsymbol{\phi}(\mathbf{x}_1), y_1), \ldots, (\boldsymbol{\phi}(\mathbf{x}_n), y_n)$, where

- $\mathbf{x}_1, \ldots, \mathbf{x}_n$: Input data (e.g., collection of all medical records of each patient)
- $\boldsymbol{\phi}_1 := \boldsymbol{\phi}(\mathbf{x}_1), \ldots, \boldsymbol{\phi}_n := \boldsymbol{\phi}(\mathbf{x}_n) \in \mathbb{R}^p$ : Features (e.g., some useful information for each patient)
- $y_1, \ldots, y_n \in \{0, 1, \ldots, \kappa\}$: $\kappa + 1$ class labels (e.g., digits from 0 to 9).

The basic idea of multinomial logistic regression is to model the output $y$ as a discrete random variable $Y$ with probability mass function $\mathbf{p} = [p_0, p_1, \ldots, p_\kappa]$ that depends on the observed feature $\boldsymbol{\phi}(\mathbf{x})$, score function $h : \mathbb{R} \to \mathbb{R}$ (strictly increasing, twice differentiable, and $h(0) = 1$), and a matrix parameter $\mathbf{W} = [\mathbf{w}_1, \ldots, \mathbf{w}_\kappa] \in \mathbb{R}^{p \times \kappa}$ through the following relation:

$$p_0 = \frac{1}{1 + \sum_{c=1}^{\kappa} h(\langle \boldsymbol{\phi}(\mathbf{x}), \mathbf{w}_c \rangle)}, \quad p_j = \frac{h(\langle \boldsymbol{\phi}(\mathbf{x}), \mathbf{w}_j \rangle)}{1 + \sum_{c=1}^{\kappa} h(\langle \boldsymbol{\phi}(\mathbf{x}), \mathbf{w}_c \rangle)}, \quad \text{for } j = 1, \ldots, \kappa. \tag{21}$$

That is, given the feature vector $\boldsymbol{\phi}(\mathbf{x})$, the probability $p_i$ of $\mathbf{x}$ having label $i$ is proportional to $h$ evaluated at the 'linear activation' $\langle \boldsymbol{\phi}(\mathbf{x}), \mathbf{w}_i \rangle$ with the base category of class 0. Note that using $h(x) = \exp(x)$, the above multiclass classification model reduces to the classical multinomial logistic regression. In this case, the corresponding predictive probability distribution $\mathbf{p}$ is called the *softmax distribution* with activation $\mathbf{a} = [a_1, \ldots, a_\kappa]$ with $a_i = \langle \boldsymbol{\phi}(\mathbf{x}), \mathbf{w}_i \rangle$ for $i = 1, \ldots, \kappa$. Notice that this model has parameter vectors $\mathbf{w}_1, \ldots, \mathbf{w}_\kappa \in \mathbb{R}^p$, one for each of the $\kappa$ nonzero class labels.

Next, we derive the maximum log-likelihood formulation for finding optimal parameter $\mathbf{W}$ for the given training set $(\boldsymbol{\phi}_i, y_i)_{i=1,\ldots,n}$. For each $1 \leq i \leq n$, define the predictive probability mass function $\mathbf{p}_i = [p_{i0}, p_{i1}, \ldots, p_{i\kappa}]$ using (21) with $\boldsymbol{\phi}(\mathbf{x})$ replaced by $\boldsymbol{\phi}_i$. We introduce the following matrix notations

$$\mathbf{Y} := \begin{bmatrix} \mathbf{1}(y_1 = 1) & \cdots & \mathbf{1}(y_1 = \kappa) \\ \vdots & & \vdots \\ \mathbf{1}(y_n = 1) & \cdots & \mathbf{1}(y_n = \kappa) \end{bmatrix}, \quad \mathbf{P} := \begin{bmatrix} p_{11} & \cdots & p_{1\kappa} \\ \vdots & & \vdots \\ p_{n1} & \cdots & p_{n\kappa} \end{bmatrix} \tag{22}$$

$$\in \{0, 1\}^{n \times \kappa} \qquad\qquad\qquad \in [0, 1]^{n \times \kappa}$$

$$\boldsymbol{\Phi} := \begin{bmatrix} \uparrow & & \uparrow \\ \boldsymbol{\phi}(\mathbf{x}_1) & \cdots & \boldsymbol{\phi}(\mathbf{x}_n) \\ \downarrow & & \downarrow \end{bmatrix}, \quad \mathbf{W} := \begin{bmatrix} \uparrow & & \uparrow \\ \mathbf{w}_1 & \cdots & \mathbf{w}_\kappa \\ \downarrow & & \downarrow \end{bmatrix}. \tag{23}$$

$$\in \mathbb{R}^{p \times n} \qquad\qquad\qquad \in \mathbb{R}^{p \times \kappa}$$

Note that the $s$th row of $\mathbf{Y}$ is a zero vector if and only if $y_s = 0$. Similarly, since $p_{s0} = 1 - (p_{s1} + \cdots + p_{s\kappa})$, the corresponding row of $\mathbf{P}$ determines its predictive probability distribution. Then the joint likelihood function of observing labels $(y_1, \ldots, y_n)$ given input data $(\mathbf{x}_1, \ldots, \mathbf{x}_n)$ under the above probabilistic model is

$$L(y_1, \ldots, y_n\,;\,\mathbf{W}) = \mathbb{P}(Y_1 = y_1, \ldots, Y_n = y_n\,;\,\mathbf{W}) = \prod_{s=1}^{n} \prod_{j=0}^{\kappa} (p_{sj})^{\mathbf{1}(y_s = j)}. \tag{24}$$

Denote $\mathbf{w}_0 = \mathbf{0}$. Then since $h(0) = 1$ by definition, we can conveniently write

$$p_{sj} = \frac{h(\langle \boldsymbol{\phi}_s, \mathbf{w}_j \rangle)}{\sum_{c=0}^{\kappa} h(\langle \boldsymbol{\phi}_s, \mathbf{w}_c \rangle)} \quad \text{for } s = 1, \ldots, n \text{ and } j = 0, 1, \ldots, \kappa. \tag{25}$$

Now we can derive the negative log-likelihood $\ell(\boldsymbol{\Phi}, \mathbf{W}) := -\sum_{s=1}^{n} \sum_{j=0}^{\kappa} \mathbf{1}(y_s = j) \log p_{sj}$ in a matrix form as follows:

$$\ell(\boldsymbol{\Phi}, \mathbf{W}) = \sum_{s=1}^{n} \log \left( 1 + \sum_{c=1}^{\kappa} h(\langle \boldsymbol{\phi}(\mathbf{x}_s), \mathbf{w}_c \rangle) \right) - \sum_{s=1}^{n} \sum_{j=0}^{\kappa} \mathbf{1}(y_s = j) \log h(\langle \boldsymbol{\phi}(\mathbf{x}_s), \mathbf{w}_j \rangle) \tag{26}$$

$$= \left( \sum_{s=1}^{n} \log \left( 1 + \sum_{c=1}^{\kappa} h(\langle \boldsymbol{\phi}(\mathbf{x}_s), \mathbf{w}_c \rangle) \right) \right) - \operatorname{tr}\left( \mathbf{Y}^T h(\boldsymbol{\Phi}^T \mathbf{W}) \right), \tag{27}$$

where $\operatorname{tr}(\cdot)$ denotes the trace operator. Then the maximum likelihood estimates $\hat{\mathbf{W}}$ is defined as the minimizer of the above loss function in $\mathbf{W}$ while fixing the feature matrix $\boldsymbol{\Phi}$.

Both the maps $\mathbf{W} \mapsto \ell(\boldsymbol{\Phi}, \mathbf{W})$ and $\boldsymbol{\Phi} \mapsto \ell(\boldsymbol{\Phi}, \mathbf{W})$ are convex and we can compute their gradients as well as the Hessian explicitly as follows. For each $y \in \{0, 1, \ldots \kappa\}$, $\boldsymbol{\phi} \in \mathbb{R}^p$, and $\mathbf{W} \in \mathbb{R}^{p \times \kappa}$, define vector and matrix functions

$$\dot{\mathbf{h}}(y, \boldsymbol{\phi}, \mathbf{W}) := (\dot{h}_1, \ldots, \dot{h}_\kappa)^T \in \mathbb{R}^{\kappa \times 1}, \quad \dot{h}_j := \frac{h'(\langle \boldsymbol{\phi}, \mathbf{w}_j \rangle)}{1 + \sum_{c=1}^{\kappa} h(\langle \boldsymbol{\phi}, \mathbf{w}_c \rangle)} - \mathbf{1}(y = j) \frac{h'(\langle \boldsymbol{\phi}, \mathbf{w}_j \rangle)}{h(\langle \boldsymbol{\phi}, \mathbf{w}_j \rangle)} \tag{28}$$

$$\ddot{\mathbf{H}}(y, \boldsymbol{\phi}, \mathbf{W}) := \left( \ddot{\mathbf{H}}_{ij} \right)_{i,j} \in \mathbb{R}^{\kappa \times \kappa}, \tag{29}$$

$$\ddot{\mathbf{H}}_{ij} = \frac{h''(\langle \boldsymbol{\phi}, \mathbf{w}_j \rangle) \mathbf{1}(i=j)}{1 + \sum_{c=1}^{\kappa} h(\langle \boldsymbol{\phi}, \mathbf{w}_c \rangle)} - \frac{h'(\langle \boldsymbol{\phi}, \mathbf{w}_i \rangle) h'(\langle \boldsymbol{\phi}, \mathbf{w}_j \rangle)}{\left( 1 + \sum_{c=1}^{\kappa} h(\langle \boldsymbol{\phi}, \mathbf{w}_c \rangle) \right)^2} - \mathbf{1}(y = i = j) \left( \frac{h''(\langle \boldsymbol{\phi}, \mathbf{w}_j \rangle)}{h(\langle \boldsymbol{\phi}, \mathbf{w}_j \rangle)} - \frac{(h'(\langle \boldsymbol{\phi}, \mathbf{w}_j \rangle))^2}{(h(\langle \boldsymbol{\phi}, \mathbf{w}_j \rangle))^2} \right). \tag{30}$$

For each $\mathbf{W} = [\mathbf{w}_1, \ldots, \mathbf{w}_\kappa] \in \mathbb{R}^{p \times \kappa}$, let $\mathbf{W}^{\mathrm{vec}} := [\mathbf{w}_1^T, \ldots, \mathbf{w}_\kappa^T]^T \in \mathbb{R}^{p\kappa}$ denote its vectorization. Then a straightforward computation shows

$$\nabla_{\mathrm{vec}(\mathbf{W})} \ell(\boldsymbol{\Phi}, \mathbf{W}) = \sum_{s=1}^{n} \dot{\mathbf{h}}(y_s, \boldsymbol{\phi}_s, \mathbf{W}) \otimes \boldsymbol{\phi}_s, \tag{31}$$

$$\mathbf{H} := \nabla_{\mathrm{vec}(\mathbf{W})} \nabla_{\mathrm{vec}(\mathbf{W})^T} \ell(\boldsymbol{\Phi}, \mathbf{W}) = \sum_{s=1}^{n} \ddot{\mathbf{H}}(y_s, \boldsymbol{\phi}_s, \mathbf{W}) \otimes \boldsymbol{\phi}_s \boldsymbol{\phi}_s^T, \tag{32}$$

where $\otimes$ above denotes the Kronecker product. Recall that the eigenvalues of $\mathbf{A} \otimes \mathbf{B}$, where $\mathbf{A}$ and $\mathbf{B}$ are two square matrices, are given by $\lambda_i \mu_j$, where $\lambda_i$ and $\mu_j$ run over all eigenvalues of $\mathbf{A}$ and $\mathbf{B}$, respectively. Also, for two square matrices $\mathbf{A}, \mathbf{B}$ of the same size, write $\mathbf{A} \preceq \mathbf{B}$ if $v^T \mathbf{A} v \le v^T \mathbf{B} v$ for all unit vectors $v$. Then denoting $\lambda^+ := \max_{1 \le s \le n} \lambda_{\max}(\ddot{\mathbf{H}}(y_s, \boldsymbol{\phi}_s, \mathbf{W}))$,

$$\mathbf{H} \preceq \lambda^+ \sum_{s=1}^{n} \mathbf{I} \otimes \boldsymbol{\phi}_s \boldsymbol{\phi}_s^T = \lambda^+ \mathbf{I} \otimes \boldsymbol{\Phi} \boldsymbol{\Phi}^T. \tag{33}$$

Similarly, $\lambda^- \mathbf{I} \otimes \boldsymbol{\Phi} \boldsymbol{\Phi}^T \preceq \mathbf{H}$, where $\lambda^-$ denotes the minimum over all $\lambda_{\min}(\ddot{\mathbf{H}}(y_s, \boldsymbol{\phi}_s, \mathbf{W}))$. Hence we can deduce

$$\lambda^- \lambda_{\min}\left( \boldsymbol{\Phi} \boldsymbol{\Phi}^T \right) \le \lambda_{\min}(\mathbf{H}) \le \lambda_{\max}(\mathbf{H}) \le \lambda^+ \lambda_{\max}\left( \boldsymbol{\Phi} \boldsymbol{\Phi}^T \right). \tag{34}$$

There are some particular cases worth noting. First, suppose binary classification case, $\kappa = 1$. Then the Hessian $\mathbf{H}$ above reduces to

$$\mathbf{H} = \sum_{s=1}^{n} \ddot{\mathbf{H}}_{11}(y_s, \boldsymbol{\phi}_s, \mathbf{W})\boldsymbol{\phi}_s\boldsymbol{\phi}_s^T. \tag{35}$$

Second, let $h(x) = \exp(x)$ and consider the multinomial logistic regression case. Then $h = h' = h''$ so the above yields the following concise matrix expression

$$\nabla_{\mathbf{W}} \ell(\boldsymbol{\Phi}, \mathbf{W}) = \boldsymbol{\Phi}(\mathbf{P} - \mathbf{Y}) \in \mathbb{R}^{p \times \kappa}, \qquad \nabla_{\boldsymbol{\Phi}} \ell(\boldsymbol{\Phi}, \mathbf{W}) = \mathbf{W}(\mathbf{P} - \mathbf{Y})^T \in \mathbb{R}^{p \times n}, \tag{36}$$

$$\mathbf{H} = \sum_{s=1}^{n} \begin{bmatrix} p_{s1}(1 - p_{s1}) & -p_{s1}p_{s2} & \cdots & -p_{s1}p_{s\kappa} \\ -p_{s2}p_{s1} & p_{s2}(1 - p_{s2}) & \cdots & -p_{s2}p_{s\kappa} \\ \vdots & \vdots & \ddots & \vdots \\ -p_{s\kappa}p_{s1} & -p_{s\kappa}p_{s2} & \cdots & p_{s\kappa}(1 - p_{s\kappa}) \end{bmatrix} \otimes \boldsymbol{\phi}_s\boldsymbol{\phi}_s^T. \tag{37}$$

Note that $\mathbf{H}$ in this case does not depend on $y_s$ for $s = 1, \ldots, n$. The bounds on the eigenvalues depends on the range of linear activation $\langle \phi_i, \mathbf{w}_j \rangle$ may take. For instance, if we restrict the norms of the input feature vector $\phi_i$ and parameter $\mathbf{w}_j$, then we can find a suitable positive uniform lower bound on the eigenvalues of $\mathbf{H}$.

**Lemma B.1.** *Suppose $h(\cdot) = \exp(\cdot)$. Then*

$$\lambda_{\min}\left(\ddot{\mathbf{H}}(\boldsymbol{\phi}_s, \mathbf{W})\right) \geq \min_{1 \leq i \leq \kappa} \frac{\exp(\langle \boldsymbol{\phi}_s, \mathbf{w}_i \rangle)}{\left(1 + \sum_{c=1}^{\kappa} \exp(\langle \boldsymbol{\phi}_s, \mathbf{w}_c \rangle)\right)^2}, \tag{38}$$

$$\lambda_{\max}\left(\ddot{\mathbf{H}}(\boldsymbol{\phi}_s, \mathbf{W})\right) \leq \max_{1 \leq i \leq \kappa} \frac{\exp(\langle \boldsymbol{\phi}_s, \mathbf{w}_i \rangle)}{\left(1 + \sum_{c=1}^{\kappa} \exp(\langle \boldsymbol{\phi}_s, \mathbf{w}_c \rangle)\right)^2} \left(1 + 2\sum_{c=2}^{\kappa} \exp(\langle \boldsymbol{\phi}_s, \mathbf{w}_c \rangle)\right). \tag{39}$$

*Proof.* For the lower bound on the minimum eigenvalue, we note that

$$\lambda_{\min}\left(\ddot{\mathbf{H}}(\boldsymbol{\phi}_s, \mathbf{W})\right) \geq \min_{1 \leq i \leq \kappa} \sum_{j=1}^{\kappa} \ddot{H}_{ij} = \min_{1 \leq i \leq \kappa} p_{si}p_{s0} = \min_{1 \leq i \leq \kappa} \frac{\exp(\langle \boldsymbol{\phi}_s, \mathbf{w}_i \rangle)}{\left(1 + \sum_{c=1}^{\kappa} \exp(\langle \boldsymbol{\phi}_s, \mathbf{w}_c \rangle)\right)^2}, \tag{40}$$

where the first inequality was shown in [4] using the fact that $\ddot{\mathbf{H}}(\boldsymbol{\phi}_s, \mathbf{W})$ is a diagonally dominant $M$-matrix (see [50]). The following equalities can be verified easily.

For the upper bound on the maximum eigenvalue, we use the Gershgorin circle theorem (see, e.g., [19]) to bound

$$\lambda_{\max}\left(\ddot{\mathbf{H}}(\boldsymbol{\phi}_s, \mathbf{W})\right) \leq \max_{1 \leq i \leq \kappa} \left(p_{si}(1 - p_{si}) + \sum_{c \neq i} p_{si}p_{sc}\right) \leq \max_{1 \leq i \leq \kappa} p_{si}(2 - p_{s0} - 2p_{si}). \tag{41}$$

Then simplifying the last expression gives the assertion. $\qquad\square$

## C  Exponential convergence of low-rank PGD

In Section 2.1 of the main manuscript, we outlined our key idea for solving the SMF problem (4), which involves 'double lifting' the nonconvex problem to a low-rank matrix estimation problem. In this section, we make this approach precise by considering an abstract form of optimization problems that generalizes the SMF problem (4).

Fix a function $f : \mathbb{R}^{d_1 \times d_2} \times \mathbb{R}^{d_3 \times d_4} \to \mathbb{R}$, which takes the input of a $d_1 \times d_2$ matrix and an augmented variable in $\mathbb{R}^{d_3 \times d_4}$. Consider the following *constrained and augmented low-rank estimation* (CALE) problem

$$\min_{\mathbf{z} = [\mathbf{X}, \boldsymbol{\Gamma}] \in \subseteq \mathbb{R}^{d_1 \times d_2} \times \mathbb{R}^{d_3 \times d_4}} f(\mathbf{Z}), \qquad \text{subject to } \mathbf{Z} \in \boldsymbol{\Theta} \text{ and } \text{rank}(\mathbf{X}) \leq r, \tag{42}$$

where $\boldsymbol{\Theta}$ is a convex subset of $\mathbb{R}^{d_1 \times d_2} \times \mathbb{R}^{d_3 \times d_4}$. Here, we seek to find a global minimizer $\mathbf{Z}^\star = [\mathbf{X}^\star, \boldsymbol{\Gamma}^\star]$ of the objective function $f$ over the convex set $\boldsymbol{\Theta}$, consisting of a low-rank component

$\mathbf{X}^\star \in \mathbb{R}^{d_1 \times d_2}$ and an auxiliary variable $\mathbf{\Gamma}^\star \in \mathbb{R}^{d_3 \times d_4}$. In a statistical inference setting, the loss function $f = f_n$ may be based on $n$ noisy observations according to a probabilistic model, and the true parameter $\mathbf{Z}^*$ to be estimated may approximately minimize $f$ over the constraint set $\mathbf{\Theta}$, with some statistical error $\varepsilon(n)$ depending on the sample size $n$. In this case, a global minimizer $\mathbf{Z}^\star \in \arg\min_{\mathbf{\Theta}} f$ serves as an estimate of the true parameter $\mathbf{Z}^*$. The matrix completion and low-rank matrix estimation problem [34, 46] can be considered as special cases of (42) without constraint $\mathbf{\Theta}$ and the auxiliary variable $\mathbf{\Gamma}$. This problem setting has been one of the most important research topics in the machine learning and statistics literature for the past few decades. More importantly for our purpose, we have seen in (11) and (12) in the main manuscript that both the feature- and filter-based SMF problems can be cast as the form of (42) after some lifting and change of variables.

One can reformulate (42) as the following nonconvex problem, where one parameterizes the low-rank matrix variable $\mathbf{X}$ with product $\mathbf{UV}^T$ of two matrices, which we call the *constrained and augmented factored estimation* (CAFE) problem:

$$\min_{\mathbf{U} \in \mathbb{R}^{d_1 \times r}, \mathbf{V} \in \mathbb{R}^{d_2 \times r}, \mathbf{\Gamma} \in \mathbb{R}^{d_3 \times d_4}} f(\mathbf{UV}^T, \mathbf{\Gamma}), \qquad \text{subject to } [\mathbf{UV}^T, \mathbf{\Gamma}] \in \mathbf{\Theta}. \qquad (43)$$

Note that a solution to (43) gives a solution to (42). Conversely, when considering (42) without any constraint on the first matrix component, the singular value decomposition of the first matrix component easily demonstrates that a solution to (42) is also a solution to (43). Recently, there has been a surge of progress in global guarantees of solving the factored problem (43) using various nonconvex optimization methods [20, 21, 63, 64, 53, 42, 56, 40, 41]. While most of the work considers (43) without the auxiliary variable and constraints, some studies consider specific types of convex constraints such as matrix norm bound. We consider $\mathbf{\Theta}$ to be a general convex constraint set.

It is common that the nonconvex factored problem (43) is introduced as a more efficient formulation of the convex problem (42). Interestingly, in the present work, we reformulate the four-factor nonconvex problem of SMF in (4) as a three-factor nonconvex CAFE problem in (43) and then realize it as a single-factor convex CALE problem in (42). We illustrated this connection briefly in Section 2.1 of the main manuscript.

In order to solve the CALE problem (42), consider the following *low-rank projected gradient descent* (LPGD) algorithm: (see Remark C.7 for more discussion on the use of projections $\Pi_\mathbf{\Theta}$ and $\Pi_r$)

$$\mathbf{Z}_t \leftarrow \Pi_r \left( \Pi_\mathbf{\Theta} \left( \mathbf{Z}_{t-1} - \tau \nabla f(\mathbf{Z}_{t-1}) \right) \right), \qquad (44)$$

where $\tau$ is a stepsize parameter, $\Pi_\mathbf{\Theta}$ denotes projection onto the convex constraint set $\mathbf{\Theta} \subseteq \mathbb{R}^{d_1 \times d_2} \times \mathbb{R}^{d_3 \times d_4}$, and $\Pi_r$ denotes the projection of the first matrix component onto matrices of rank at most $r$ in $\mathbb{R}^{d_1 \times d_2}$. More precisely, let $\mathbf{Z} = [\mathbf{X}, \mathbf{\Gamma}]$. Then $\Pi_r(\mathbf{Z}) := [\Pi_r(\mathbf{X}), \mathbf{\Gamma}]$. It is well-known that the rank-$r$ projection above can be explicitly computed by the singular value decomposition (SVD). Namely, $\Pi_r(\mathbf{X}) = \mathbf{U\Sigma V}^T$, where $\mathbf{\Sigma}$ is the $r \times r$ diagonal matrix of the top $r$ singular values of $\mathbf{X}$ and $\mathbf{U} \in \mathbb{R}^{d_1 \times r}$, $\mathbf{V} \in \mathbb{R}^{d_2 \times r}$ are semi-orthonormal matrices (i.e., $\mathbf{U}^T\mathbf{U} = \mathbf{V}^T\mathbf{V} = \mathbf{I}_r$). Note that algorithm (44) resembles the standard *projected gradient descent* (PGD) commonly used in the optimization literature. The algorithm follows a three-step procedure where a gradient descent step is performed, followed by projection onto the convex constraint set $\mathbf{\Theta}$ and subsequently the rank-$r$ projection. It is also worth noting the similarity of (44) to the 'lift-and-project' algorithm in [12] for structured low-rank approximation problem, which proceeds by alternatively applying the projections $\Pi_\mathbf{\Theta}$ and $\Pi_r$ to a given matrix until convergence.

In Theorem C.2, we will show that the iterate $\mathbf{Z}_t$ of the algorithm (44) converges exponentially to a low-rank approximation of the global minimizer of the objective $f$ over $\mathbf{\Theta}$, given that the objective $f$ satisfies the following restricted strong convexity (RSC) and restricted smoothness (RSM) properties in Definition C.1. These properties were first used in [3, 45, 36] for the class of matrix estimation problems and have found a number of applications in optimization and machine learning literature [56, 41, 51].

**Definition C.1.** (Restricted Strong Convexity and Smoothness) A function $f : \mathbb{R}^{d_1 \times d_2} \times \mathbb{R}^{d_3 \times d_4} \to \mathbb{R}$ is *r-restricted strongly convex and smooth* with parameters $\mu, L > 0$ if for all $\mathbf{Z}, \mathbf{Z}' \in \mathbb{R}^{d_1 \times d_2} \times \mathbb{R}^{d_3 \times d_4}$ whose matrix coordinates are of rank $\leq r$,

$$\frac{\mu}{2} \|\text{vec}(\mathbf{Z}) - \text{vec}(\mathbf{Z}')\|_2^2 \overset{\text{(RSC)}}{\leq} f(\mathbf{Z}') - f(\mathbf{Z}) - \langle \nabla f(\mathbf{Z}), \mathbf{Z}' - \mathbf{Z} \rangle \overset{\text{(RSM)}}{\leq} \frac{L}{2} \|\text{vec}(\mathbf{Z}) - \text{vec}(\mathbf{Z}')\|_2^2.$$

$$(45)$$

Next, we discuss optimality measures for the CALE problem (42). Recall that we want to minimize the objective $f$ subject to two constraints: (1) convex constraint $\Theta$ and (2) low-rank constraint. We first consider the following simpler problem without the low-rank constraint:

$$\min_{\mathbf{Z} \in \Theta} f(\mathbf{Z}). \tag{46}$$

A first-order optimal point $\mathbf{Z}^*$ for the above problem is called a *stationary point* of $f$ over $\Theta$, which is defined as

$$\langle \nabla f(\mathbf{Z}^*), \mathbf{Z} - \mathbf{Z}^* \rangle \geq 0 \quad \text{for all } \mathbf{Z} \in \Theta. \tag{47}$$

An alternative definition of stationary points uses gradient mapping [39, 7], which is particularly well-suited for projected gradient descent type algorithms. Define a map $G : \Theta \times (0, \infty) \to \mathbb{R}$ by

$$G(\mathbf{Z}, \tau) := \frac{1}{\tau}(\mathbf{Z} - \Pi_{\Theta}(\mathbf{Z} - \tau \nabla f(\mathbf{Z}))). \tag{48}$$

We call $G$ the *gradient mapping* associated with problem (46). In order to motivate the definition, fix $\mathbf{Z} \in \Theta$ and decompose it as

$$\mathbf{Z} = \Pi_{\Theta}(\mathbf{Z} - \tau \nabla f(\mathbf{Z})) + (\mathbf{Z} - \Pi_{\Theta}(\mathbf{Z} - \tau \nabla f(\mathbf{Z}))) \tag{49}$$

$$= \Pi_{\Theta}(\mathbf{Z} - \tau \nabla f(\mathbf{Z})) + \tau G(\mathbf{Z}, \tau). \tag{50}$$

Namely, the first term above is a one-step update of a projected gradient descent at $\mathbf{Z}$ over $\Theta$ with stepsize $\tau$, and the second term above is the error term. If $\mathbf{Z}$ is a stationary point of $f$ over $\Theta$, then $-\nabla f(\mathbf{Z})$ lies in the normal cone of $\Theta$ at $\mathbf{Z}$, so $\mathbf{Z}$ is invariant under the projected gradient descent and the error term above is zero. If $\mathbf{Z}$ is only approximately stationary, then the error above is nonzero. In fact, $G(\mathbf{Z}, \tau) = 0$ if and only if $\mathbf{Z}$ is a stationary point of $f$ over $\Theta$ (see Theorem 10.7 in [7]). Therefore, we may use the size of $G(\mathbf{Z}, \tau)$ (measured using an appropriate norm) as a measure of first-order optimality of $\mathbf{Z}$ for the problem (42). In the special cases when $\Theta$ is the whole space or when $\mathbf{Z}$ is in the interior of $\Theta$, if $\tau$ is sufficiently small (so that $\mathbf{Z} - \tau \nabla f(\mathbf{Z}) \in \Theta$), then $\|G(\mathbf{Z}, \tau)\|_F = \|\nabla f(\mathbf{Z})\|_F$, which is the standard measure of first-order optimality of $\mathbf{Z}$ for minimizing the objective $f$. In general, it holds that $\|G(\mathbf{Z}, \tau)\|_F \leq \|\nabla f(\mathbf{Z})\|_F$ (see Lemma F.1).

Now we turn our attention to (42). An optimal solution for (46) need not be an optimal solution for (42), since it may or may not satify the low-rank constraint. Our theoretical convergence guarantee of the LPGD algorithm (44) for CALE (42) covers these two cases.

**Theorem C.2.** *(Exponential convergence of LPGD) Let* $f : \mathbb{R}^{d_1 \times d_2} \times \mathbb{R}^{d_3 \times d_4} \to \mathbb{R}$ *be twice differentiable and $r$-restricted strongly convex and smooth with parameters $\mu$ and $L$, respectively, with $L/\mu < 3$. Let $(\mathbf{Z}_t)_{t \geq 0}$ be the iterates generated by algorithm (44). Suppose $\Theta \subseteq \mathbb{R}^{d_1 \times d_2} \times \mathbb{R}^{d_3 \times d_4}$ is a convex subset and fix a stepsize $\tau \in (\frac{1}{2\mu}, \frac{3}{2L})$. Then the 'contraction constant' $\rho := 2 \max(|1 - \tau\mu|, |1 - \tau L|) < 1$ and the followings hold:*

**(i)** *(Correctly specified case) Suppose $\mathbf{Z}^\star = [\mathbf{X}^\star, \mathbf{\Gamma}^\star]$ is a stationary point of $f$ over $\Theta$ such that $\text{rank}(\mathbf{X}^\star) \leq r$. Then $\mathbf{Z}^\star$ is the unique global minimizer of (42), $\lim_{t \to \infty} \mathbf{Z}_t = \mathbf{Z}^\star$, and for $t \geq 1$,*

$$\|\mathbf{Z}_t - \mathbf{Z}^\star\|_F \leq \rho^t \|\mathbf{Z}_0 - \mathbf{Z}^\star\|_F. \tag{51}$$

**(ii)** *(Possibly misspecified case) Let $\mathbf{Z}^\star = [\mathbf{X}^\star, \mathbf{\Gamma}^\star]$ be an arbitrary point in the interior of $\Theta$ with $\text{rank}(\mathbf{X}^\star) \leq r$. Then for $t \geq 1$,*

$$\|\mathbf{Z}_t - \mathbf{Z}^\star\|_F \leq \rho^t \|\mathbf{Z}_0 - \mathbf{Z}^\star\|_F + \frac{\tau}{1 - \rho}\left(\sqrt{3r}\|\nabla_{\mathbf{X}} f(\mathbf{Z}^\star)\|_2 + \|\nabla_{\mathbf{\Gamma}} f(\mathbf{Z}^\star)\|_F\right). \tag{52}$$

*In general, if $\mathbf{Z}^\star$ is an arbitrary point of $\Theta$ with $\text{rank}(\mathbf{X}^\star) \leq r$, then denoting the gradient mapping $[\Delta \mathbf{X}^\star, \Delta \mathbf{\Gamma}^\star] := \frac{1}{\tau}(\mathbf{Z}^\star - \Pi_{\Theta}(\mathbf{Z}^\star - \tau \nabla f(\mathbf{Z}^\star)))$ at $\mathbf{Z}^\star$, then for $t \geq 1$,*

$$\|\mathbf{Z}_t - \mathbf{Z}^\star\|_F \leq \rho^t \|\mathbf{Z}_0 - \mathbf{Z}^\star\|_F + \frac{2\tau}{1 - \rho}\left(\sqrt{3r}\|\Delta \mathbf{X}^\star\|_2 + \|\Delta \mathbf{\Gamma}^\star\|_F\right). \tag{53}$$

Theorem C.2 **(i)** asserts that the LPGD algorithm (44) converges at a linear rate to the unique global minimizer $\mathbf{Z}^\star$, provided that there exists a stationary point $\mathbf{Z}^\star$ of $f$ over the convex constraint set $\Theta$

with the first matrix factor $\mathbf{X}$ having rank at most $r$. This assumption holds in the standard statistical estimation setting, where one seeks to estimate a 'ground-truth' parameter $\mathbf{Z}^\star$ with a low-rank matrix factor from noisy observations. In this case, the objective $f$ represents the empirical error. Hence in this case, it is reasonable to assume that the gradient $\nabla f(\mathbf{Z}^\star)$ (in general, the gradient mapping $G(\mathbf{Z}, \tau)$) is small or at least $\mathbf{Z}^\star$ is near-stationary. In fact, Wang et al. [56, Condition 5.7] makes such an assumption.

In contrast, Theorem C.2 does not require such an assumption of near-optimality of the parameter $\mathbf{Z}^\star$ to be estimated. In practical situations, the rank of the ground-truth parameter is often unknown, and one attempts to explain observed data by using a low-rank model, in which case the assumed rank $r$ could be much lower than the true rank. For such generic situations, let $\mathbf{Z}^\star$ be an admissible parameter such that the second term in (53) is minimized. Then Theorem C.2 **(ii)** shows that the LPGD algorithm (44) converges linearly to such $\mathbf{Z}^\star$ up to a 'model misspecification error', the minimum value of the second term in (53). In the proof of statistical estimation guarantees of SMF stated in Theorems 3.5 and 4.1, we show that such a model misspecification error is small with high probability.

The general framework of proof in Theorem C.2 shares similarities with the standard argument used to demonstrate the exponential convergence of projected gradient descent with a fixed step size for constrained strongly convex problems (as shown in Theorem 10.29 in [7]). However, a key technical challenge arises due to the absence of non-expansiveness (i.e., 1-Lipschitzness) in our case. This challenge stems from the fact that the constraint set of low-rank matrices is not convex when minimizing a strongly convex objective with a rank-constrained matrix parameter. Consequently, we cannot rely on the non-expansiveness of the convex projection operator, especially considering that the rank-$r$ projection $\Pi_r$ obtained via truncated SVD is not guaranteed to be non-expansive.

To address this issue, we employ a strategy that involves comparing the iterates $\mathbf{Z}_t$ obtained from (44) with auxiliary iterates $\hat{\mathbf{Z}}_t$. These auxiliary iterates are derived using a carefully designed linear projection (see Lemma C.3) that incorporates non-expansiveness. Then we can establish that the original rank-$r$ projection is essentially 2-Lipschitz. So if the distance between the auxiliary iterate $\hat{\mathbf{Z}}_t$ and the global minimizer contracts with a ratio $< 1/2$, then the distance between the actual iterate $\mathbf{Z}_t$ and the global minimizer contracts with a ratio $< 1$. This contraction property ensures exponential convergence of the distance between $\mathbf{Z}_t$ and the global minimizer in Theorem C.2.

***Proof of Theorem** C.2*. We first derive **(i)** assuming **(ii)**. Suppose $\mathbf{Z}^\star = [\mathbf{X}^\star, \mathbf{\Gamma}^\star]$ is a stationary point of $f$ over $\mathbf{\Theta}$ such that $\text{rank}(\mathbf{X}^\star) \leq r$. Let $\mathbf{Z} = [\mathbf{X}, \mathbf{\Gamma}]$ be arbitrary in $\mathbf{\Theta}$ with $\text{rank}(\mathbf{X}) \leq r$. By stationarity of $\mathbf{Z}^\star$, we have $\langle \nabla f(\mathbf{Z}^\star), \bar{\mathbf{Z}} - \mathbf{Z}^\star \rangle \geq 0$, so by RSC (45),

$$\frac{\mu}{2} \|\text{vec}(\mathbf{Z}) - \text{vec}(\mathbf{Z}^\star)\|^2 \leq f(\mathbf{Z}) - f(\mathbf{Z}^\star). \tag{54}$$

Hence $f(\mathbf{Z}) \geq f(\mathbf{Z}^\star)$. Thus $\mathbf{Z}^\star$ is the unique global minimizer of (42). Also, since $\mathbf{Z}^\star$ is a stationary point of $f$ over $\mathbf{\Theta}$, the gradient mapping $\frac{1}{\tau}(\mathbf{Z}^\star - \Pi_{\mathbf{\Theta}}(\mathbf{Z}^\star - \tau \nabla f(\mathbf{Z}^\star)))$ is zero. Thus the rest of **(i)** follows from **(ii)**.

Next, we prove **(ii)**. Let $\mathbf{Z}^\star = [\mathbf{X}^\star, \mathbf{\gamma}^\star] \in \mathbf{\Theta}$ be arbitrary with $\text{rank}(\mathbf{X}^\star) \leq r$. Fix an iteration counter $t \geq 1$. Our proof consists of several steps.

**Step 1: Constructing a suitable linear projection**

Let $\mathbf{X}^\star = \mathbf{U}^\star \mathbf{\Sigma}^\star (\mathbf{V}^\star)^T$ denote the SVD of $\mathbf{X}^\star$. For each iteration $t$, denote $\mathbf{Z}_t = [\mathbf{X}_t, \mathbf{\gamma}_t]$ and let $\mathbf{X}_t = \mathbf{U}_t \mathbf{\Sigma}_t \mathbf{V}_t^T$ denote the SVD of $\mathbf{X}_t$. Since $\mathbf{X}_t$ and $\mathbf{X}^\star$ have rank at most $r$, all of both $\mathbf{U}^\star$, $\mathbf{U}_t$, $\mathbf{V}^\star$, and $\mathbf{V}_t$ have at most $r$ columns. Define a matrix $\mathbf{U}_{3r}$ so that its columns form an orthonormal basis for the subspace spanned by the columns of $[\mathbf{U}^\star, \mathbf{U}_{t-1}, \mathbf{U}_t]$. Then $\mathbf{U}_{3r}$ has at most $3r$ columns. Similarly, let $\mathbf{V}_{3r}$ be a matrix so that its columns form an orthonormal basis for the subspace spanned by the columns of $[\mathbf{V}^\star, \mathbf{V}_{t-1}, \mathbf{V}_t]$. Then $\mathbf{V}_{3r}$ has at most $3r$ columns. Now, define the subspace

$$\mathcal{A} := \left\{ \Delta \in \mathbb{R}^{d_1 \times d_2} \mid \text{span}(\Delta^T) \subseteq \text{span}(\mathbf{V}_{3r}), \, \text{span}(\Delta) \subseteq \text{span}(\mathbf{U}_{3r}) \right\}. \tag{55}$$

Note that $\mathcal{A}$ is a convex subset of $\mathbb{R}^{d_1 \times d_2}$. Also note that, by definition, $\mathbf{X}^\star, \mathbf{X}_t, \mathbf{X}_{t-1} \in \mathcal{A}$. Let $\Pi_{\mathcal{A}}$ denote the linear projection operator of $\mathbb{R}^{d_1 \times d_2}$ onto $\mathcal{A}$.

**Step 2: Constructing auxiliary iterates $\hat{\mathbf{Z}}_t$**

Let $\mathcal{A}$ denote the linear subspace of $\mathbb{R}^{d_1 \times d_2}$ in (55). Let

$$\Pi' := \Pi_{\mathcal{A} \times \mathbb{R}^{d_3 \times d_4}} \tag{56}$$

denote the projection operator of $\mathbb{R}^{d_1 \times d_2} \times \mathbb{R}^{d_3 \times d_4}$ onto $\mathcal{A} \times \mathbb{R}^{d_3 \times d_4}$. Define the following auxiliary iterates

$$\hat{\mathbf{Z}}_t = [\hat{\mathbf{X}}_t, \mathbf{\Gamma}_t] := \Pi' \left( \Pi_{\mathbf{\Theta}} \left( \mathbf{Z}_{t-1} - \tau \nabla f(\mathbf{Z}_{t-1}) \right) \right). \tag{57}$$

By Lemma C.3 and the choice of $\mathcal{A}$, we have

$$\mathbf{X}_t = \Pi_r(\hat{\mathbf{X}}_t) \in \underset{\mathbf{X}, \mathrm{rank}(\mathbf{X}) \leq r}{\arg\min} \|\hat{\mathbf{X}}_t - \mathbf{X}\|_F \quad \text{and} \quad \mathbf{Z}_t, \mathbf{Z}_{t-1}, \mathbf{Z}^\star \in \mathcal{A} \times \mathbb{R}^{d_3 \times d_4}. \tag{58}$$

It follows that

$$\|\mathbf{Z}_t - \mathbf{Z}^\star\|_F \leq \|\mathbf{Z}_t - \hat{\mathbf{Z}}_t\|_F + \|\hat{\mathbf{Z}}_t - \mathbf{Z}^\star\|_F \tag{59}$$

$$= \|\mathbf{X}_t - \hat{\mathbf{X}}_t\|_F + \|\hat{\mathbf{Z}}_t - \mathbf{Z}^\star\|_F \tag{60}$$

$$\leq \|\mathbf{X}^\star - \hat{\mathbf{X}}_t\|_F + \|\hat{\mathbf{Z}}_t - \mathbf{Z}^\star\|_F \leq 2\|\hat{\mathbf{Z}}_t - \mathbf{Z}^\star\|_F. \tag{61}$$

Hence if we can show $\|\hat{\mathbf{Z}}_t - \mathbf{Z}^\star\|_F$ is small, then $\|\mathbf{Z}_t - \mathbf{Z}^\star\|_F$ is also small.

**Step 3. Showing $\|\hat{\mathbf{Z}}_t - \mathbf{Z}^\star\|_F$ is small**

Denote the gradient mapping $\Delta \mathbf{Z}^\star := \frac{1}{\tau}(\mathbf{Z}^\star - \Pi_{\mathbf{\Theta}}(\mathbf{Z}^\star - \tau \nabla f(\mathbf{Z}^\star)))$ (Recall that this equals zero if $\mathbf{Z}^\star$ is a stationary point of $f$ over $\mathbf{\Theta}$, but we do not make such an assumption in this proof). We claim that

$$\|\hat{\mathbf{Z}}_t - \mathbf{Z}^\star\|_F \leq \eta \|\mathbf{Z}_{t-1} - \mathbf{Z}^\star\|_F + \|\Pi'(\tau \Delta \mathbf{Z}^\star)\|_F, \tag{62}$$

where $\eta := \max(|1 - \tau L|, |1 - \tau \mu|)$.

Below we show (62). Using $\mathbf{Z}^\star \in \mathcal{A} \times \mathbb{R}^{d_3 \times d_4}$ and linearity of the linear projection $\Pi'$, write

$$\mathbf{Z}^\star = \Pi'(\mathbf{Z}^\star) \tag{63}$$

$$= \Pi' \left( \Pi_{\mathbf{\Theta}}(\mathbf{Z}^\star - \tau \nabla f(\mathbf{Z}^\star)) \right) + \Pi' \left( \mathbf{Z}^\star - \Pi_{\mathbf{\Theta}}(\mathbf{Z}^\star - \tau \nabla f(\mathbf{Z}^\star)) \right) \tag{64}$$

$$= \Pi' \left( \Pi_{\mathbf{\Theta}}(\mathbf{Z}^\star - \tau \nabla f(\mathbf{Z}^\star)) \right) + \Pi'(\tau \Delta \mathbf{Z}^\star). \tag{65}$$

Using the non-expansiveness of $\Pi'$ and $\Pi_{\mathbf{\Theta}}$ and linearity $\Pi'$,

$$\|\hat{\mathbf{Z}}_t - \mathbf{Z}^\star\|_F \tag{66}$$

$$= \|\Pi' \left( \Pi_{\mathbf{\Theta}} \left( \mathbf{Z}_{t-1} - \tau \nabla f(\mathbf{Z}_{t-1}) \right) \right) - \Pi' \left( \Pi_{\mathbf{\Theta}} \left( \mathbf{Z}^\star - \tau \nabla f(\mathbf{Z}^\star) \right) \right) - \Pi'(\tau \Delta \mathbf{Z}^\star)\|_F \tag{67}$$

$$\leq \|\mathbf{Z}_{t-1} - \tau \nabla f(\mathbf{Z}_{t-1}) - \mathbf{Z}^\star + \tau \nabla f(\mathbf{Z}^\star)\|_F + \|\Pi'(\tau \Delta \mathbf{Z}^\star)\|_F. \tag{68}$$

Hence in order to derive (62), it is enough to show that

$$\|\mathbf{Z}_{t-1} - \tau \nabla f(\mathbf{Z}_{t-1}) - \mathbf{Z}^\star + \tau \nabla f(\mathbf{Z}^\star)\|_F \leq \eta \|\mathbf{Z}_{t-1} - \mathbf{Z}^\star\|_F. \tag{69}$$

The above follows from the fact that $\mathbf{Z}_{t-1}$ and $\mathbf{Z}^\star$ have rank $\leq r$ and the restricted strong convexity and smoothness properties (Definition C.1). Indeed, since $\nabla^2 f$ is continuous,

$$\mathbf{Z}_{t-1} - \tau \nabla f(\mathbf{Z}_{t-1}) - \mathbf{Z}^\star + \tau \nabla f(\mathbf{Z}^\star) \tag{70}$$

$$= (\mathbf{Z}_{t-1} - \mathbf{Z}^\star) - \tau (\nabla f(\mathbf{Z}_{t-1}) - \nabla f(\mathbf{Z}^\star)) \tag{71}$$

$$= \int_0^1 \left( \mathbf{I} - \tau \nabla^2 f(\mathbf{Z}^\star + s(\mathbf{Z}_{t-1} - \mathbf{Z}^\star)) \right) (\mathbf{Z}_{t-1} - \mathbf{Z}^\star) \, ds. \tag{72}$$

From the above with the inequality $\|\mathbf{A}\mathbf{B}\|_F \leq \|\mathbf{A}\|_2 \|\mathbf{B}\|_F$,

$$\|\mathbf{Z}_{t-1} - \tau \nabla f(\mathbf{Z}_{t-1}) - \mathbf{Z}^\star + \tau \nabla f(\mathbf{Z}^\star)\|_F \tag{73}$$

$$\leq \sup_{\tilde{\mathbf{Z}} = [\mathbf{Z}_1, \mathbf{Z}_2]: \mathrm{rank}(\mathbf{Z}_1) \leq r} \|\mathbf{I} - \tau \nabla^2 f(\tilde{\mathbf{Z}})\|_2 \|\mathbf{Z}_{t-1} - \mathbf{Z}^\star\|_F. \tag{74}$$

Since the eigenvalues of $\nabla^2 f(\mathbf{Z}_{t-1})$ are contained in $[\mu, L]$, the eigenvalues of $\mathbf{I} - \tau \nabla^2 f(\mathbf{Z}_{t-1})$ are between $1 - \tau L$ and $1 - \tau \mu$. Hence the right hand side above is at most

$$\eta \|\mathbf{Z}_{t-1} - \mathbf{Z}^\star\|_F, \tag{75}$$

verifying (69). This shows (62).

**Step 4: Bounding the error term**

From (61) and (62), we deduce

$$\|\mathbf{Z}_t - \mathbf{Z}^\star\|_F \leq 2\eta \|\mathbf{Z}_{t-1} - \mathbf{Z}^\star\|_F + 2\|\Pi'(\tau\Delta\mathbf{Z}^\star)\|_F. \tag{76}$$

Note that $\Pi'(\Delta\mathbf{X}^\star, \Delta\boldsymbol{\gamma}^\star) = [\Pi_{\mathcal{A}}(\Delta\mathbf{X}^\star), \Delta\boldsymbol{\gamma}^\star]$ and $\text{rank}(\mathcal{A}) \leq 3r$. Thus by triangle inequality,

$$\|\Pi'(\Delta\mathbf{X}^\star, \Delta\boldsymbol{\gamma}^\star)\|_F \leq \|\Pi'(\Delta\mathbf{X}^\star)\|_F + \|\Delta\boldsymbol{\gamma}^\star\|_F \tag{77}$$

$$\leq \sqrt{3r}\|\Delta\mathbf{X}^\star\|_2 + \|\Delta\boldsymbol{\gamma}^\star\|_F. \tag{78}$$

Also note that $0 \leq \eta < 1/2$ if and only if $\tau \in (\frac{1}{2\mu}, \frac{3}{2L})$, and this interval is non-empty if and only if $L/\mu < 3$. Hence for such choice of $\tau$, $0 < 2\eta < 1$, so by a recursive application of the above inequality, we obtain

$$\|\mathbf{Z}_t - \mathbf{Z}^\star\|_F \leq (2\eta)^t \|\mathbf{Z}_0 - \mathbf{Z}^\star\|_F + \frac{2\tau}{1 - 2\eta}\left(\sqrt{3r}\|\Delta\mathbf{X}^\star\|_2 + \|\Delta\boldsymbol{\gamma}^\star\|_F\right). \tag{79}$$

This completes the proof of **(ii)**. $\qquad\square$

The following lemma is inspired by the proof of Thm. 5.9 in [56].

**Lemma C.3.** *(Linear projection factoring through rank-$r$ projection) Fix $\mathbf{Y} \in \mathbb{R}^{d_1 \times d_2}$, $R \geq r \in \mathbb{N}$, and denote $\mathbf{X} = \Pi_r(\mathbf{Y})$ and $\hat{\mathbf{X}} = \Pi_{\mathcal{A}}(\mathbf{Y})$, where $\mathcal{A} \subseteq \mathbb{R}^{d_1 \times d_2}$ is a linear subspace. Let $\mathbf{X} = \mathbf{U}\boldsymbol{\Sigma}\mathbf{V}^T$ denote the SVD of $\mathbf{X}$. Suppose there exists orthonormal matrices $\overline{\mathbf{U}} \in \mathbb{R}^{d_1 \times R}$ and $\overline{\mathbf{V}} \in \mathbb{R}^{d_2 \times R}$ such that*

$$\mathcal{A} = \left\{\mathbf{A} \in \mathbb{R}^{d_1 \times d_2} \,\middle|\, \text{col}(\mathbf{A}^T) \subseteq \text{col}(\overline{\mathbf{V}}), \text{col}(\mathbf{A}) \subseteq \text{col}(\overline{\mathbf{U}})\right\}, \tag{80}$$

$$\text{col}(\mathbf{U}) \subseteq \text{col}(\overline{\mathbf{U}}), \quad \text{col}(\mathbf{V}) \subseteq \text{col}(\overline{\mathbf{V}}). \tag{81}$$

*Then $\mathbf{X} = \Pi_r(\hat{\mathbf{X}})$.*

*Proof.* Write $\mathbf{Y} - \mathbf{X} = \dot{\mathbf{U}}\dot{\boldsymbol{\Sigma}}\dot{\mathbf{V}}^T$ for its SVD. Let $d := \text{rank}(\mathbf{Y})$ and let $\sigma_1 \geq \cdots \geq \sigma_d > 0$ denote the nonzero singular values of $\mathbf{Y}$. Since $\mathbf{X} = \Pi_r(\mathbf{Y}) = \mathbf{U}\boldsymbol{\Sigma}\mathbf{V}^T$ and $\mathbf{Y} = \mathbf{U}\boldsymbol{\Sigma}\mathbf{V}^T + \dot{\mathbf{U}}\dot{\boldsymbol{\Sigma}}\dot{\mathbf{V}}^T$, we must have that $\boldsymbol{\Sigma}$ consists of the top $r$ singular values of $\mathbf{Y}$ and the rest of $d - r$ singular values are contained in $\dot{\boldsymbol{\Sigma}}$. Furthermore, $\text{col}(\mathbf{U}) \perp \text{col}(\dot{\mathbf{U}})$.

Now, since $\mathbf{X} \in \mathcal{A}$ and $\Pi_{\mathcal{A}}$ is linear, we get

$$\hat{\mathbf{X}} = \Pi_{\mathcal{A}}(\mathbf{X} + (\mathbf{Y} - \mathbf{X})) = \mathbf{U}\boldsymbol{\Sigma}\mathbf{V}^T + \Pi_{\mathcal{A}}(\dot{\mathbf{U}}\dot{\boldsymbol{\Sigma}}\dot{\mathbf{V}}^T). \tag{82}$$

Let $\mathbf{Z} := \Pi_{\mathcal{A}}(\dot{\mathbf{U}}\dot{\boldsymbol{\Sigma}}\dot{\mathbf{V}}^T)$ and write its SVD as $\mathbf{Z} = \widetilde{\mathbf{U}}\widetilde{\boldsymbol{\Sigma}}\widetilde{\mathbf{V}}^T$. Then note that $(\mathbf{U}^T\overline{\mathbf{U}}\,\overline{\mathbf{U}}^T)^T = \overline{\mathbf{U}}\,\overline{\mathbf{U}}^T\mathbf{U} = \mathbf{U}$ since $\overline{\mathbf{U}}\,\overline{\mathbf{U}}^T : \mathbb{R}^{d_1} \to \mathbb{R}^{d_1}$ is the orthogonal projection onto $\text{col}(\overline{\mathbf{U}}) \supseteq \text{col}(\mathbf{U})$. Hence $\mathbf{U}^T\overline{\mathbf{U}}\,\overline{\mathbf{U}}^T = \mathbf{U}^T$. Also note that, by the definition of $\mathcal{A}$, for each $\mathbf{B} \in \mathbb{R}^{d_1 \times d_2}$,

$$\Pi_{\mathcal{A}}(\mathbf{B}) = \overline{\mathbf{U}\mathbf{U}}^T\mathbf{B}\overline{\mathbf{V}}^T\overline{\mathbf{V}}. \tag{83}$$

Hence, noting that $\text{col}(\mathbf{U}) \perp \text{col}(\dot{\mathbf{U}})$, we get

$$\mathbf{U}^T\mathbf{Z} = \mathbf{U}^T\left(\overline{\mathbf{U}}\,\overline{\mathbf{U}}^T\dot{\mathbf{U}}\dot{\boldsymbol{\Sigma}}\dot{\mathbf{V}}^T\overline{\mathbf{V}}^T\overline{\mathbf{V}}\right) \tag{84}$$

$$= \left(\mathbf{U}^T\overline{\mathbf{U}}\,\overline{\mathbf{U}}^T\right)\dot{\mathbf{U}}\dot{\boldsymbol{\Sigma}}\dot{\mathbf{V}}^T\overline{\mathbf{V}}^T\overline{\mathbf{V}} \tag{85}$$

$$= \left(\mathbf{U}^T\dot{\mathbf{U}}\right)\dot{\boldsymbol{\Sigma}}\dot{\mathbf{V}}^T\overline{\mathbf{V}}^T\overline{\mathbf{V}} = O. \tag{86}$$

It follows that $\mathbf{U}^T\widetilde{\mathbf{U}} = O$, since $\mathbf{U}^T\widetilde{\mathbf{U}} = (\mathbf{U}^T\mathbf{Z})\widetilde{\mathbf{V}}(\widetilde{\boldsymbol{\Sigma}})^{-1} = O$. Therefore, rewriting (82) gives the SVD of $\hat{\mathbf{X}}$ as

$$\hat{\mathbf{X}} = \begin{bmatrix} \mathbf{U} & \widetilde{\mathbf{U}} \end{bmatrix} \begin{bmatrix} \boldsymbol{\Sigma} & O \\ O & \widetilde{\boldsymbol{\Sigma}} \end{bmatrix} \begin{bmatrix} \mathbf{V} \\ \widetilde{\mathbf{V}} \end{bmatrix}. \tag{87}$$

Furthermore, $\|\Pi_{\mathcal{A}}(\dot{\mathbf{U}}\dot{\boldsymbol{\Sigma}}\dot{\mathbf{V}}^T)\|_2 \leq \|\dot{\boldsymbol{\Sigma}}\|_2 = \sigma_{r+1}$, so $\boldsymbol{\Sigma}$ consists of the top $r$ singular values of $\hat{\mathbf{X}}$. It follows that $\mathbf{X} = \mathbf{U}\boldsymbol{\Sigma}\mathbf{V}^T$ is the best rank-$r$ approximation of $\hat{\mathbf{X}}$, as desired. $\qquad\square$

**Remark C.4.** Note that in (77), we could have used the following crude bound

$$\|\Pi'\left(\Delta\mathbf{X}^\star, \Delta\boldsymbol{\gamma}^\star\right)\|_F \leq \|[\Delta\mathbf{X}^\star, \Delta\boldsymbol{\gamma}^\star]\|_F \leq \|\Delta\mathbf{X}^\star\|_F + \|\Delta\boldsymbol{\gamma}^\star\|_F \tag{88}$$

$$\leq \sqrt{\operatorname{rank}(\Delta\mathbf{X}^\star)}\|\Delta\mathbf{X}^\star\|_2 + \|\Delta\boldsymbol{\gamma}^\star\|_F, \tag{89}$$

which is also the bound we would have obtained if we chose the trivial linear subspace $\mathcal{A} = \mathbb{R}^{d_1 \times d_2}$ in the proof of Theorem C.2 above. While we know $\operatorname{rank}(\mathbf{X}^\star) \leq r$, we do not have an a priori bound on $\operatorname{rank}(\Delta\mathbf{X}^\star)$, which could be much larger than $\sqrt{3r}$. A smarter choice of the subspace $\mathcal{A}$ as we used in the proof of Theorem C.2 ensures that we only need the factor $\sqrt{3r}$ in place of the unknown factor $\sqrt{\operatorname{rank}(\Delta\mathbf{X}^\star)}$ as in (77).

**Remark C.5.** Suppose $f$ is not only rank-restricted smooth, but also $L'$-smooth on $\boldsymbol{\Theta}$ for some $L' > 0$. Then we have

$$f\left(\mathbf{Z}_t\right) - f(\mathbf{Z}^\star) \leq \left(\|\nabla f(\mathbf{Z}^\star)\| + L\rho^t\right)\rho^t\|\mathbf{Z}_0 - \mathbf{Z}^\star\|_F \tag{90}$$

for $t \geq 1$. Indeed, note that

$$|f(\mathbf{Z}_n) - f(\mathbf{Z}^\star)| = \left|\int_0^1 \langle\nabla f\left(\mathbf{Z}^\star + s(\mathbf{Z}_n - \mathbf{Z}^\star)\right), \mathbf{Z}_n - \mathbf{Z}^\star\rangle \, ds\right| \tag{91}$$

$$\leq \int_0^1 \|\nabla f\left(\mathbf{Z}^\star + s(\mathbf{Z}_n - \mathbf{Z}^\star)\right)\| \|\mathbf{Z}_n - \mathbf{Z}^\star\| \, ds \tag{92}$$

$$\leq \int_0^1 \left(\|\nabla f(\mathbf{Z}^\star)\| + sL'\|\mathbf{Z}_n - \mathbf{Z}^\star\|\right)\|\mathbf{Z}_n - \mathbf{Z}^\star\| \, ds \tag{93}$$

$$\leq \left(\|\nabla f(\mathbf{Z}^\star)\| + L'\|\mathbf{Z}_n - \mathbf{Z}^\star\|\right)\|\mathbf{Z}_n - \mathbf{Z}^\star\|. \tag{94}$$

Then (90) follows from Theorem C.2 **(ii)**.

**Remark C.6.** A similar approach as in our proof of Theorem C.2 was used in [56] for analyzing a similar problem without auxiliary features and under a stronger assumption that the gradient $\nabla f(\mathbf{Z}^\star)$ is small. Our analysis is for a more general setting but is a bit simpler and gives a weaker requirement $L/\mu < 3$ for the well-conditioning of the objective $f$ instead of $L/\mu < 4/3$ in [56].

**Remark C.7.** We give some salient remarks on the use of projections $\Pi_{\boldsymbol{\Theta}}$ and $\Pi_r$ in our LPGD algorithm (44). First, in principle, one could alternate between the two projections $\Pi_r$ and $\Pi_{\boldsymbol{\Theta}}$ at every iteration after a gradient descent step until convergence, similarly to the alternating projection in [12]. However, this would make each iteration of the algorithm prohibitively expensive as this requries to perform rank-$r$ SVD until convergence at *every iteration*. The problem in [12] is much simpler than ours as the objective function is simply the Frobenius norm between the target and the estimated low-rank constrained matrix.

Second, what happens if we switch the order of two projections $\Pi_r$ and $\Pi_{\boldsymbol{\Theta}}$ in (44)? Our proposed LPGD algorithm performs the convex projection $\Pi_{\boldsymbol{\Theta}}$ first and then applies the low-rank projection $\Pi_r$. The key inequality we derive in the proof of Thm. C.2 is (62):

$$\|\mathbf{Z}_t - \mathbf{Z}^\star\|_F \leq 2\eta\|\mathbf{Z}_{t-1} - \mathbf{Z}^\star\|_F + \|\Pi_t\left(\tau\Delta\mathbf{Z}^\star\right)\|_F, \tag{95}$$

where $\Pi_t$ is a linear projection onto a $3r$-dimensional linear subspace that depends on $\mathbf{Z}^\star$, $\mathbf{Z}_t$, and $\mathbf{Z}_{t-1}$. The last error term above can be bounded above uniformly in $t$ using $\|\Pi_t(\mathbf{A})\|_F \leq \sqrt{3r}\|\mathbf{A}\|_2$. So we can apply the above inequality recursively to obtain the desired result.

Now if we consider an alternative algorithm that uses the low-rank projections $\Pi_r$ first and then the convex projection $\Pi_{\boldsymbol{\Theta}}$, then we can derive a corresponding key inequality:

$$\|\mathbf{Z}_t - \mathbf{Z}^\star\|_F \leq 2\eta\|\mathbf{Z}_{t-1} - \mathbf{Z}^\star\|_F + \|\tau\Delta^t\mathbf{Z}^\star\|_F, \tag{96}$$

where $\tau\Delta^t\mathbf{Z}^\star := \mathbf{Z}^\star - \Pi_t(\mathbf{Z}^\star - \tau\nabla f(\mathbf{Z}^\star))$ denotes the gradient mapping at $\mathbf{Z}^\star$ w.r.t. the 'virtual' linear constraint that we constructed during the proof to approximate the low-rank constraint.

To give more detail, it amounts to derive similar inequalities in (65)-(69) assuming the reverse order of projections. Namely, in place of (65), we use

$$\mathbf{Z}^\star = \Pi_{\boldsymbol{\Theta}}(\mathbf{Z}^\star) \tag{97}$$

$$= \Pi_{\boldsymbol{\Theta}}\left(\Pi'(\mathbf{Z}^\star - \tau\nabla f(\mathbf{Z}^\star)) + \mathbf{Z}^\star - \Pi'(\mathbf{Z}^\star - \tau\nabla f(\mathbf{Z}^\star))\right). \tag{98}$$

Note that unlike in (65) we cannot distribute the convex projection $\Pi_\Theta$ since it is not in general a linear projection as $\Pi_t$ is. Then in place of (66), using the non-expansiveness of $\Pi_t$ and $\Pi_\Theta$, we get

$$\|\hat{\mathbf{Z}}_t - \mathbf{Z}^\star\|_F \tag{99}$$

$$= \left\| \begin{array}{c} \Pi_\Theta \left( \Pi' \left( \mathbf{Z}_{t-1} - \tau \nabla f(\mathbf{Z}_{t-1}) \right) \right) \\ -\Pi_\Theta \left( \Pi'(\mathbf{Z}^\star - \tau \nabla f(\mathbf{Z}^\star)) + \mathbf{Z}^\star - \Pi'(\mathbf{Z}^\star - \tau \nabla f(\mathbf{Z}^\star)) \right) \end{array} \right\|_F \tag{100}$$

$$\leq \left\| \begin{array}{c} \left( \Pi' \left( \mathbf{Z}_{t-1} - \tau \nabla f(\mathbf{Z}_{t-1}) \right) \right) \\ - \left( \Pi'(\mathbf{Z}^\star - \tau \nabla f(\mathbf{Z}^\star)) + \mathbf{Z}^\star - \Pi'(\mathbf{Z}^\star - \tau \nabla f(\mathbf{Z}^\star)) \right) \end{array} \right\|_F \tag{101}$$

$$\leq \|\mathbf{Z}_{t-1} - \tau \nabla f(\mathbf{Z}_{t-1}) - \mathbf{Z}^\star + \tau \nabla f(\mathbf{Z}^\star)\|_F + \|\tau \Delta^t \mathbf{Z}^\star\|_F. \tag{102}$$

The first term in the last expression can be bounded by the same argument as in (69)-(75). Then by recursively applying the inequality (96), we can obtain

$$\|\mathbf{Z}_t - \mathbf{Z}^\star\|_F \leq (2\eta)^t \|\mathbf{Z}_0 - \mathbf{Z}^\star\|_F + \sum_{k=1}^{t} (2\eta)^{t-k} \|\tau \Delta^k \mathbf{Z}^\star\|_F. \tag{103}$$

Hence the rate of convergence we would get is the same as the original algorithm, but the additive error takes a different form. Since the 'low-rank gradient mapping' $\Delta^k \mathbf{Z}^\star$ depends on the iterates $\mathbf{Z}_k, \mathbf{Z}_{k-1}$, we find it easier to control the gradient mapping with respect to the convex projection that comes out from the analysis of the original algorithm.

# D  Proof of Theorems 3.5 and 4.1

In this section, we prove the main results for SMF, Theorems 3.5 and 4.1. In the main text, we explained that our algorithm for SMF (Alg. 1) is exactly an LPGD for the reformulated problems (11) (for SMF-$\mathbf{H}$) and (12) (for SMF-$\mathbf{W}$). Therefore, our proofs of Theorems 3.5 and 4.1 are essentially verifying the well-conditioning hypothesis $L/\mu < 3$ of the general result for the LPGD algorithm (Theorem C.2).

## D.1  Proof of Theorem 3.5 and its generalization

Theorem 3.5 in the main text is a special case of the following more general result, which we prove in this section.

**Theorem D.1.** *(Exponential convergence for SMF) Let $\mathbf{Z}_t := [\boldsymbol{\theta}_t, \boldsymbol{\gamma}_t]$ denote the iterates of Algorithm 1. Assume 3.1-3.3 hold. Let $\mu$ and $L$ be as in (15), fix stepsize $\tau \in (\frac{1}{2\mu}, \frac{3}{2L})$, and let $\rho := 2(1-\tau\mu) \in (0,1)$. Suppose $L/\mu < 3$.*

**(i)** *(Correctly specified case; Theorem 3.5 in the main text) Suppose there exists a stationary point $\mathbf{Z}^* = [\boldsymbol{\theta}^*, \boldsymbol{\gamma}^*]$ of $F$ over the convex constraint set $\boldsymbol{\Theta}$ s.t. $\mathrm{rank}(\boldsymbol{\theta}^*) \leq r$. Then $\mathbf{Z}^*$ is the unique global minimizer of $F$ among all $\mathbf{Z} = [\boldsymbol{\theta}, \boldsymbol{\gamma}]$ with $\mathrm{rank}(\boldsymbol{\theta}) \leq r$. Moreover,*

$$\|\mathbf{Z}_t - \mathbf{Z}^*\|_F \leq \rho^t \|\mathbf{Z}_0 - \mathbf{Z}^*\|_F \qquad \text{for } t \geq 1. \tag{104}$$

**(ii)** *(Possibly misspecified case) Let $\mathbf{Z}^\star = [\boldsymbol{\theta}^\star, \boldsymbol{\gamma}^\star]$ be arbitrary in $\boldsymbol{\Theta}$ s.t. $\mathrm{rank}(\boldsymbol{\theta}^\star) \leq r$. Denote the gradient mapping at $\mathbf{Z}^\star$ as $[\Delta\boldsymbol{\theta}^\star, \Delta\boldsymbol{\Gamma}^\star] := \frac{1}{\tau} (\mathbf{Z}^\star - \Pi_\Theta(\mathbf{Z}^\star - \tau \nabla F(\mathbf{Z}^\star))$. Then for $t \geq 1$,*

$$\|\mathbf{Z}_t - \mathbf{Z}^\star\|_F \leq \rho^t \|\mathbf{Z}_0 - \mathbf{Z}^\star\|_F + \frac{2\tau}{1-\rho} \left( \sqrt{3r} \|\Delta\boldsymbol{\theta}^\star\|_2 + \|\Delta\boldsymbol{\gamma}^\star\|_F \right). \tag{105}$$

We remark that even in the presence of a nonzero additive error (which bounds the unnormalized estimation error $\|\mathbf{Z}_\infty - \mathbf{Z}^\star\|_F$), our Theorem 4.1 demonstrates that, under natural generative models for SML, this error becomes vanishingly small with high probability with noise variance $\sigma^2 = O(1/n)$ for SMF-$\mathbf{W}$ and $\sigma^2 = o(1/\sqrt{n})$ for SMF-$\mathbf{H}$. Roughly speaking, these results indicate that the generative SMF models are nearly correctly specified with high probability. As a result, the algorithm achieves exponential convergence to the correct parameters for the generative SMF model up to a statistical error that vanishes as the sample size tends to infinity.

We begin with some preliminary computations. Let $\mathbf{a}_s$ denote the activation corresponding to the $s$th sample (see (2)). More precisely, $\mathbf{a}_s = \mathbf{A}^T \mathbf{x}_s + \boldsymbol{\gamma}^T \mathbf{x}'_s$ for the filter-based model with $\mathbf{A} \in \mathbb{R}^{p \times \kappa}$,

and $\mathbf{a}_s = \mathbf{A}[:, s] + \boldsymbol{\gamma}^T \mathbf{x}'_s$ with $\mathbf{A} \in \mathbb{R}^{\kappa \times n}$ for the feature-based model. In both cases, $\mathbf{B} \in \mathbb{R}^{p \times n}$ and $\boldsymbol{\gamma} \in \mathbb{R}^{q \times \kappa}$. Then the objective function $f$ in (4) can be written as

$$f(\mathbf{A}, \mathbf{B}, \boldsymbol{\gamma}) := \left( -\sum_{s=1}^{n} \sum_{j=0}^{\kappa} \mathbf{1}(y_s = j) \log g_j(\mathbf{a}_s) \right) + \xi \|\mathbf{X}_{\text{data}} - \mathbf{B}\|_F^2 + \lambda \left( \|\mathbf{A}\|_F^2 + \|\boldsymbol{\gamma}\|_F^2 \right) \tag{106}$$

$$= \sum_{s=1}^{n} \left( \log \left( 1 + \sum_{c=1}^{\kappa} h(\mathbf{a}_s[c]) \right) - \sum_{c=1}^{\kappa} \mathbf{1}(y_s = c) \log h(\mathbf{a}_s[c]) \right) \tag{107}$$

$$+ \xi \|\mathbf{X}_{\text{data}} - \mathbf{B}\|_F^2 + \lambda \left( \|\mathbf{A}\|_F^2 + \|\boldsymbol{\gamma}\|_F^2 \right), \tag{108}$$

where $\mathbf{a}_s[i] \in \mathbb{R}$ denotes the $i$th component of $\mathbf{a}_s \in \mathbb{R}^{\kappa}$. In the proofs we provided below, we compute the Hessian of $f$ above explicitly for the filter- and the feature-based SMF models and use Theorem C.2 to derive the result.

For each label $y \in \{0, \ldots, \kappa\}$ and activation $\mathbf{a} \in \mathbb{R}^{\kappa}$, recall the negative log-likelihood

$$\ell(y, \mathbf{a}) = \log \sum_{c=1}^{\kappa} h(a_c) - \sum_{c=1}^{\kappa} \mathbf{1}_{\{y=c\}} \log h(a_c) \tag{109}$$

of observing label $y$ from the probability distribution $\mathbf{g}(\mathbf{a})$ defined in (1). An easy computation shows

$$\nabla_{\mathbf{a}} \ell(y, \mathbf{a}) = \dot{\mathbf{h}}(y, \mathbf{a}) = (\dot{h}_1, \ldots, \dot{h}_{\kappa}) \in \mathbb{R}^{\kappa}, \quad \nabla_{\mathbf{a}} \nabla_{\mathbf{a}^T} \ell(y, \mathbf{a}) = \ddot{\mathbf{H}}(y, \mathbf{a}) = (\ddot{h}_{ij}) \in \mathbb{R}^{\kappa \times \kappa}, \tag{110}$$

where

$$\dot{h}_j = \dot{h}_j(y, \mathbf{a}) := \left( \frac{h'(a_j)}{1 + \sum_{c=1}^{\kappa} h(a_c)} - \mathbf{1}(y = j) \frac{h'(a_j)}{h(a_j)} \right) \tag{111}$$

$$\ddot{h}_{ij} := \left( \frac{h''(a_j) \mathbf{1}(i = j)}{1 + \sum_{c=1}^{\kappa} h(a_c)} - \frac{h'(a_i) h'(a_j)}{(1 + \sum_{c=1}^{\kappa} h(a_c))^2} \right) - \mathbf{1}(y = i = j) \left( \frac{h''(a_j)}{h(a_j)} - \frac{(h'(a_j))^2}{(h(a_j))^2} \right). \tag{112}$$

***Proof of Theorem D.1** for SMF-**W**.* Let $f = F$ denote the loss function for the filter-based SMF model in (12). Fix $\mathbf{Z}_1, \mathbf{Z}_2 \in \boldsymbol{\Theta} \subseteq \mathbb{R}^{d_1 \times d_2} \times \mathbb{R}^{d_3 \times d_4}$. Since the constraint set $\boldsymbol{\Theta}$ is convex (see Algorithm 1), $t\mathbf{Z}_1 + (1 - t)\mathbf{Z}_2 \in \boldsymbol{\Theta}$ for all $t \in [0, 1]$. Then by the mean value theorem, there exists $t^* \in [0, 1]$ such that for $\mathbf{Z}^* = t^*\mathbf{Z}_1 + (1 - t^*)\mathbf{Z}_2$,

$$f(\mathbf{Z}_2) - f(\mathbf{Z}_1) - \langle \nabla f(\mathbf{Z}_1), \mathbf{Z}_2 - \mathbf{Z}_1 \rangle \tag{113}$$

$$= \frac{1}{2} \left( \text{vec}(\mathbf{Z}_2) - \text{vec}(\mathbf{Z}_1) \right)^T \nabla_{\text{vec}(\mathbf{Z})} \nabla_{\text{vec}(\mathbf{Z})^T} f(\mathbf{Z}^*) \left( \text{vec}(\mathbf{Z}_2) - \text{vec}(\mathbf{Z}_1) \right). \tag{114}$$

Hence, according to Theorem C.2, it suffices to verify that for some $\mu, L > 0$ such that $L/\mu < 3$ and

$$\mu \mathbf{I} \preceq \nabla_{\text{vec}(\mathbf{Z})} \nabla_{\text{vec}(\mathbf{Z})^T} f(\mathbf{Z}^*) \preceq L\mathbf{I} \tag{115}$$

for all $\mathbf{Z}^* = [\mathbf{X}, \boldsymbol{\gamma}]$ with $\text{rank}(\mathbf{X}^*) \le r$.

To this end, let $\mathbf{a}_s = \mathbf{A}^T \mathbf{x}_s + \boldsymbol{\gamma}^T \mathbf{x}'_s$ for the filter-based model we consider here. We discussed that the objective function $f$ in (12) can be written as (106). Denote

$$\mathbf{a}_s = \mathbf{A}^T \mathbf{x}_s + \boldsymbol{\gamma}^T \mathbf{x}'_s =: \left[ \left\langle \underbrace{\begin{bmatrix} \mathbf{A}[:, j] \\ \boldsymbol{\gamma}[:, j] \end{bmatrix}}_{=:\mathbf{u}_j}, \underbrace{\begin{bmatrix} \mathbf{x}_s \\ \mathbf{x}'_s \end{bmatrix}}_{=:\boldsymbol{\phi}_s} \right\rangle; \ j = 1, \ldots, \kappa \right]^T \in \mathbb{R}^{\kappa}, \tag{116}$$

where we have introduced the notations $\mathbf{u}_j \in \mathbb{R}^{(p+q) \times 1}$ for $j = 1, \ldots, \kappa$ and $\boldsymbol{\phi}_s \in \mathbb{R}^{(p+q) \times 1}$ for $s = 1, \ldots, n$. Denote $\mathbf{U} := [\mathbf{u}_1, \ldots, \mathbf{u}_{\kappa}] = [\mathbf{A} \| \boldsymbol{\gamma}] \in \mathbb{R}^{(p+q) \times \kappa}$, which is a matrix parameter that vertically concatenates $\mathbf{A}$ and $\boldsymbol{\gamma}$. Also denote $\boldsymbol{\Phi} = (\boldsymbol{\phi}_1, \ldots, \boldsymbol{\phi}_n) \in \mathbb{R}^{(p+q) \times n}$, which is the feature matrix of $n$ observations. Writing

$$f(\mathbf{U}, \mathbf{B}) = \sum_{s=1}^{n} \ell(y_s, \mathbf{U}^T \boldsymbol{\phi}_s) + \xi \|\mathbf{X}_{\text{data}} - \mathbf{B}\|_F^2 + \lambda \|\mathbf{U}\|_F^2 \tag{117}$$

and using (109), we can compute the gradient and the Hessian of $f$ above as follows:

$$\nabla_{\text{vec}(\mathbf{U})} f(\mathbf{U}, \mathbf{B}) = \left( \sum_{s=1}^{n} \dot{\mathbf{h}}(y_s, \mathbf{U}^T \boldsymbol{\phi}_s) \otimes \boldsymbol{\phi}_s \right) + 2\lambda \, \text{vec}(\mathbf{U}), \tag{118}$$

$$\nabla_{\mathbf{B}} f(\mathbf{U}, \mathbf{B}) = 2\xi (\mathbf{B} - \mathbf{X}_{\text{data}}), \tag{119}$$

$$\nabla_{\text{vec}(\mathbf{U})} \nabla_{\text{vec}(\mathbf{U})^T} f(\mathbf{U}, \mathbf{B}) = \left( \sum_{s=1}^{n} \ddot{\mathbf{H}}(y_s, \mathbf{U}^T \boldsymbol{\phi}_s) \otimes \boldsymbol{\phi}_s \boldsymbol{\phi}_s^T \right) + 2\lambda \mathbf{I}_{(p+q)\kappa}, \tag{120}$$

$$\nabla_{\text{vec}(\mathbf{B})} \nabla_{\text{vec}(\mathbf{B})^T} f(\mathbf{U}, \mathbf{B}) = 2\xi \mathbf{I}_{pn}, \qquad \nabla_{\text{vec}(\mathbf{B})} \nabla_{\text{vec}(\mathbf{U})^T} f(\mathbf{U}, \mathbf{B}) = O, \tag{121}$$

where $\otimes$ above denotes the Kronecker product and the functions $\dot{\mathbf{h}}$ and $\ddot{\mathbf{H}}$ are defined in (110).

Recall that the eigenvalues of $\mathbf{A} \otimes \mathbf{B}$, where $\mathbf{A}$ and $\mathbf{B}$ are two square matrices, are given by $\lambda_i \mu_j$, where $\lambda_i$ and $\mu_j$ run over all eigenvalues of $\mathbf{A}$ and $\mathbf{B}$, respectively. Hence denoting $\mathbf{H}_{\mathbf{U}} := \sum_{s=1}^{n} \ddot{\mathbf{H}}(y_s, \mathbf{U}^T \boldsymbol{\phi}_s,) \otimes \boldsymbol{\phi}_s \boldsymbol{\phi}_s^T$ and using 3.1-3.2, we can deduce

$$\lambda_{\min}(\mathbf{H}_{\mathbf{U}}) \geq n \lambda_{\min} \left( n^{-1} \boldsymbol{\Phi} \boldsymbol{\Phi}^T \right) \min_{1 \leq s \leq n, \, \mathbf{U}} \lambda_{\min} \left( \ddot{\mathbf{H}}(y_s, \boldsymbol{\phi}_s, \mathbf{U}) \right) \geq n \delta^- \alpha^- \geq n \mu^* > 0, \tag{122}$$

$$\lambda_{\max}(\mathbf{H}_{\mathbf{U}}) \leq n \lambda_{\max} \left( n^{-1} \boldsymbol{\Phi} \boldsymbol{\Phi}^T \right) \max_{1 \leq s \leq n, \, \mathbf{U}} \lambda_{\max} \left( \ddot{\mathbf{H}}(y_s, \boldsymbol{\phi}_s, \mathbf{U}) \right) \leq n \delta^+ \alpha^+ \leq n L^*. \tag{123}$$

It follows that the eigenvalues of the Hessian $\mathbf{H}_{\text{filt}}$ of the loss function $f$ satisfy

$$\lambda_{\min}(\mathbf{H}_{\text{filt}}) \geq \min(2\lambda + n\mu^*, \, 2\xi), \tag{124}$$

$$\lambda_{\max}(\mathbf{H}_{\text{filt}}) \leq \max \left( 2\lambda + nL^*, \, 2\xi \right). \tag{125}$$

This holds for all $\mathbf{A}, \mathbf{B}, \boldsymbol{\gamma}$ such that $\text{rank}([\mathbf{A}, \mathbf{B}]) \leq r$ and under the convex constraint (also recall that $\mathbf{U}$ is the vertical stack of $\mathbf{A}$ and $\boldsymbol{\gamma}$). Hence we conclude that the objective function $F$ in (12) verifies RSC and RSM properties (Def. C.1) with parameters $\mu = \min(n\mu^* + 2\lambda, \, 2\xi)$ and $L = \max(nL^* + 2\lambda, \, 2\xi)$. This verifies (115) for the chosen parameters $\mu$ and $L$. Then the rest follows from Theorem C.2. $\qquad\square$

Next, we prove Theorem D.1 for SMF-**H**, the exponential convergence of Algorithm 1 for the feature-based SMF.

***Proof of Theorem D.1 for SMF*-H.** We will use the same setup as in the proof of Theorem D.1 for SMF-**W**. The main part of the argument is the computation of the Hessian of loss function $f := F$ for SMF-**H** in (11), which is straightforward but substantially more involved than the corresponding computation for the filter-based case in the proof of Theorem D.1. Let $\mathbf{a}_s = \mathbf{A}[:, s] + \boldsymbol{\gamma}^T \mathbf{x}'_s$ for the feature-based model with $\mathbf{A} \in \mathbb{R}^{\kappa \times n}$. Denote

$$\mathbf{a}_s = \mathbf{I}_\kappa \mathbf{A}[:, s] + \boldsymbol{\gamma}^T \mathbf{x}'_s =: \left[ \left\langle \underbrace{\begin{bmatrix} \mathbf{I}_\kappa[:, j] \\ \boldsymbol{\gamma}[:, j] \end{bmatrix}}_{=: \mathbf{v}_j}, \underbrace{\begin{bmatrix} \mathbf{A}[:, s] \\ \mathbf{x}'_s \end{bmatrix}}_{=: \boldsymbol{\psi}_s} \right\rangle; \; j = 1, \dots, \kappa \right]^T \in \mathbb{R}^\kappa. \tag{126}$$

Note that in the above representation we have concatenated $\mathbf{A}[:, s]$ with the auxiliary covariate $\mathbf{x}'_s$, whereas previously for SMF-**W** (see (116)), we concatenated $\mathbf{A}[:, j]$ with regression coefficient $\boldsymbol{\gamma}[:, j]$ for the auxiliary covarate for the $j$th class[2]. A straightforward computation shows the following

---

[2]This is because for the feature-based model, the column $\mathbf{A}[:, s] \in \mathbb{R}^\kappa$ for $s = 1, \dots, n$ represent a feature of the $s$th sample, whereas for the filter-based model, $\mathbf{A}[:, j]$ for $j = 1, \dots, \kappa$ represents the $j$th filter that is applied to the feature $\mathbf{x}_s$ of the $s$th sample.

gradient formulas:

$$\nabla_{\mathrm{vec}(\mathbf{A})} f(\mathbf{A},\boldsymbol{\gamma},\mathbf{B}) = \left(\sum_{s=1}^{n} \dot{\mathbf{h}}(y_s,\mathbf{a}_s)\otimes \mathbf{I}_n[:,s]\right) + 2\lambda\,\mathrm{vec}(\mathbf{A}) = \begin{bmatrix} \dot{\mathbf{h}}(y_1,\mathbf{a}_1)\\ \vdots \\ \dot{\mathbf{h}}(y_n,\mathbf{a}_n)\end{bmatrix} + 2\lambda\,\mathrm{vec}(\mathbf{A}),$$

(127)

$$\nabla_{\mathrm{vec}(\boldsymbol{\gamma})} f(\mathbf{A},\boldsymbol{\gamma},\mathbf{B}) = \left(\sum_{s=1}^{n} \dot{\mathbf{h}}(y_s,\mathbf{a}_s)\otimes \mathbf{x}'_s\right) + 2\lambda\,\mathrm{vec}(\boldsymbol{\gamma}),$$

(128)

$$\nabla_{\mathbf{B}} f(\mathbf{A},\boldsymbol{\gamma},\mathbf{B}) = 2\xi(\mathbf{B}-\mathbf{X}_{\mathrm{data}})$$

(129)

$$\nabla_{\mathrm{vec}(\mathbf{A})}\nabla_{\mathrm{vec}(\mathbf{A})^T} f(\mathbf{A},\boldsymbol{\gamma},\mathbf{B}) = \mathrm{diag}\left(\ddot{\mathbf{H}}(y_1,\mathbf{a}_1),\dots,\ddot{\mathbf{H}}(y_n,\mathbf{a}_n)\right) + 2\lambda\mathbf{I}_{\kappa n},$$

(130)

$$\nabla_{\mathrm{vec}(\boldsymbol{\gamma})}\nabla_{\mathrm{vec}(\boldsymbol{\gamma})^T} f(\mathbf{A},\boldsymbol{\gamma},\mathbf{B}) = \left(\sum_{s=1}^{n} \ddot{\mathbf{H}}(y_s,\mathbf{a}_s)\otimes \mathbf{x}'_s(\mathbf{x}'_s)^T\right) + 2\lambda\mathbf{I}_{q\kappa},$$

(131)

$$\nabla_{\mathrm{vec}(\mathbf{B})}\nabla_{\mathrm{vec}(\mathbf{B})^T} f(\mathbf{A},\boldsymbol{\gamma},\mathbf{B}) = 2\xi\mathbf{I}_{pn},$$

(132)

$$\nabla_{\mathrm{vec}(\boldsymbol{\gamma})}\nabla_{\mathrm{vec}(\mathbf{A})^T} f(\mathbf{A},\boldsymbol{\gamma},\mathbf{B}) = \left[\ddot{\mathbf{H}}(y_1,\mathbf{a}_1)\otimes \mathbf{x}'_1,\dots,\ddot{\mathbf{H}}(y_n,\mathbf{a}_n)\otimes \mathbf{x}'_n\right] \in \mathbb{R}^{\kappa q\times\kappa n},$$

(133)

$$\nabla_{\mathrm{vec}(\mathbf{B})}\nabla_{\mathrm{vec}(\mathbf{V})^T} f(\mathbf{A},\boldsymbol{\gamma},\mathbf{B}) = O \qquad \text{for } \mathbf{V}=\mathbf{A},\boldsymbol{\gamma}.$$

(134)

From this, we will compute the eigenvalues of the Hessian $\mathbf{H}_{\mathrm{feat}}$ of the loss function $f$. In order to illustrate our computation in a simple setting, we first assume $\kappa = 1 = q$, which corresponds to binary classification $\kappa = 1$ with one-dimensional auxiliary features $q = 1$. In this case, we have

$$\mathbf{H}_{\mathrm{feat}} := \nabla_{\mathrm{vec}(\mathbf{A},\boldsymbol{\gamma},\mathbf{B})}\nabla_{\mathrm{vec}(\mathbf{A},\boldsymbol{\gamma},\mathbf{B})^T} f(\mathbf{A},\boldsymbol{\gamma},\mathbf{B})$$

(135)

$$= \begin{bmatrix} \ddot{h}(y_1,\mathbf{a}_1)+2\lambda & 0 & \dots & 0 & \ddot{h}(y_1,\mathbf{a}_1)x'_1 & O \\ 0 & \ddot{h}(y_2,\mathbf{a}_2)+2\lambda & \dots & 0 & \ddot{h}(y_2,\mathbf{a}_2)x'_2 & O \\ \vdots & \vdots & \ddots & \vdots & \vdots & \vdots \\ 0 & \dots & 0 & \ddot{h}(y_n,\mathbf{a}_n)+2\lambda & \ddot{h}(y_n,\mathbf{a}_n)x'_n & O \\ \ddot{h}(y_1,\mathbf{a}_1)x'_1 & \ddot{h}(y_2,\mathbf{a}_2)x'_2 & \dots & \ddot{h}(y_n,\mathbf{a}_n)x'_n & \left(\sum_{s=1}^n \ddot{h}(y_s,\mathbf{a}_s)(x'_s)^2\right)+2\lambda & O \\ O & O & \dots & O & O & 2\xi\mathbf{I}_{pn} \end{bmatrix},$$

(136)

where we denoted $\ddot{h} = \ddot{h}_{11} \in \mathbb{R}$ and $x'_s = \mathbf{x}'_s \in \mathbb{R}$ for $s = 1,\dots,n$. In order to compute the eigenvalues of the above matrix, we will use the following formula for the determinant of $3\times 3$ block matrix: ($O$ representing matrices of zero entries with appropriate sizes)

$$\det\left(\begin{bmatrix} A & B & O \\ B^T & C & O \\ O & O & D \end{bmatrix}\right) = \det\left(C - B^T A^{-1} B\right)\det(A)\det(D).$$

(137)

This yields the following simple formula for the characteristic polynomial of $\mathbf{H}_{\mathrm{feat}}$:

$$\det(\mathbf{H}_{\mathrm{feat}} - t\mathbf{I})$$

(138)

$$= \left(\sum_{s=1}^{n}\ddot{h}(y_s,\mathbf{a}_s)(x'_s)^2 - \sum_{s=1}^{n}\frac{(\ddot{h}(y_s,\mathbf{a}_s))^2(x'_s)^2}{\ddot{h}(y_s,\mathbf{a}_s)+2\lambda-t}+2\lambda-t\right)\prod_{s=1}^{n}\left(\ddot{h}(\mathbf{y}_s,\mathbf{a}_s)+2\lambda-t\right)(2\xi-t)^{pn}$$

(139)

$$= \left(\sum_{s=1}^{n}\frac{(2\lambda-t)\ddot{h}(y_s,\mathbf{a}_s)(x'_s)^2}{\ddot{h}(y_s,\mathbf{a}_s)+2\lambda-t}+2\lambda-t\right)\prod_{s=1}^{n}\left(\ddot{h}(\mathbf{y}_s,\mathbf{a}_s)+2\lambda-t\right)(2\xi-t)^{pn}.$$

(140)

Since the first term in the parenthesis in the above equation has solution $2\lambda$, it follows that

$$\lambda_{\min}(\mathbf{H}_{\mathrm{feat}}) \geq \min(2\lambda,\ \alpha^- + 2\lambda,\ 2\xi) = \min(2\lambda,\ 2\xi),$$

(141)

$$\lambda_{\max}(\mathbf{H}_{\mathrm{feat}}) \leq \max\left(2\lambda,\ \alpha^+ + 2\lambda,\ 2\xi\right) = \max\left(\alpha^+ + 2\lambda,\ 2\xi\right).$$

(142)

Now we generalize the above computation for the general case $\kappa, q \geq 1$. First, note the general form of the Hessian as below:

$$\mathbf{H}_{\text{feat}} := \nabla_{\text{vec}(\mathbf{A}, \boldsymbol{\gamma}, \mathbf{B})} \nabla_{\text{vec}(\mathbf{A}, \boldsymbol{\gamma}, \mathbf{B})^T} f(\mathbf{A}, \boldsymbol{\gamma}, \mathbf{B}) \tag{143}$$

$$= \begin{bmatrix} \ddot{\mathbf{H}}(y_1, \mathbf{a}_1) + 2\lambda \mathbf{I}_\kappa & 0 & \dots & 0 & (\ddot{\mathbf{H}}(y_1, \mathbf{a}_1) \otimes \mathbf{x}_1')^T & O \\ 0 & \ddot{\mathbf{H}}(y_2, \mathbf{a}_2) + 2\lambda \mathbf{I}_\kappa & \dots & 0 & (\ddot{\mathbf{H}}(y_2, \mathbf{a}_2) \otimes \mathbf{x}_2')^T & O \\ \vdots & \vdots & \ddots & \vdots & \vdots & \vdots \\ 0 & \dots & 0 & \ddot{\mathbf{H}}(y_n, \mathbf{a}_n) + 2\lambda \mathbf{I}_\kappa & (\ddot{\mathbf{H}}(y_n, \mathbf{a}_n) \otimes \mathbf{x}_n')^T & O \\ \ddot{\mathbf{H}}(y_1, \mathbf{a}_1) \otimes \mathbf{x}_1' & \ddot{\mathbf{H}}(y_2, \mathbf{a}_2) \otimes \mathbf{x}_2' & \dots & \ddot{\mathbf{H}}(y_n, \mathbf{a}_n) \otimes \mathbf{x}_n' & \sum_{s=1}^n \ddot{\mathbf{H}}(y_s, \mathbf{a}_s) \otimes \mathbf{x}_s'(\mathbf{x}_s')^T + 2\lambda \mathbf{I}_{q\kappa} & O \\ O & O & \dots & O & O & 2\xi \mathbf{I}_{pn} \end{bmatrix}. \tag{144}$$

Note that for any square symmetric matrix $B$ and a column vector $\mathbf{x}$ of matching size,

$$B \otimes \mathbf{x}\mathbf{x}^T - (B \otimes \mathbf{x})^T (B + w\mathbf{I})^{-1} (B \otimes \mathbf{x}) = (B - B(B + w\mathbf{I})^{-1}B) \otimes (\mathbf{x}\mathbf{x}^T) \tag{145}$$

$$= ((B + wI) - B)(B + w\mathbf{I})^{-1}B \otimes (\mathbf{x}\mathbf{x}^T) \tag{146}$$

$$= w(B + wI)^{-1}B \otimes \mathbf{x}\mathbf{x}^T. \tag{147}$$

Hence by a similar computation as before, we obtain

$$\det(\mathbf{H}_{\text{feat}} - t\mathbf{I}) \tag{148}$$

$$= \det\left( (2\lambda - t) \sum_{s=1}^n \left( \ddot{\mathbf{H}}(y_s, \mathbf{a}_s) + (2\lambda - t)\mathbf{I}_\kappa \right)^{-1} \ddot{\mathbf{H}}(y_s, \mathbf{a}_s) \otimes \mathbf{x}_s'(\mathbf{x}_s')^T + (2\lambda - t)\mathbf{I}_{q\kappa} \right) \tag{149}$$

$$\times \left( \prod_{s=1}^n \det\left( \ddot{\mathbf{H}}(y_s, \mathbf{a}_s) + (2\lambda - t)\mathbf{I}_\kappa \right) \right) (2\xi - t)^{pn}. \tag{150}$$

It follows that

$$\lambda_{\min}(\mathbf{H}_{\text{feat}}) \geq \min(2\lambda, 2\xi), \tag{151}$$

$$\lambda_{\max}(\mathbf{H}_{\text{feat}}) \leq \max\left( \alpha^+ + 2\lambda, 2\xi \right). \tag{152}$$

Then the rest follows from Theorem C.2. $\qquad \square$

## D.2 Proof of Theorem 4.1

In this section, we prove the statistical estimation guarantee for SMF in Theorem 4.1. Recall the generative model for SMF in (17). Our proof is based on Theorem 3.5 that we have established previously and standard matrix concentration bounds, which we provide below:

**Lemma D.2** (2-norm of matrices with bounded and independent columns). *Let* $\mathbf{X}$ *be a* $d_1 \times d_2$ *random matrix of independent, mean zero, real-valued columns such that* $\|\mathbf{X}\|_\infty < L$ *almost surely for some constant* $L > 0$. *Then*

$$\mathbb{P}\left( \|\mathbf{X}\|_2 \geq t \right) \leq (d_1 + d_2) \exp\left( \frac{-t^2/2}{\max\{d_1, d_2\}L^2 + (Lt/3)} \right). \tag{153}$$

*Proof.* This lemma is a simple consequence of the matrix Burnstein's inequality (see, e.g., [52, Thm. 6.1.1]). Indeed, write $\mathbf{X} = [\mathbf{x}_1, \dots, \mathbf{x}_{d_2}]$, where $\mathbf{x}_j$s are the columns of $\mathbf{X}$. Note that, by the hypothesis,

$$\|\mathbb{E}[\mathbf{X}^T \mathbf{X}]\|_2 = \|\text{diag}(\mathbb{E}[\|\mathbf{x}_1\|_2^2], \dots, \mathbb{E}[\|\mathbf{x}_{d_2}\|_2^2])\|_2 \leq d_1 L^2. \tag{154}$$

Similarly,

$$\|\mathbb{E}[\mathbf{X}\mathbf{X}^T]\|_2 = \left\| \sum_{j=1}^{d_2} \mathbb{E}[\mathbf{x}_j \mathbf{x}_j^T] \right\|_2 \leq d_2 L^2. \tag{155}$$

It follows that matrix variance statistics $v(\mathbf{X})$ of $\mathbf{X}$ satisfies

$$v(\mathbf{X}) := \max\left\{ \|\mathbb{E}[\mathbf{X}\mathbf{X}^T]\|_2, \|\mathbb{E}[\mathbf{X}^T \mathbf{X}]\|_2 \right\} \leq \max\{d_1, d_2\}L^2. \tag{156}$$

Then the tail bound on the 2-norm of $\mathbf{X}$ as asserted follows immediately from the matrix Burnstein's inequality. $\qquad \square$

**Lemma D.3.** *(2-norm of matrices with independent sub-gaussian entries) Let $\mathbf{A}$ be an $m \times n$ random matrix with independent subgaussian entries $\mathbf{A}_{ij}$ of mean zero. Denote $K$ to be the maximum subgaussian norm of $\mathbf{A}_{ij}$, that is, $K > 0$ is the smallest number such that $\mathbb{E}[\exp(\mathbf{A}_{ij})^2/K^2] \leq 2$. Then for each $t > 0$,*

$$\mathbb{P}\left(\|\mathbf{A}\|_2 \geq 3K(\sqrt{m} + \sqrt{n} + t)\right) \leq 2\exp(-t^2). \tag{157}$$

*Proof.* See Theorem 4.4.5 in [54]. $\qquad\qquad\qquad\qquad\qquad\qquad\qquad\qquad\qquad\qquad\square$

Now we prove Theorem 4.1 for SMF-$\mathbf{W}$.

Recall that the ($L_2$-regularized) negative log-likelihood of observing triples $(y_i, \mathbf{x}_i, \mathbf{x}'_i)$ for $i = 1, \ldots, n$ is given as

$$\mathcal{L}_n := F(\mathbf{A}, \mathbf{B}, \boldsymbol{\gamma}) + \frac{1}{2(\sigma')^2}\|\mathbf{X}_{\text{aux}} - \mathbf{C}\|_F^2 + c, \tag{158}$$

where $c$ is a constant and $F$ is as in (11) or (12) depending on the activation type with tuning parameter $\xi = \frac{1}{2\sigma^2}$. Write the true parameter $\mathbf{Z}^\star = [\boldsymbol{\theta}^\star, \boldsymbol{\gamma}^\star]$. Recall that $\text{rank}(\boldsymbol{\theta}^\star) \leq r$ by the model assumption in (17).

***Proof of Theorem 4.1 for SMF-$\mathbf{W}$.*** Let us define the expected loss function $\bar{F}(\mathbf{A}, \mathbf{B}, \boldsymbol{\gamma}) := \mathbb{E}_{\boldsymbol{\varepsilon}_i, \boldsymbol{\varepsilon}'_i, 1 \leq i \leq n}[F(\mathbf{A}, \mathbf{B}, \boldsymbol{\gamma})]$. Define the following gradient mappings of $\mathbf{Z}^\star$ with respect to the empirical $F$ and the expected $\bar{F}$ loss functions:

$$G(\mathbf{Z}^\star, \tau) := \frac{1}{\tau}\left(\mathbf{Z}^\star - \Pi_\Theta\left(\mathbf{Z}^\star - \tau\nabla F(\mathbf{Z}^\star)\right)\right), \quad \bar{G}(\mathbf{Z}^\star, \tau) := \frac{1}{\tau}\left(\mathbf{Z}^\star - \Pi_\Theta\left(\mathbf{Z}^\star - \tau\nabla\bar{F}(\mathbf{Z}^\star)\right)\right). \tag{159}$$

It is elementary to show that the true parameter $\mathbf{Z}^\star$ is a stationary point of $\bar{F} - \lambda(\|\mathbf{A}\|_F^2 + \|\boldsymbol{\gamma}\|_F^2)$ over $\Theta \subseteq \mathbb{R}^{p \times (\kappa + n)} \times \mathbb{R}^{q \times \kappa}$. Hence we have $\bar{G}(\mathbf{Z}^\star, \tau) = 2\lambda[\mathbf{A}^\star, O, \boldsymbol{\gamma}^\star]$, so we may write

$$G(\mathbf{Z}^\star, \tau) = G(\mathbf{Z}^\star, \tau) - \bar{G}(\mathbf{Z}^\star, \tau) + 2\lambda[\mathbf{A}^\star, O, \boldsymbol{\gamma}^\star] \tag{160}$$

$$= \frac{1}{\tau}\left[\Pi_\Theta\left(\mathbf{Z}^\star - \tau\nabla F(\mathbf{Z}^\star)\right) - \Pi_\Theta\left(\mathbf{Z}^\star - \tau\nabla\bar{F}(\mathbf{Z}^\star)\right)\right] + 2\lambda[\mathbf{A}^\star, O, \boldsymbol{\gamma}^\star] \tag{161}$$

We will consider two cases, depending on whether the true parameter $\mathbf{Z}^\star$ satisfies the first-order optimality condition for $f$ over the convex constraint $\Theta$. The first-order optimality w.r.t. the low-rank constraint is handled directly by Theorem D.1.

**Case 1.** $\mathbf{Z}^\star - \tau\nabla F(\mathbf{Z}^\star) \in \Theta$ (In particular, this is the case where $\Theta$ equals the whole space).

In this case, we can disregard the projection $\Pi_\Theta$ in the above display so we get

$$G(\mathbf{Z}^\star, \tau) - 2\lambda[\mathbf{A}^\star, O, \boldsymbol{\gamma}^\star] = \nabla F(\mathbf{Z}^\star) - \nabla\bar{F}(\mathbf{Z}^\star) =: [\Delta\widetilde{\boldsymbol{\theta}}^\star, \Delta\widetilde{\boldsymbol{\gamma}}^\star]. \tag{162}$$

We will show that, for some constants $c, C > 0$, with probability at least $1 - n^{-1}$,

$$S := \sqrt{3r}\|\Delta\widetilde{\boldsymbol{\theta}}^\star\|_2 + \|\Delta\widetilde{\boldsymbol{\gamma}}^\star\|_F \leq c\sqrt{n}\log n + 3C\sigma(\sqrt{p} + \sqrt{n} + c\sqrt{\log n}). \tag{163}$$

By Theorem D.1 with $[\Delta\boldsymbol{\theta}^\star, \Delta\boldsymbol{\Gamma}^\star] := G(\mathbf{Z}^\star, \tau)$,

$$\|\mathbf{Z}_t - \mathbf{Z}^\star\|_F - \rho^t\|\mathbf{Z}_0 - \mathbf{Z}^\star\|_F \tag{164}$$

$$\leq \frac{\tau}{1 - \rho}\left(\sqrt{3r}\|\Delta\boldsymbol{\theta}^\star\|_2 + \|\Delta\boldsymbol{\gamma}^\star\|_F\right) \tag{165}$$

$$\leq \frac{\tau}{1 - \rho}\left(\sqrt{3r}(\|\Delta\widetilde{\boldsymbol{\theta}}^\star\|_2 + 2\lambda\|\mathbf{A}^\star\|_2) + (\|\Delta\widetilde{\boldsymbol{\gamma}}^\star\|_F + 2\lambda\|\boldsymbol{\gamma}^\star\|_F)\right). \tag{166}$$

It follows that with probability at least $1 - n^{-1}$,

$$\|\mathbf{Z}_t - \mathbf{Z}^\star\|_F \leq \rho^t\|\mathbf{Z}_0 - \mathbf{Z}^\star\|_F + \frac{\tau}{1 - \rho}\left(c\sqrt{n}\log n + 3C\sigma(\sqrt{p} + \sqrt{n} + c\sqrt{\log n})\right) \tag{167}$$

$$+ \frac{2\lambda\tau}{1 - \rho}\left(\sqrt{3r}\|\mathbf{A}^\star\|_2 + \|\boldsymbol{\gamma}^\star\|_F\right). \tag{168}$$

Now since $L/\mu < 3$ and $\tau \in (\frac{1}{2\mu}, \frac{3}{2L})$, there exists $\varepsilon > 0$ such that $\tau = \frac{1}{(2-\varepsilon)\mu}$. Then $\frac{\tau}{1-\rho} = \frac{\tau}{2\tau\mu-1} = \frac{1}{\varepsilon\mu}$. Thus, with probability at least $1 - n^{-1}$,

$$\|\mathbf{Z}_t - \mathbf{Z}^\star\|_F - \rho^t \|\mathbf{Z}_0 - \mathbf{Z}^\star\|_F \leq O\left(\frac{\sqrt{n}\log n + \lambda}{\mu}\right), \tag{169}$$

as desired.

Now we show (163). The argument is that, the norm of $[\Delta\widetilde{\boldsymbol{\theta}}^\star, \Delta\widetilde{\boldsymbol{\gamma}}^\star]$ can be decomposed into the sum of norms of random matrices with independent mean zero columns or mean zero Gaussian random matrices, which should have norm at most $\sqrt{n}\log n$ with high probability by standard matrix concentration inequalities.

We use the notation $\mathbf{U} = [\mathbf{A}^T, \boldsymbol{\gamma}^T]^T$, $\mathbf{U}^\star = [(\mathbf{A}^\star)^T, (\boldsymbol{\gamma}^\star)^T]^T$, $\boldsymbol{\Phi} = [\boldsymbol{\phi}_1, \ldots, \boldsymbol{\phi}_n] = [\mathbf{X}_{\text{data}}^T, \mathbf{X}_{\text{aux}}^T]^T$ (see also the proof of Theorem 3.5). Denote $\mathbf{a}_s = \mathbf{U}^T\boldsymbol{\phi}_s$ and $\mathbf{a}_s^\star = (\mathbf{U}^\star)^T\boldsymbol{\phi}_s$ for $s = 1, \ldots, n$ and introduce the following random quantities

$$\mathsf{Q}_1 := \left[\dot{\mathbf{h}}(y_1, \mathbf{a}_1), \ldots, \dot{\mathbf{h}}(y_n, \mathbf{a}_n)\right] \in \mathbb{R}^{\kappa \times n}, \tag{170}$$

$$\mathsf{Q}_2 := [\boldsymbol{\varepsilon}_1, \ldots, \boldsymbol{\varepsilon}_n] \in \mathbb{R}^{p \times n}, \quad \mathsf{Q}_3 := [\boldsymbol{\varepsilon}_1', \ldots, \boldsymbol{\varepsilon}_n'] \in \mathbb{R}^{p \times n}. \tag{171}$$

Recall that

$$\nabla_{\text{vec}(\mathbf{U})}F(\mathbf{U}, \mathbf{B}) = \left(\sum_{s=1}^n \dot{\mathbf{h}}(y_s, \mathbf{a}_s) \otimes \boldsymbol{\phi}_s\right) + 2\lambda\,\text{vec}(\mathbf{U}), \quad \nabla_{\mathbf{B}}F(\mathbf{U}, \mathbf{B}) = \frac{2}{2\sigma^2}(\mathbf{B} - \mathbf{X}_{\text{data}}), \tag{172}$$

$$\nabla_{\text{vec}(\mathbf{U})}\bar{F}(\mathbf{U}, \mathbf{B}) = \left(\sum_{s=1}^n \mathbb{E}\left[\dot{\mathbf{h}}(y_s, \mathbf{a}_s) \otimes \boldsymbol{\phi}_s\right]\right) + 2\lambda\,\text{vec}(\mathbf{U}), \quad \nabla_{\mathbf{B}}\bar{F}(\mathbf{U}, \mathbf{B}) = \frac{2}{2\sigma^2}(\mathbf{B} - \mathbf{B}^\star), \tag{173}$$

where $\dot{\mathbf{h}}$ is defined in (111). Note that

$$\mathbb{E}\left[\dot{\mathbf{h}}(y_s, \mathbf{a}_s)\,\Big|\,\boldsymbol{\phi}_s\right] = \left[\left(\frac{h'(\mathbf{a}[j])}{1 + \sum_{c=1}^\kappa h(\mathbf{a}[c])} - g_j(\mathbf{a}_s^\star)\frac{h'(\mathbf{a}[j])}{h(\mathbf{a}[j])}\right)_{\mathbf{a}=\mathbf{a}_s} ; j = 1, \ldots, \kappa\right] \tag{174}$$

$$= \left[\left(\frac{h'(\mathbf{a}[j])}{1 + \sum_{c=1}^\kappa h(\mathbf{a}[c])} - \frac{h(\mathbf{a}_s^\star[j])}{1 + \sum_{c=1}^\kappa h(\mathbf{a}_s^\star[c])}\frac{h'(\mathbf{a}[j])}{h(\mathbf{a}[j])}\right)_{\mathbf{a}=\mathbf{a}_s} ; j = 1, \ldots, \kappa\right], \tag{175}$$

so the above vanishes when $\mathbf{a}_s = \mathbf{a}_s^\star$. Hence

$$\mathbb{E}\left[\dot{\mathbf{h}}(y_s, \mathbf{a}_s^\star) \otimes \boldsymbol{\phi}_s\right] = \mathbb{E}\left[\mathbb{E}\left[\dot{\mathbf{h}}(y_s, \mathbf{a}_s^\star) \otimes \boldsymbol{\phi}_s\,\Big|\,\boldsymbol{\phi}_s\right]\right] = \mathbf{0}, \tag{176}$$

Hence we can compute the following gradients

$$\nabla_{\text{vec}(\mathbf{A})}(F - \bar{F})(\mathbf{A}, \mathbf{B}, \boldsymbol{\gamma}) = \left(\sum_{s=1}^n \dot{\mathbf{h}}(y_s, \mathbf{a}_s) \otimes \mathbf{x}_s\right) \tag{177}$$

$$\nabla_{\text{vec}(\boldsymbol{\gamma})}(F - \bar{F})(\mathbf{A}, \mathbf{B}, \boldsymbol{\gamma}) = \left(\sum_{s=1}^n \dot{\mathbf{h}}(y_s, \mathbf{a}_s) \otimes \mathbf{x}_s'\right) \tag{178}$$

$$\nabla_{\mathbf{B}}(F - \bar{F})(\mathbf{A}, \mathbf{B}, \boldsymbol{\gamma}) = \frac{2}{2\sigma^2}(\mathbf{B}^\star - \mathbf{X}_{\text{data}}) = \frac{2}{2\sigma^2}[\boldsymbol{\varepsilon}_1, \ldots, \boldsymbol{\varepsilon}_n]. \tag{179}$$

It follows that (recall the definition of $\gamma_{\max}$ in Assumption 3.3)

$$\|\nabla_{\mathbf{A}}(F - \bar{F})(\mathbf{A}^\star, \mathbf{B}^\star, \boldsymbol{\gamma}^\star)\|_2 = \left\|\sum_{s=1}^n (\mathbf{B}^\star[:, s] + \boldsymbol{\varepsilon}_s)\dot{\mathbf{h}}(y_s, \mathbf{a}_s^\star)^T\right\|_2 \tag{180}$$

$$\leq \left\|\sum_{s=1}^n \mathbf{B}^\star[:, s]\dot{\mathbf{h}}(y_s, \mathbf{a}_s^\star)^T\right\|_2 + \left\|\sum_{s=1}^n \boldsymbol{\varepsilon}_s\dot{\mathbf{h}}(y_s, \mathbf{a}_s^\star)^T\right\|_2 \tag{181}$$

$$\leq \|\mathbf{B}^\star\|_\infty \|\mathsf{Q}_1\|_2 + \gamma_{\max}\|\mathsf{Q}_2\|_2. \tag{182}$$

Similarly, we have

$$\|\Delta\widetilde{\boldsymbol{\gamma}}^{\star}\|_F = \|\nabla_{\boldsymbol{\gamma}}(F - \bar{F})(\mathbf{A}^{\star}, \mathbf{B}^{\star}, \boldsymbol{\gamma}^{\star})\|_F \leq \sqrt{q}\|\nabla_{\mathrm{vec}(\boldsymbol{\gamma})}(F - \bar{F})(\mathbf{A}^{\star}, \mathbf{B}^{\star}, \boldsymbol{\gamma}^{\star})\|_2 \qquad (183)$$

$$\leq \sqrt{q}\|\mathbf{C}^{\star}\|_{\infty}\|\mathsf{Q}_1\|_2 + \sqrt{q}\gamma_{\max}\|\mathsf{Q}_3\|_2. \qquad (184)$$

Using the fact that $\|[A, B]\|_2 \leq \|A\|_2 + \|B\|_2$ for two matrices $A, B$ with the same number of rows, we have

$$\left\|\Delta\widetilde{\boldsymbol{\theta}}^{\star}\right\|_2 = \left\|\nabla_{\mathbf{A}}(F - \bar{F})(\mathbf{A}^{\star}, \mathbf{B}^{\star}, \boldsymbol{\gamma}^{\star})\right\|_2 + \left\|\nabla_{\mathbf{B}}(F - \bar{F})(\mathbf{A}^{\star}, \mathbf{B}^{\star}, \boldsymbol{\gamma}^{\star})\right\|_2 \qquad (185)$$

$$\leq \|\mathbf{B}^{\star}\|_{\infty}\|\mathsf{Q}_1\|_2 + \gamma_{\max}\|\mathsf{Q}_2\|_2 + \frac{2}{2\sigma^2}\|\mathsf{Q}_2\|_2. \qquad (186)$$

Thus, combining the above bounds, we obtain

$$S = \sqrt{3r}\|\Delta\widetilde{\boldsymbol{\theta}}^{\star}\|_2 + \|\Delta\widetilde{\boldsymbol{\gamma}}^{\star}\|_F \leq \sum_{i=1}^{3} c_i\|\mathsf{Q}_i\|_2, \qquad (187)$$

where the constants $c_1, c_2, c_3 > 0$ are given by

$$c_1 := \left(\sqrt{3r}\|\mathbf{B}^{\star}\|_{\infty} + \sqrt{q}\|\mathbf{C}^{\star}\|_{\infty}\right), \quad c_2 := \sqrt{3r}(\gamma_{\max} + \frac{2}{2\sigma^2}), \quad c_3 := \sqrt{q}\gamma_{\max}. \qquad (188)$$

Next, we will use concentration inequalities to argue that the right hand side in (187) is small with high probability and obtain the following tail bound on $S$:

$$\mathbb{P}\left(S > c\sqrt{n}\log n + 3C\sigma(\sqrt{p} + \sqrt{n} + c\sqrt{\log n})\right) \leq \frac{1}{n}, \qquad (189)$$

where $C > 0$ is an absolute constant and $c > 0$ can be written explicitly in terms of the constants we use in this proof. This is enough to conclude (163).

Recall that for a random variable $Z$, its sub-Gaussian norm, denoted as $\|Z\|_{\psi_2}$, is the smalleset number $K > 0$ such that $\mathbb{E}[\exp(Z^2/K^2)] \leq 2$. The constant $C >$ above is the sub-gaussian norm of the standard normal variable, which can be taken as $C \leq 36e/\log 2$. Using union bound with Lemmas D.2 and D.3, for each $t, t' > 0$, we get

$$\mathbb{P}\left(S > c_1 t + 3(c_2 + c_3)C\sigma(\sqrt{p} + \sqrt{n} + t')\right) \qquad (190)$$

$$\leq \mathbb{P}\left(\|\mathsf{Q}_1\|_2 > t\right) + \left(\sum_{i=2}^{3}\mathbb{P}\left(\|\mathsf{Q}_i\|_2 > \frac{3C\sigma}{2}(\sqrt{p} + \sqrt{n} + t')\right)\right) \qquad (191)$$

$$\leq 2\kappa\exp\left(\frac{-t^2}{C_1^2\kappa^2 n}\right) + 2\exp(-(t')^2). \qquad (192)$$

Indeed, for bounding $\mathbb{P}(\|\mathsf{Q}_1\|_2 > t)$, we used Lemma D.2; for bounding tail probabilities of $\|\mathsf{Q}_2\|_2$ and $\|\mathsf{Q}_3\|_2$, we used Lemma D.3 with $K = \frac{C\sigma}{2}$ and $K = \frac{C\sigma'}{2}$, respectively. Observe that in order to make the last expression in (190) small, we will chose $t = c_4\sqrt{n}\log n$ and $t' = c_4\sqrt{\log n}$, where $c_4 > 0$ is a constant to be determined. This yields

$$\mathbb{P}\left(S > c_1 c_4\sqrt{n}\log n + 3(c_2 + c_3)C\sigma(\sqrt{p} + \sqrt{n} + c_4\sqrt{\log n})\right) \leq n^{-c_5}, \qquad (193)$$

where $c_5 > 0$ is an explicit constant that grows in $c_4$. We assume $c_4 > 0$ is such that $c_5 \geq 1$. This shows (189).

**Case 2. $\mathbf{Z}^{\star} - \tau\nabla F(\mathbf{Z}^{\star}) \notin \boldsymbol{\Theta}$.**

In this case, we cannot directly simplify the expression (160). In this case, we take the Frobenius norm and use non-expansiveness of the projection operator (onto convex set $\boldsymbol{\Theta}$):

$$\|G(\mathbf{Z}^{\star}, \tau) - \bar{G}(\mathbf{Z}^{\star}, \tau)\|_F = \frac{1}{\tau}\left\|\left[\Pi_{\boldsymbol{\Theta}}\left(\mathbf{Z}^{\star} - \tau\nabla F(\mathbf{Z}^{\star})\right) - \Pi_{\boldsymbol{\Theta}}\left(\mathbf{Z}^{\star} - \tau\nabla\bar{F}(\mathbf{Z}^{\star})\right)\right]\right\|_F \qquad (194)$$

$$\leq \|\nabla F(\mathbf{Z}^{\star}) - \nabla\bar{F}(\mathbf{Z}^{\star})\|_F \qquad (195)$$

$$\leq \|\Delta\widetilde{\boldsymbol{\theta}}^{\star}\|_F + \|\Delta\widetilde{\boldsymbol{\gamma}}^{\star}\|_F. \qquad (196)$$

According to Remark C.4, we also have Theorem C.2 (and hence Theorem 3.5) with $\sqrt{3r}\|\Delta\widetilde{\boldsymbol{\theta}}^{\star}\|_2$ replaced with $\|\Delta\widetilde{\boldsymbol{\theta}}^{\star}\|_F$. Then an identical argument with $\|\mathbb{Q}_i\|_F \leq \sqrt{\min(p,n)}\|\mathbb{Q}_i\|_2$ for $i=2,3$ shows

$$S' := \|\Delta\widetilde{\boldsymbol{\theta}}^{\star}\|_F + \|\Delta\widetilde{\boldsymbol{\gamma}}\|_F \leq c_1\|\mathbb{Q}_1\|_2 + c_2\sqrt{\min(p,n)}\|\mathbb{Q}_2\|_2 + c_3\sqrt{\min(p,n)}\|\mathbb{Q}_3\|_2, \quad (197)$$

where the constants $c_1, c_2, c_3 > 0$ are the same as in (188). So we have

$$\|\mathbf{Z}_t - \mathbf{Z}^{\star}\|_F \leq \rho^t\|\mathbf{Z}_0 - \mathbf{Z}^{\star}\|_F + \frac{\tau}{1-\rho}(S' + 2\lambda(\|\mathbf{A}^{\star}\|_2 + \|\boldsymbol{\gamma}^{\star}\|_F)). \quad (198)$$

Then an identical argument shows

$$\mathbb{P}\left(S' > c_1 t + 3(c_2 + c_3)C\sigma(\sqrt{p} + \sqrt{n} + t')\sqrt{\min(p,n)}\right) \quad (199)$$

$$\leq \mathbb{P}(\|\mathbb{Q}_1\|_2 > t) + \sum_{i=2}^{3}\mathbb{P}\left(\|\mathbb{Q}_i\|_2 > \frac{3C\sigma}{2}(\sqrt{p} + \sqrt{n} + t')\right), \quad (200)$$

and the assertion follows similarly as before. $\qquad\square$

It remains to show Theorem 4.1 for SMF-**H**.

***Proof of Theorem 4.1 for SMF*-H.** The argument is entirely similar to the proof of Theorem 4.1 for SMF-**W**. Indeed, denoting $\mathbf{a}_s = \mathbf{A}[:, s] + \boldsymbol{\gamma}^T\mathbf{x}'_s$ for $s = 1, \ldots, n$ and keeping the other notations the same as in the proof of Theorem 4.1, we can compute the following gradients

$$\nabla_{\mathbf{A}}(F - \bar{F})(\mathbf{A}, \mathbf{B}, \boldsymbol{\gamma}) = \left[\dot{\mathbf{h}}(y_1, \mathbf{a}_1), \ldots, \dot{\mathbf{h}}(y_n, \mathbf{a}_n)\right] \quad (201)$$

$$\nabla_{\text{vec}(\boldsymbol{\gamma})}(F - \bar{F})(\mathbf{A}, \mathbf{B}, \boldsymbol{\gamma}) = \left(\sum_{s=1}^{n}\dot{\mathbf{h}}(y_s, \mathbf{a}_s) \otimes \mathbf{x}'_s\right) \quad (202)$$

$$\nabla_{\mathbf{B}}(F - \bar{F})(\mathbf{A}, \mathbf{B}, \boldsymbol{\gamma}) = \frac{2}{2\sigma^2}(\mathbf{B}^{\star} - \mathbf{X}_{\text{data}}) = \frac{2}{2\sigma^2}[\boldsymbol{\varepsilon}_1, \ldots, \boldsymbol{\varepsilon}_n]. \quad (203)$$

Hence repeating the same argument as before, using concentration inequalities for the following random quantities $\mathbb{Q}_1, \mathbb{Q}_2, \mathbb{Q}_3$ we defined in (170), one can bound the size of $G(\mathbf{Z}^{\star}, \tau)$ with high probability. The rest of the details are omitted. $\qquad\square$

# E   Auxiliary computations

**Remark E.1.** Denoting $\xi = \xi'n$ and $\lambda = \lambda'n$, the condition $L/\mu$ in Theorem 3.5 for SMF-**W** reduces to

$$\frac{L^*}{\mu^*} < 3 \implies \left(\frac{L^*}{6} < \xi' < \frac{3\mu^*}{2}, \quad 0 \leq \lambda' < \frac{6\xi' - L^*}{2}\right) \cup \left(\xi' > \frac{3\mu^*}{2}, \quad \frac{2\xi' - 3\mu^*}{6} < \lambda' < \frac{6\xi' - L^*}{2}\right) \quad (204)$$

$$\frac{L^*}{\mu^*} \geq 3 \implies \left(\frac{L^* - \mu^*}{4} < \xi' < \frac{3(L^* - \mu^*)}{4}, \frac{L^* - 3\mu^*}{4} < \lambda' < \frac{6\xi' - L^*}{2}\right) \quad (205)$$

$$\cup \left(\xi' > \frac{3(L^* - \mu^*)}{2}, \frac{2\xi' - 3\mu^*}{6} < \lambda' < \frac{6\xi' - L^*}{2}\right). \quad (206)$$

# F   Auxiliary lemmas

**Lemma F.1.** *Fix a differentiable function* $f : \mathbb{R}^p \times \mathbb{R}$ *and a convex set* $\boldsymbol{\Theta} \subseteq \mathbb{R}^p$. *Fix* $\tau > 0$ *and*

$$G(\boldsymbol{\theta}, \tau) := \frac{1}{\tau}(\boldsymbol{\theta} - \Pi_{\boldsymbol{\Theta}}(\boldsymbol{\theta} - \tau\nabla f(\boldsymbol{\theta}))). \quad (207)$$

*Then for each* $\boldsymbol{\theta} \in \boldsymbol{\Theta}$, $\|G(\boldsymbol{\theta}, \tau)\| \leq \|\nabla f(\boldsymbol{\theta})\|$.

*Proof.* The assertion is clear if $\|G(\boldsymbol{\theta}, \tau)\| = 0$, so we may assume $\|G(\boldsymbol{\theta}, \tau)\| > 0$. Denote $\hat{\boldsymbol{\theta}} := \Pi_{\boldsymbol{\Theta}}(\boldsymbol{\theta} - \tau \nabla f(\boldsymbol{\theta}))$. Note that

$$\hat{\boldsymbol{\theta}} = \arg\min_{\boldsymbol{\theta}'} \|\boldsymbol{\theta} - \tau \nabla f(\boldsymbol{\theta}) - \boldsymbol{\theta}'\|^2, \tag{208}$$

so by the first-order optimality condition,

$$\langle \hat{\boldsymbol{\theta}} - \boldsymbol{\theta} + \tau \nabla f(\boldsymbol{\theta}), \, \boldsymbol{\theta}' - \hat{\boldsymbol{\theta}} \rangle \geq 0 \quad \forall \boldsymbol{\theta}' \in \boldsymbol{\Theta}. \tag{209}$$

Plugging in $\boldsymbol{\theta}' = \boldsymbol{\theta}$ and using Cauchy-Schwarz inequality,

$$\tau^2 \|G(\boldsymbol{\theta}, \tau)\|^2 = \|\boldsymbol{\theta} - \hat{\boldsymbol{\theta}}\|^2 \leq \tau \langle \nabla f(\boldsymbol{\theta}), \, \boldsymbol{\theta} - \hat{\boldsymbol{\theta}} \rangle \leq \tau \|\nabla f(\boldsymbol{\theta})\| \tau \|G(\boldsymbol{\theta}, \tau)\|. \tag{210}$$

Hence the assertion follows by dividing both sides by $\tau^2 \|G(\boldsymbol{\theta}, \tau)\| > 0$. $\square$

## G Experimental details

All numerical experiments were performed on a 2022 Macbook Air with M1 chip and 16 GB of RAM.

### G.1 Experiments on semi-synthetic MNIST dataset

We give more details on the semi-synthetic MNIST we used in the experiment in Figure 2. Denote $p = 28^2 = 784$, $n = 500$, $r = 2$, and $\kappa = 1$. First, we randomly select 10 images each from digits '2' and '5'. Vectorizing each image as a column in $p = 784$ dimension, we obtain a true factor matrix for features $\mathbf{W}_{\text{true},X} \in \mathbb{R}^{p \times r}$. Similarly, we randomly sample 10 images of each from digits '4' and '7' and obtain the true factor matrix of labels $\mathbf{W}_{\text{true},Y} \in \mathbb{R}^{p \times r}$. Next, we sample a code matrix $\mathbf{H}_{\text{true}} \in \mathbb{R}^{r \times n}$ whose entries are i.i.d. with the uniform distribution $U([0, 1])$. Then the 'pre-feature' matrix $\mathbf{X}_0 \in \mathbb{R}^{p \times n}$ of vectorized synthetic images is generated by $\mathbf{W}_{\text{true},X} \mathbf{H}_{\text{true}}$. The feature matrix $\mathbf{X}_{\text{data}} \in \mathbb{R}^{p \times n}$ is then generated by adding an independent Gaussian noise $\varepsilon_j \sim N(\mathbf{0}, \sigma^2 I_p)$ to the $j$th column of $\mathbf{X}_0$ for $j = 1, \ldots, n$, with $\sigma = 0.5$. We generate the binary label matrix $\mathbf{Y} = [y_1, \ldots, y_n] \in \{0, 1\}^{1 \times n}$ (recall $\kappa = 1$) as follows: Each entry $y_i$ is an independent Bernoulli variable with probability $p_i = \left(1 + \exp\left(-\boldsymbol{\beta}_{\text{true},Y}^T \mathbf{W}_{\text{true},Y}^T \mathbf{X}_{\text{data}}[:, i]\right)\right)^{-1}$, where $\boldsymbol{\beta}_{\text{true},Y} = [1, -1]$. No auxiliary features were used for the semi-synthetic dataset (i.e., $q = 0$).

### G.2 Experiments on the Job postings dataset

Next, we give details on the job postings dataset [6]. There are 17,880 postings and 15 variables in the dataset including binary variables, categorical variables, and textual information of *job description*. Among the 17,880 postings, 17,014 are true job postings (95.1%) and 866 are fraudulent postings (4.84%). This reveals a significant class imbalance, where the number of true postings greatly outweighs fraudulent ones, making this class imbalance a noteworthy characteristic of the dataset. In our analysis, we have coded the fake job postings as positive examples and the true job postings as negative examples.

In our experiments, we represented each job posting as a $p = 2480$ dimensional word frequency vector computed from its *job description* and augmented with $q = 72$ auxiliary features of binary and categorical variables, including indicators that specify whether a job posting has a company logo or if the posted job is in the United States. For computing the word frequency vectors, we represent the job description variable as a term/document frequency matrix with Term Frequency-Inverse Document Frequency (TF-IDF) normalization [43]. Common words that appear in all documents are assigned lower importance, while words specific to particular documents are deemed more significant. In our analysis, we utilized the 2,480 most frequent words for further examination.

### G.3 Details on CNN and FFNN

For the task of classifying microarray data into cancer classes, we compared the performance of our method with both CNN and FFNN in Figure 3. Specifically, the CNN architecture was designed with a convolutional layer with 32 filters and a kernel size of 3, followed by an average pooling layer

with a pool size of 2. Subsequently, a second convolutional layer with 64 filters and a kernel size of 3 was integrated, further followed by another average pooling layer with the same pool size. The architecture was finalized with a flatten layer, a fully connected layer of 128 neurons activated by ReLU, a dropout layer with a rate of 0.5, and a final fully connected layer with a sigmoid activation. On the other hand, the FFNN consisted of a fully connected layer featuring 64 neurons with ReLU, followed by a dropout layer with a regularization rate of 0.5. A subsequent fully connected layer with 32 neurons activated by the ReLU was incorporated, followed by a fully connected layer with a sigmoid function. This comparative analysis was repeated five times, consistent with the procedure outlined in the main paper.

An intriguing observation emerges from our benchmarking analysis. While the FFNN's performance on the breast cancer dataset is comparable to ours (The LPGD algorithms for SMF), the overall performance of CNN is notably inferior to ours. This disparity can primarily be attributed to the small sample size of the training set (145 samples for breast cancer and 25 samples for pancreatic cancer) in comparison to the substantial dimensionality of gene features (exceeding 30,000 features). We note that obtaining a substantial volume of biomedical data for cancer research is very expensive, making it challenging to feasibly train complex models such as deep neural networks. The significance of our approach becomes evident in its ability to retain robust performance even when facing the challenges posed by a restricted sample size and a complex high-dimensional feature landscape. Moreover, our method augments this resilience with the advantage of interpretability.