# OpenReview forum: "Exponentially Convergent Algorithms for Supervised Matrix Factorization"
_NeurIPS.cc/2023/Conference — NeurIPS 2023 poster_

### Official Review · Reviewer_ucPD · 2023-06-19

**Soundness:** 3 good
**Presentation:** 4 excellent
**Contribution:** 3 good
**Rating:** 7
**Confidence:** 3

**Summary:**

This paper proposed a novel supervised dictionary model, with two variations, one feature-based and one filter-based. The problem setting is on classification tasks with both high- and low-dimensional feautures, where the high dimensional features are learnd through dictionary learning, and then intergrated in to multinomial logistic regression.

The problem estimation is formulated to use a low-rank projected gradient descent algorithm. Exponential convergence guarantee is provided for the proposed algorithm.

**Strengths:**

The paper, overall, is written clearly and it is a pleasure to read.

1.  The proposed algorithm provides theoretical guarantees in terms of convergence.
2.  The proposed algorithm demonstrates improved performance compared to prior algorithms, as well as selected standard classification algorithms. It additionally exhibits good interpretability.

**Weaknesses:**

It would be beneficial to conduct benchmarking against deep neural networks in order to gain insights into gaps, if any, in performance and to understand potential trade-offs between interpretability and classification accuracy.

**Questions:**

A few typos:

-   Line 151, "is scalar and continuous"
-   Line 156, the equation should be the one mentioned on line 154 instead of (6).



**Limitations:**

The authors have sufficiently addressed the limitations.

---

> ### Author Rebuttal · Authors · 2023-08-09
>
> Thank you very much for your positive feedback on our submission. The reviewer has provided the following comments:
>
> **`Q1. It would be beneficial to conduct benchmarking against deep neural networks in order to gain insights into gaps, if any, in performance and to understand potential trade-offs between interpretability and classification accuracy.`**
>
> **Response**: Thank you very much for the suggestion. In the revision, we have included an additional benchmarking analysis involving deep convolutional neural networks (CNN) and feedforward neural networks (FFNN) to provide deeper insights into the performance of our proposed methods. (We report this result in the supplementary 1-page author response PDF).
>
> For the task of classifying microarray data into cancer classes, we compared the performance of our method with both CNN and FFNN. Specifically, the CNN architecture was designed with a convolutional layer with 32 filters and a kernel size of 3, followed by an average pooling layer with a pool size of 2. Subsequently, a second convolutional layer with 64 filters and a kernel size of 3 was integrated, further followed by another average pooling layer with the same pool size. The architecture was finalized with a flatten layer, a fully connected layer of 128 neurons activated by ReLU, a dropout layer with a rate of 0.5, and finally a fully connected layer with a sigmoid activation. On the other hand, the FFNN consists of a fully connected layer featuring 64 neurons with ReLU activation, followed by a dropout layer with a regularization rate of 0.5. A subsequent fully connected layer with 32 neurons activated by the ReLU was incorporated, followed by a fully connected layer with a sigmoid function. This comparative analysis was repeated five times, consistent with the procedure outlined in the main paper.
>
> In our experiment,  the CNN achieved an average accuracy of 0.769 (0.07) (here 0.07 is the standard deviation, and we use this notation hereafter) on the pancreatic cancer dataset, and an average accuracy of 0.854 (0.06) on the breast cancer dataset. In contrast, the FFNN yielded an average accuracy of 0.816 (0.04) for pancreatic cancer and 0.890 (0.02) for breast cancer. We also included the revised table with all methods in the one-page supplementary PDF for rebuttal so that the reviewer can check.
>
> An intriguing observation emerges from our benchmarking analysis. While the FFNN's performance on the breast cancer dataset is comparable to ours (The LPGD algorithms for SDL), the overall performance of CNN is notably inferior to ours. This disparity can primarily be attributed to the small sample size of the training set (145 samples for breast cancer and 25 samples for pancreatic cancer) in comparison to the substantial dimensionality of gene features (exceeding 30,000 features). We note that obtaining a substantial volume of biomedical data for cancer research is very expensive, making it challenging to feasibly train complex models such as deep neural networks. The significance of our approach becomes evident in its ability to retain robust performance even when facing the challenges posed by a restricted sample size and a complex high-dimensional feature landscape. Moreover, our method augments this resilience with the advantage of interpretability. We will include the comparison with CNN and FFNN, as well as the above discussion, in the revision.
>
> **`Q2. A few typos: Line 151, "is scalar and continuous"`**
>
> **Response**: Thank you for this comment. We fixed this in the revision.
>
> **`Q3. Line 156, the equation should be the one mentioned on line 154 instead of (6)."`**
>
> **Response**: Thank you for this comment. We fixed this in the revision.

---

### Official Review · Reviewer_bLMw · 2023-07-07

**Soundness:** 4 excellent
**Presentation:** 3 good
**Contribution:** 3 good
**Rating:** 7
**Confidence:** 4

**Summary:**

The work focuses on a generic supervised dictionary learning formulation, which is convex in each of the variable-blocks but not overall. The idea is to stack up the matrix-variables  and obtain an equivalent low-rank optimisation problem with a convex objective. And, then this is solved using a projected gradient descent algorithm (algo1).

Interestingly, under mild conditions, exponentially fast convergence to global minimiser for algo1 is proven. This is a far stronger guarantee than the popular plain block-coordinate descent (BCD), which (in general case) guarantees only stationary-point convergence.

 As a side-result, assuming a generative model for the data and under low noise conditions (<O(1/n)), statistical consistency of the sample based-optimal solution is also presented.

The convergences rates are empirically verified on synthetic data. Interestingly in fig3a, it is shown that just by using the proposed pgd solver instead of the popular BCD, one may achieve considerable improvement in application performance.

**Strengths:**

1. While BCD style of algorithms are popularly studied in this context, I think studying the lifted-pgd is interesting.
2. Theorem 3.5 result is interesting and potentially can be applied elsewhere.
3. Improvement over BCD in fig3a is considerable, highlight the impact of the study.

**Weaknesses:**

1. Though theoretical guarantees are proven, from simulations it is not clear how much is the improvement of pgd vs baseline (BCD) in terms of time. For e.g. in figure 2, it would have been very helpful if BCD is also included.



Very minor comment:
1. The presentation of Fig 2 & Eqn 9 could be adjusted to not interfere with the text.


**Questions:**

Following points may improve the presentation of this already good submission

1. I guess the order in which we perform the projection \pi_\Theta and \pi_r matters for the rate of convergence. Also, as noted in the paper, works like [12] alternate between projections. Any discussion on this may help the reader appreciate the algorithm better. For example if one does \pi_r followed by \pi_\Theta, then what may happen? Why not alternate between projections? do these make the convergence analysis difficult or is it that the convergence actually slows down if the order of projection is tampered with ?

2. Will a distribution free analysis in section 4 be more interesting ? With feasibility set being bounded and the objective in terms of the data is lips.conts., this should be possible.


**Limitations:**

Limitations are clearly mentioned in the paper.

---

> ### Author Rebuttal · Authors · 2023-08-09
>
> We greatly appreciate the reviewer’s overall positive comments.
>
> > Though theoretical guarantees are proven, from simulations it is not clear how much is the improvement of pgd vs baseline (BCD) in terms of time. For e.g. in figure 2, it would have been very helpful if BCD is also included.
>
> **Response:** We sincerely thank your suggestion here! We added comparison with BCD in Figure 2 in the revision. We also included the revised Figure 2 in the one-page supplementary PDF for rebuttal so that the reviewer can check.
>
>
> > Very minor comment: The presentation of Fig 2 & Eqn 9 could be adjusted to not interfere with the text.
>
> **Response:** Fixed.
>
>
> >Questions: I guess the order in which we perform the projection $\Pi_{\Theta}$ and $\Pi_r$ matters for the rate of convergence. Also, as noted in the paper, works like [12] alternate between projections. Any discussion on this may help the reader appreciate the algorithm better. For example if one does $\Pi_r$ followed by $\Pi_\Theta$, then what may happen? Why not alternate between projections? do these make the convergence analysis difficult or is it that the convergence actually slows down if the order of projection is tampered with ?
>
>
> **Response:** Thanks for these great questions. First, in principle, one could alternate between the two projections $\Pi_{r}$ and $\Pi_{\Theta}$ at every iteration after a gradient descent step until convergence, similarly to the alternating projection in [12]. However, this would make each iteration of the algorithm prohibitively expensive as this requries to perform rank-$r$ SVD until convergence at *every iteration*. The problem in [12] is much simpler than ours as the objective function is simply the Frobenius norm between the target and the estimated low-rank constrained matrix.
>
> Second, it is a great insight to switch the order of two projections $\Pi_{r}$ and $\Pi_{\Theta}$. Our proposed LPGD algorithm performs the convex projection $\Pi_{\Theta}$ first and then applies the low-rank projection $\Pi_{r}$. The key inequality we derive in the proof of Thm. C.2 is (eq. (60) in the submission)
> $$
> \lVert \mathbf{Z}_\text{$t$} - \mathbf{Z}^{\star}  \rVert_F  \le 2\eta  \lVert \mathbf{Z} _{t-1} - \mathbf{Z}^{\star}\rVert_F + \lVert \Pi_t (\tau \Delta _{\Theta} \mathbf{Z}^{\star} )  \rVert_F,
> $$
>
>
>
> where $\tau \Delta_\text{$\Theta$}\mathbf{Z}^{\star}:=\mathbf{Z}^{\star} - \Pi_\text{$\Theta$}(\mathbf{Z}^{\star}-\tau \nabla f(\mathbf{Z}^{\star}))$ denotes the gradient mapping at $\mathbf{Z}^{\star}$ w.r.t. the convex constraint $\Theta$, and $\Pi_\text{$t$}$ is a linear projection onto a $3r$-dimensional linear subspace that depends $\mathbf{Z}^{\star}$, $\mathbf{Z} _{t}$, and $\mathbf{Z} _\text{$t-1$}$. The last error term above can be bounded above uniformly in $t$ using $\lVert \Pi _\text{$t$}( A)\rVert _\text{$F$}\le \sqrt{3r} \lVert A \rVert _\text{2}$. So we can apply the above inequality recursively to obtain the desired result.
>
> Now if we consider an alternative algorithm that uses the  low-rank projections $\Pi_{r}$  first and then the convex projection $\Pi_{\Theta}$, then we can derive a corresponding key inequality:
> $$
> 		\hspace{-2cm}(*)\cdots  \qquad \qquad 	\lVert \mathbf{Z} _\text{$t$} - \mathbf{Z}^{\star}  \rVert _\text{$F$}  \le  2\eta \, \lVert  \mathbf{Z} _\text{$t-1$} - \mathbf{Z}^{\star}\rVert _\text{$F$}+ \lVert  \tau\Delta^{t} \mathbf{Z}^{\star} \rVert _\text{$F$},
> $$
>
> where $\tau \Delta^{t}\mathbf{Z}^{\star}:=\mathbf{Z}^{\star} - \Pi _{t}(\mathbf{Z}^{\star}-\tau \nabla f(\mathbf{Z}^{\star}))$ denotes the gradient mapping at $\mathbf{Z}^{\star}$ w.r.t. the *virtual* linear constraint that we constructed during the proof to approximate the low-rank constraint. Indeed, the above inequality can be derived by modifying the argument in (42)-(47) in the submission assuming the reverse order of projections. We omit the details due to the space constraint.
>
>
>
> Then by recursively applying the inequality $(*)$, we can obtain
> $$
> \lVert \mathbf{Z} _\text{$t$} - \mathbf{Z}^{\star}  \rVert _\text{$F$}  \le  (2\eta)^{t}   \lVert  \mathbf{Z} _\text{$0$} - \mathbf{Z}^{\star}\rVert _\text{$F$}  +  \sum _\text{$k=1$} ^{t} (2\eta)^{t-k} \lVert  \tau\Delta^{k} \mathbf{Z}^{\star} \rVert _\text{$F$}.
> $$
>
> Hence the rate of convergence we would get is the same as the original algorithm, but the additive error takes a different form. Since the "low-rank gradient mapping" $\Delta^{k}\mathbf{Z}^{\star}$ depends on the iterates $\mathbf{Z} _\text{$k$}, \mathbf{Z} _\text{$k-1$}$, we find it easier to control the gradient mapping with respect to the convex projection that comes out from the analysis of the original algorithm.
>
> We absolutely agree with the reviewer that this discussion will be helpful for the readers. We will add this discussion as a remark in the revision.
>
>
>
> > Question: Will a distribution free analysis in section 4 be more interesting ? With feasibility set being bounded and the objective in terms of the data is lips.conts., this should be possible.
>
> **Response:** This is an excellent suggestion. The only distributional assumption we have made in Section 4 is that the noise terms $\epsilon_{i}$ and $\epsilon_{i}'$ follow Gaussian distributions with a mean of zero. In fact, our current analysis holds when assuming only a sub-Gaussian distribution, as the only place we used Gaussian distributional assumption was in the concentration inequalities in Lemmas D.2 and D.3 (in order to obtain tail bounds in eq. (162) and (174)-(175)), which are already stated for the sub-Gaussian case. If we restrict the support of noise to be bounded, then similar concentration inequalities hold without any distribution assumption (e.g., by matrix Burnstein inequality). Such bounded noise assumption might be reasonable when the feasibility set is bounded, as the reviewer pointed out. We will add this discussion in the revision.

---

> > ### Comment · Reviewer_bLMw · 2023-08-15
> >
> > Thanks for the detailed rebuttal. I have read it and other reviews. My concerns have been addressed and I feel it’s a nice idea with publishing. I would like to keep my previous score. Thanks.

---

> > > ### Author Response · Authors · 2023-08-16
> > >
> > > Thank you very much. We again appreciate your thoughtful comments and effort for reviewing our submission.

---

### Official Review · Reviewer_zTTY · 2023-07-09

**Soundness:** 3 good
**Presentation:** 3 good
**Contribution:** 2 fair
**Rating:** 6
**Confidence:** 2

**Summary:**

This paper explores the optimization of supervised dictionary learning, focusing on a non-convex objective function with a matrix factorization structure. The authors propose a reformulation of the problem as a minimization task with a low-rank constraint. To solve this reformulated problem, they employ a projection gradient descent-type method. The paper provides convergence results and statistical analysis, along with practical applications to showcase the efficacy of the proposed algorithm.

**Strengths:**

Optimizing non-convex structured problems poses intriguing and challenging tasks. The problem addressed in this paper represents a classic learning paradigm in the machine learning field, making the authors' contribution well-motivated. Additionally, they demonstrate the effectiveness of their new PGD-variant through classification tasks using medical data. The paper is well-written, presenting its ideas coherently.

**Weaknesses:**

Overall, the paper is of a standard quality. Considering the extensive research on optimization problems with low-rank matrix factorization structure, the newly proposed techniques appear nontrivial yet not entirely surprising. My primary concern lies with the claim in the abstract (and other sections of the paper, e.g., L55) that the proposed method "provably converges exponentially fast to a global minimizer of the objective." Here are my questions:

* [Clarification question A] It appears that the global convergence result (Theorem 3.5) holds only when a low-rank stationary point Z^* exists, where "stationary" is defined as first-order optimality with respect to F under the convex constraint \Theta. In such a case, the objective function seems to be already strongly convex, and the mentioned existence might imply that the low-rank constraint could be eliminated (please correct me if I am mistaken). This assumption is quite strong and essentially transforms a non-convex problem into a convex one. I believe this could be a significant limitation of the analysis.

* [Clarification question B] I acknowledge that the authors present a general convergence result in Theorem D.1. My second question pertains to Theorem D.1(ii). It seems that, under the "possibly misspecified case," Theorem D.1(ii) cannot even guarantee the sequential convergence of {Z_t} since the residual term (second term on the right-hand side of Equation (91)) might not be zero even at the global optimal Z^*. I fail to see (please correct me if I am mistaken) why the gradient mapping would be zero at optimal Z^*, as this mapping does not consider the normal cone (defined in a certain generalized sense) of the rank constraint.

**Questions:**

See above.

**Limitations:**

Yes.

---

> ### Author Rebuttal · Authors · 2023-08-09
>
> **`Clarification question A`**
>
> **Response:** We would like to express our gratitude for the thoughtful comments provided by the reviewer. To begin, we wish to emphasize that our theoretical analysis comprises three parts:
> > 1. (Thm. C.2) Establishing exponential convergence results for the general low-rank projected gradient descent (LPGD) algorithm for $r$-restricted strongly convex (RSC) and $r$-restricted smooth (RSM) objectives.
>
> > 2. (Thm. 3.5, Thm. D.1) Conducting an extensive second-order analysis of the SDL objective to verify that the reformulated SDL problems satisfy the hypotheses of Thm. C.2.
>
> > 3. (Thm 4.1) Establishing statistical estimation guarantee for SDL under generative models by using previous computational guarantees (Thm D.1).
>
> The reviewer's concern appears to stem from mixing the two statements in Thm 3.5 (on SDL) and Thm C.2 (on LPGD). Indeed, the objective functions in Thm 3.5 correspond to the lifted SDL objectives in eq. (8) and (9). Notably, these objectives are not inherently ($r$-restricted) strongly convex. In fact, verifying that the lifted SDL objectives do satisfy the RSC/RSM properties is the content of the contribution 2 mentioned above.
>
> Furthermore, we emphasize that the proof of Thm. C.2.(i) for the correctly specified case, where the assumed RSC/RSM objective has a rank-$r$ stationary point, is nontrivial, since the strong convexity holds only when we restrict the objective on matrices with rank at most $r$. Thus one cannot eliminate the low-rank constraint, as opposed to the reviewer's comment.
>
> More specifically, even if the objective function admits a low-rank stationary point $\mathbf{Z}^{*}$, there could be many other stationary points among matrices with rank $>r$. Moreover, since our LPGD algorithm in general goes in (after low-rank projection) and out (after gradient descent in the ambient space) of the low-rank space, one needs to carefully address that the possibly wild landscape outside of the low-rank space does not significantly impact the convergence.
>
> In addition, Theorem 3 of Zhu et al. '18 (see below) shows that if an objective function with a matrix input is both RSC and RSM, with a condition number $L/\mu<3/2$, and if it admits a low-rank critical point (without additional convex constraint), then there are no spurious local minima in the factored parameter space:
>
> > [1] Zhihui Zhu, Qiuwei Li, Gongguo Tang, and Michael B Wakin. *Global optimality in low-rank matrix optimization.* IEEE Transactions on Signal Processing, 66(13):3614–3628, 2018.
>
> This result also indicates that minimizing an RSC function with a low-rank critical point is a nontrivial problem. To put in context, our Thm C.2(i) shows that under the hypothesis of a condition number $L/\mu<3$ (weaker than in Zhu et al.) and with a convex constraint, the assumed low-rank stationary point $\mathbf{Z}^{\star}$ is the global minimizer of the objective among the low-rank matrices and that the LPGD algorithm converges to $\mathbf{Z}^{\star}$.
>
>
> **`Clarification question B`**
>
> **Response:** Thank you for your thoughtful question.
>
>  We do see the reviewer's concern regarding the claimed global exponential convergence to the global minimizer of the SDL problem. The exponential convergence technically holds for the correctly specified case, and for the misspecified case, the exponential convergence is up to an additive error that depends on the extent of misspecification. We will revise the statement in the abstract and throughout the paper to clarify this point. We would also like to bring the Reviewer’s attention to two relevant points below.
>
> First, even in the presence of a nonzero misspecification error (which bounds the unnormalized estimation error $\lVert \mathbf{Z}^{\star}-\mathbf{Z}_\text{$\infty$} \rVert_F$, $\mathbf{Z}^{\star}\in \mathbb{R}^{p\times n}\times \mathbb{R}^{q\times n}$), our Thm. 4.1 demonstrates that, under natural generative models for SDL, this error becomes vanishingly small with high probability with noise variance $\sigma^{2}=O(1/n)$ for SDL-$\mathbf{W}$ and $\sigma^{2}=o(1/\sqrt{n})$ for SDL-$\mathbf{H}$. Roughly speaking, these results indicate that the generative SDL models are nearly correctly specified with high probability. As a result, our algorithm achieves exponential convergence to the correct parameters for the generative SDL model up to a statistical error that vanishes as the sample size $n$ tends to infinity.
>
> Secondly, it seems to be challenging to exactly recover the global minimizer of an RSC function under a low-rank constraint even when there is no additional convex constraint. To the best of our knowledge, all existing works on similar low-rank matrix estimation problems without additional assumptions (e.g., incoherence assumption for matrix sensing problems) recover global optimum up to a misspecification error, which is zero when there is a low-rank stationary point (correctly specified) but is nonzero otherwise. This is the case, for example, in the following references:
>
> >[2] Lingxiao Wang, Xiao Zhang, and Quanquan Gu. *A unified computational and statistical framework for nonconvex low-rank matrix estimation.* In Artificial Intelligence and Statistics, pages 981–990. PMLR, 2017.
>
> >[3] Dohyung Park, Anastasios Kyrillidis, Constantine Caramanis, and Sujay Sanghavi. *Finding low-rank solutions via nonconvex matrix factorization, efficiently and provably.* SIAM Journal on Imaging Sciences, 11(4):2165–2204, 2018.
>
> >[4] Sahand Negahban and Martin J Wainwright. *Estimation of (near) low-rank matrices with noise and high-dimensional scaling.* The Annals of Statistics, 39(2):1069–1097, 2011.

---

> > ### Comment · Reviewer_zTTY · 2023-08-18
> >
> > Thank the authors for the clarification, which has addressed my previous concerns. I have adjusted my score accordingly.

---

### Official Review · Reviewer_pg2E · 2023-07-16

**Soundness:** 2 fair
**Presentation:** 3 good
**Contribution:** 2 fair
**Rating:** 4
**Confidence:** 4

**Summary:**

This paper proposes a variant of supervised dictionary learning (SDL), provides some theoretical guarantee on finding the global minimizer of the problem with arbitrary initialization, and showcase its application in pancreatic cancer.

**Strengths:**

Theoretical analysis on dictionary learning (DL) has a rich literature. However, theoretical analysis on convergence of supervised DL is scarce. So, this is of interest. The paper is written clear.

**Weaknesses:**

The paper lacks a proper literature, introduction, and formulation of the dictionary learning problem. Several statements in this paper are not properly claimed. The paper in its formulation is not rigorous. Here are some indication of this.

1. Abstract:
	- SDL is not properly defined. There is no info on its key property which is trying to represent data with a sparse combination of columns of a dictionary, which itself is learned.

	- The global convergence with random initialization is a bold statement. Indeed, this is misleading, as this paper (I explain below) is solving a different problem than DL or SDL. I note that all theoretical analysis on DL work with the assumption that the initial estimation of the dictionary must be close to the true one [1,2,3,4] (to name a few)

	- There is no reference to the sparsity characteristic of sparse coding/dictionary learning. So, then what is the motivation behind using SDL and not another method?

2. Line 24, DL is not introduced properly. DL is a method trying to learn a sparse representation that can explain the data through a dictionary in a generating fashion.

3. Line 26, This is confusing. DL is a specific model with sparsity. NMF, PCA, ... are different models. However, the statement is mixing all up.

4. Line 26, These citations are proper but old (there are many other recent works on DL).

5. Line 41, DL extract a high-dimension sparse features (D is overcomplete) not a low-dim feature.

6. Line 48, Literature on theoretical analysis on convergence and recovery for DL is missing.

7. Line 50, Theoretical analysis on DL have generative perspective such that there is a sparse representation that has generated the data (or in the SDL case, is also mapped to a class label), and the goal is given data, learn the dictionary and recover that sparse representation. DL is not about global minimizer of an objective (as the solution to overcomplete dictionary may not be unique without sparsity).

8. Why there is no notion of sparsity on H in (3)? Same on line 117.

9. Related work is missing DL literature. Moreover, DL is a bi-convex problem.

10. Line 154 is bi-convex in (beta, W) and (H). (6) objective is exactly similar to the one above it. Moreover, the objective is bi-convex not convex.

11. For section 3, theoretical analysis on SDL may aim to recover the sparse representation that has generated the data, and has produced the label through a probabilistic generative model. This paper is solving another problem, hence their convergence analysis does not address the SDL problem.

Overall, this paper's analysis is not on recovery of the sparse representation that has generated (explained) the data. Hence, statements are confusing in this regard. The paper provides convergence on another parameter theta.


[1] Agarwal, A., Anandkumar, A., Jain, P., & Netrapalli, P. (2016). Learning sparsely used overcomplete dictionaries via alternating minimization. SIAM Journal on Optimization, 26(4), 2775-2799.

[2] Chatterji, N. S., & Bartlett, P. L. (2017). Alternating minimization for dictionary learning: Local convergence guarantees. arXiv preprint arXiv:1711.03634.

[3] Rambhatla, S., Li, X., & Haupt, J. (2018, September). NOODL: Provable Online Dictionary Learning and Sparse Coding. In International Conference on Learning Representations.

[4] Arora, S., Ge, R., Ma, T., & Moitra, A. (2015, June). Simple, efficient, and neural algorithms for sparse coding. In Conference on learning theory (pp. 113-149). PMLR.

**Questions:**

See above.

**Limitations:**

See above.

---

> ### Author Rebuttal · Authors · 2023-08-09
>
> We appreciate your evaluation of our work, which greatly contributes to its enhancement. We value your expertise in the field and understand your concern about the term DL being closely linked to sparse representation. While we acknowledge the prevalent association of DL with recovering overcomplete dictionaries and sparse representations, our primary focus is on *undercomplete/low rank* SDL in *high-dimensional* setting.
>
> We would like to request the opportunity to delve deeper into our approach's motivations and implications either during the rebuttal process or through subsequent discussions. At a minimum, we are committed to clearly distinguishing our high-dimensional SDL via undercomplete DL from the conventional overcomplete DL. To facilitate reader comprehension, we are open to altering our terminology. Terms like "supervised undercomplete dictionary learning" or "supervised matrix factorization" could aptly encapsulate our approach's essence and better distinguish it from the supervised version of the standard overcomplete DL with sparse representation.
>
> In response to your feedback:
>
> **(1)-(2)**
>
> **Motivation for Undercomplete SDL**: Our motivation for undercomplete dictionaries in SDL stems from the classification of high-dimensional data (e.g., genomic data analysis for cancer classification). We aim to learn a low-rank basis that offers interpretable, data-reconstructive, and class-discriminative features, addressing challenges posed by high-dimensional data. The overcomplete dictionary approach, with its associated sparse representation, proves computationally infeasible and complex to interpret. Our preference for undercomplete dictionaries aligns with our goals. Additionally, we find that assuming sparse representation over an undercomplete dictionary for high-dimensional data isn't always reasonable. As a result, we've excluded the sparsity regularizer for the code matrix $H$.
>
> **Related Literature on Undercomplete/Low-rank DL**: Although most DL literature emphasizes overcomplete dictionaries and sparse representation, notable works explore undercomplete or low-rank dictionaries for various purposes (examples refs below). Notably, [1] introduces undercomplete dictionaries for efficient SDL tasks on high-dimensional data, resembling our approach (without theoretical guarantees). In [2], the identifiability of sparse component analysis is examined under an undercomplete dictionary. [3] employs undercomplete dictionary learning for low-rank dictionary learning, leading to unsupervised low-rank feature extraction.
>
> [1] Mohseni-Sehdeh et al. "A Fast Dictionary-Learning-based Classification Scheme Using Undercomplete Dictionaries." Signal Processing (2023): 109124.
>
> [2] Cohen et al. "Identifiability of complete dictionary learning." SIAM Journal on Mathematics of Data Science 1.3 (2019): 518-536.
>
> [3] Parsa et al. "Low-rank dictionary learning for unsupervised feature selection." Expert Systems with Applications 202 (2022): 117149.
>
> **(3)**
> We'll revise this to "matrix factorization" to better encompass a broader spectrum of basis-learning problems.
>
> **(4)**
> We'll enhance our reference section with more recent DL sources.
>
> **(5)**
> DL with overcomplete dictionary does extract high-dimensional features, but DL with either low-rank or undercomplete dictionary extracts low-dimensional features. We will clarify this point.
>
> **(6)**
> Acknowledging the importance of theoretical analysis on DL, we'll incorporate the references you suggested, which delve into aspects like local convergence and recovery guarantees.
>
> **(7)**
> Thank you for your insightful comment. The classical references [30, 33] define the SDL problem as an optimization problem, with the possibility of undercomplete $r<p$ dictionary not excluded. Our formulation of SDL (eq. (3)) mirrors [30] (eq. (4)) without sparseness regularizer under $r<p$. Our work guarantees that one can find *some* global minimizer of the non-convex SDL objective exponentially fast from any initialization (under some conditions). This is by no means about recovering the ground-truth dictionary and representation. In fact, as we noted in the first paragraph of Sec. 3 (and also as the reviewer correctly pointed out), our SDL optimization problem does not have a unique global minimizer. However, if we transform separate matrix factors into a combined low-rank matrix (denoted $\theta$), then uniqueness is guaranteed. Under this setting, we also obtain statistical estimation guarantee under generative SDL models, see Sec. 4.
>
> **(8)**
> See our response for 1 and 2.
>
> **(9)**
> While we're constrained by space, we concur on the importance of showcasing recent DL advances. These references would fortify our work's positioning.
>
> **(10)**
> Acknowledging the non-convexity of (6), if we introduce the combined low-rank matrix factor $\theta$ (in line 162), then the objective function in (6) is quadratic $\theta\mapsto \lVert \theta_{0} - \theta \rVert_{F}^{2}$ for a fixed $\theta_{0}$. One can then minimize this convex function under a low-rank constraint on $\theta$,  yielding solutions for $\beta$, $W$, and $H$. These points are stated in lines 160-163. We will further clarify our discussion in lines 156-163.
>
> **(11)**
> While the recovery of the ground truths is a natural goal of an SDL that draws a direct analogy from the literature of overcomplete DL, it is not what we aim for in this work. Our underlying assumption is that the observed high-dimensional labeled data is generated by a linear combination of unknown low-rank features. In this setting, there are infinitely many, equally effective undercomplete dictionaries and representations (not necessarily sparse) that could have generated the observed labeled data. Hence our goal is to find an optimally effective pair of class-discriminative dictionaries and data representation by solving an optimization problem.
>
> We sincerely hope our clarifications assuage your concerns, encouraging your reconsideration.

---

> > ### Comment · Reviewer_pg2E · 2023-08-14
> > **Reviewer's Comment after Rebuttal**
> >
> > I thank the authors for their rebuttal, and appreciate their explanation. My concern remains on the following: this paper is applying a matrix factorization (a step very similar to PCA) but refers to the method as dictionary learning (which is unfamiliar to the general reader of the literature). I found the text to have several statements (which I noted in my original review, e.g., the discussion around Eq. (6)) that are not fully correct; it's hard to evaluate if those will be fully addressed upon acceptance (see below for more explanation).
> >
> > - Overcomplete/undercomplete: I now understand that your focus is on learning a low-dimensional representation and provide a low-rank matrix factorization. I strongly suggest improving the literature on including sparse (overcomplete) dictionary learning, as this is a most well-known case; when one talks about dictionary learning. Moreover, I note that one may still want to apply sparsity on the low-dimensional representation (see [1] which uses an undercomplete dictionary and apply sparsity on high-dimensional gene perturbation data).
> >
> > - On the usage of dictionary learning: this paper is performing a matrix factorization or the method (in line 174) is explaining an approach similar to PCA through SVD (if not exactly the same). Applying a least square optimization on data, while enforcing orthogonality on the basis, recovers PC (up to permutations). I suspect that applying PCA on the data, following a similar procedure in this paper, should result in very similar performance. I appreciate the authors to provide such comparison and an explanation on how this approach differs from using PC.
> >
> > [1] Pan, J., Kwon, J. J., Talamas, J. A., Borah, A. A., Vazquez, F., Boehm, J. S., ... & Hahn, W. C. (2022). Sparse dictionary learning recovers pleiotropy from human cell fitness screens. Cell systems, 13(4), 286-303.

---

> > > ### Author Response · Authors · 2023-08-14
> > >
> > > We appreciate the reviewer's valuable feedback on our rebuttal.
> > >
> > > >*I found the text to have several statements (which I noted in my original review, e.g., the discussion around Eq. (6)) that are not fully correct*..
> > >
> > > **Response.** We have elaborated on this point in item 10 in the rebuttal, but here take another chance for further clarification. In the submission lines 155-162, we wrote
> > >
> > > >>Instead, consider reformulating the above nonconvex problem into a problem with a convex objective function by suitably stacking up the matrices using the following matrix factorization: .. eq. (6) .. Proceeding one step further, another important observation we make is that it is also equivalent to finding a *single* matrix $\theta:=[ \beta^{T} H \parallel  W H ]\in \mathbb{R}^{(1+p)\times n}$ of rank at most $r$ that minimizes the function $f$ in (6), which is convex (specifically, quadratic) in $\theta$.
> > >
> > > The first sentence reads as (6) itself has a convex objective function, which is indeed not correct.  Our claimed reformulation of the nonconvex problem is by the matrix stacking in (6) AND the change of variable using $\theta$ after "proceeding one step further" above. We fully agree that our phrasing was not very clear, and we will clarify this point in the revision.
> > >
> > >
> > > >Overcomplete/undercomplete: I now understand that your focus ..
> > >
> > > **Response.** We very much appreciate that the reviewer now understands that we seek for low-dimensional representation of labeled data. As we have written in our previous rebuttal, as the reviewer suggests strongly, we will improve on reviewing the literature of sparse (overcomplete) dictionary learning.
> > >
> > > >Moreover, I note that one may still want to apply sparsity on..
> > >
> > > **Response.** We appreciate the reviewer providing this valuable reference. We will add a discussion on low-dimensional and sparse representation with the suggested reference.
> > >
> > > >On the usage of dictionary learning: this paper is performing a matrix factorization or the method (in line 174) is explaining an approach similar to PCA through SVD (if not exactly the same). ..  I appreciate the authors providing such a comparison and an explanation of how this approach differs from using PC.
> > >
> > >
> > > **Response.** The reviewer's suspicion that whether our method of effectively solving a supervised low-rank matrix factorization problem, is essentially (if not exactly) the same as applying standard PCA via SVD, is entirely not true. We have already discussed thoroughly why supervision makes a great difference in unsupervised matrix factorization. Our experiment in Section 6 is devoted to demonstrating this point, which the reviewer might have missed. We will elaborate on the difference in experimental and theoretical aspects below.
> > >
> > > As we have demonstrated in Figure 3 in the original submission, unsupervised PCs could result in poor performance in classification tasks. In Figure 3 **a**, the benchmark method "MF-LR" refers to applying the standard PCA first and then using the resulting low-dimensional representation for logistic regression. This method significantly underperforms our SDL method, especially poorly on the breast cancer dataset. Also in Figure 3 **b**, we visualize the PCs learned by the standard PCA from the pancreatic cancer dataset. Not only this method underperforms our SDL (73% vs. 96%), the PCs do not detect any clinically known prognostic markers. On the contrary, our SDL method (shown in Fig 3 **c**) achieves much higher accuracy (96%) and the detected *supervised* dictionary contains several clinically known prognostic markers (panel **d**). We also have provided extended experiments in the rebuttal on breast cancer, where our method even detected the well-known oncogene BRCA1 of breast cancer (provided in our 1-page supplementary rebuttal). There have also previous works on supervised PCA (e.g., [45]), experimentally validating the need for supervising PCs.
> > >
> > > Theoretically, the objective function of SDL in (3) combines the matrix factorization and classification loss. For instance, for the filter-based SDL (SDL-$W$), the objective function is
> > > \begin{align}
> > > 		\min _{W,H,\beta,\gamma }     \sum _{i=1}^{n}  \ell(y _{i},  \beta^T W^{T} x _i+\gamma^T x' _i)  +   \xi \lVert X _{\text{data}} - W H \rVert _{F}^{2},
> > > 	\end{align}
> > > The above optimization problem cannot be solved by a single application of SVD due to the coupling between the classification and matrix factorization loss. This is why we propose to use the low-rank projected gradient descent (LPGD) algorithm, which iteratively applies gradient descent and low-rank SVD to solve the above non-convex problem in the combined factor space. To our best knowledge, this work is the first to provide an algorithm and convergence guarantee to an optimal solution of the joint optimization problem above and more generally in (3).
> > >
> > >
> > > We eagerly await further discussions and value any additional insights you may have.

---

> > > > ### Comment · Reviewer_pg2E · 2023-08-15
> > > > **Eq (6) and PCA**
> > > >
> > > > I thank the authors for their clarifications.
> > > >
> > > > - Eq. (6): The problem is convex only after the change of variable and the low rank constraint. In the paper, the variables of Eq. (6) are still $W$, $H$, and $\beta$; hence, it is not fully correct to say that stacking of the variables results in a convex function without the change of variable. To argue for convexity, I recommend the authors to modify the objective variable to $\theta$ in the equation.
> > > >
> > > > - PCA: I would like to clarify my previous comment. I do acknowledge that adding supervision while learning the basis and coefficient outperforms the case of unsupervised learning. This is indeed having been shown in numerous prior work (one example from dictionary learning which authors have also cited, [2]). My raised concern is as follows: a) The basis that this method will learn on the reconstruction+classification loss is basically a basis that PCA will learn on the same loss function through some iterative optimization method, as the formulation looks for a few orthogonal bases in the optimization. b) I do not find the application and outperformance of the proposed method surprising, as including supervision into reconstruction (unsupervised) has offered before and shown to outperform [2]. Hence, I find this paper standing more on applying a known method/approach to a breast cancer dataset and its theoretical analysis. Overall, my concerns remain, but I acknowledge that the paper has additional theoretical analysis on the problem; hence, I am willing to raise my score.
> > > >
> > > > Minor comment: There is a line of work on optimization of dictionary learning problem (unsupervised case [3] and supervised [4] to cite a few) through deep learning. I am curious how is the approach in this paper compared to the cited unrolling framework in terms of performance.
> > > >
> > > > [2] Mairal, J., Ponce, J., Sapiro, G., Zisserman, A., & Bach, F. (2008). Supervised dictionary learning. Advances in neural information processing systems, 21.
> > > > [3] Malézieux, B., Moreau, T., & Kowalski, M. (2021). Understanding approximate and unrolled dictionary learning for pattern recovery. arXiv preprint arXiv:2106.06338.
> > > > [4] Rolfe, J. T., & LeCun, Y. (2013). Discriminative recurrent sparse auto-encoders. arXiv preprint arXiv:1301.3775.

---

> > > > > ### Author Response · Authors · 2023-08-16
> > > > >
> > > > > We greatly appreciate the reviewer's follow-up comments. We sincerely appreciate that the reviewer is willing to raise the score. Below we provide further comments and responses to the reviewer's follow-up comments.
> > > > >
> > > > > >Eq. (6): The problem is convex only after the change of variable and the low-rank constraint. ..  To argue for convexity, I recommend the authors to modify the objective variable to $\theta$ in the equation.
> > > > >
> > > > > **Response.** Yes, this is precisely the reasoning we intended to convey. We will revise the manuscript as the reviewer suggests.
> > > > >
> > > > > >PCA: .. : a) The basis that this method will learn on the reconstruction+classification loss is basically a basis that PCA will learn on the same loss function through some iterative optimization method, as the formulation looks for a few orthogonal bases in the optimization.
> > > > >
> > > > > **Response.**
> > > > >
> > > > > a) While we partly agree with the reviewer's point, we would like to provide additional comments on this reviewer's point. It is not entirely clear to us how the joint optimization formulation directly yields that the supervised basis vectors will also be orthogonal. However, our LPGD algorithm iteratively applies a gradient descent step in the low-rank space (that combines all matrix factors to be found) followed by a convex projection $\Pi_{\Theta}$ and then lastly the rank-r SVD $\Pi_{r}$. Our algorithm always ends with rank-r SVD, so the final output of our algorithm will consist of orthogonal basis vectors. More precisely, in order to compute the final output of our LPGD algorithm, we apply rank-r SVD to the matrices
> > > > > \begin{align}
> > > > > [ \beta_{N}^{T} H_{N} \Vert W_{N} H_{N}] \in \mathbb{R}^{(\kappa+p)\times n}\qquad \text{for SDL-$H$}
> > > > > \end{align}
> > > > > \begin{align}
> > > > > [ W_{N} \beta_{N}, W_{N} H_{N}] \in \mathbb{R}^{p\times (\kappa+n)}\qquad \text{for SDL-$W$},
> > > > > \end{align}
> > > > > where $N$ denotes the number of iterations of the LPGD steps. So, algorithmically, we do find orthogonal basis vectors that are iteratively supervised, as correctly pointed out by the reviewer with great insight.
> > > > >
> > > > > However, we also note that our LPGD algorithm is not the only algorithm that iteratively solves the SDL optimization problem. For instance, if one uses the block coordinate descent type algorithm, which has been frequently used in the literature (without needing to combine the factors and perform low-rank projection, with a much weaker theoretical guarantee than ours), it is not entirely clear if the resulting dictionary matrix $W$ will still consist of orthogonal basis vectors. Moreover, reviewer **bLMw** gave a very interesting question on swapping the order of the two projections (i.e., the low-rank projection and the convex projection). (We have included a detailed analysis for this alternative algorithm that first applies the low-rank projection and then the convex projection in our rebuttal.) This alternative algorithm ends with convex projection, so again it is not entirely clear if the supervised basis vectors will be orthogonal.
> > > > >
> > > > > >b) I do not find the application and outperformance of the proposed method surprising, as including supervision in reconstruction (unsupervised) has been offered before and shown to outperform [2]. Hence, I find this paper standing more on applying a known method/approach to a breast cancer dataset and its theoretical analysis. Overall, my concerns remain, but I acknowledge that the paper has additional theoretical analysis on the problem; hence, I am willing to raise my score.
> > > > >
> > > > > **Response.** We absolutely agree with the reviewer that our work does not reinvent the SDL in the high-dimensional setting. We formulate two versions of SDL (SDL-$H$ and SDL-$W$) that encapsulate previously studied SDL-type models and provide a strong theoretical analysis using an LPGD algorithm. We also demonstrate the relevance and potential application of low-rank SDL using our LPGD algorithm in cancer-related gene sequence analysis.
> > > > >
> > > > > Thank you very much for your thoughtful comments and discussion.

---

> > > > > > ### Comment · Reviewer_pg2E · 2023-08-19
> > > > > > **Appreciate the Simplicity of the Method and Two Questions**
> > > > > >
> > > > > > I thank the authors for their response and explanation. Perhaps, the power of their method is in its simplicity that combines supervision with reconstruction to learn a low rank matrix and perform factorization + the theoretical guarantees of their framework. I recommend the authors to incorporate the clarifications they have made here into the paper. I also strongly suggest revising their wording of supervised dictionary learning into supervision low-rank matrix factorization.
> > > > > >
> > > > > > I would like again to ask my question, which left unanswered, and an additional one:
> > > > > >
> > > > > > - There is a line of work on optimization of dictionary learning problem (unsupervised case [3] and supervised [4] to cite a few) through deep learning. I am curious how is the approach in this paper compared to the cited unrolling framework in terms of performance.
> > > > > >
> > > > > > - I appreciate it if the authors comment on deployment of their method on GPU?

---

> > > > > > > ### Author Response · Authors · 2023-08-19
> > > > > > >
> > > > > > > >I thank the authors for their response and explanation. Perhaps, the power of their method is in its simplicity that combines supervision with reconstruction to learn a low rank matrix and perform factorization + the theoretical guarantees of their framework. I recommend the authors to incorporate the clarifications they have made here into the paper. I also strongly suggest revising their wording of supervised dictionary learning into supervision low-rank matrix factorization (SLMF).
> > > > > > >
> > > > > > > **Response.** Thank you for your positive evaluation of our work. We will make sure that we incorporate the clarification we discussed with the reviewer into the revision. Moreover, we absolutely agree with the reviewer that renaming our formulation as "supervision low-rank matrix factorization" instead of SDL is more consistent with the literature and better describes the gist of our problem set-up. We will make this change in the revision.
> > > > > > >
> > > > > > >
> > > > > > >
> > > > > > > >I would like again to ask my question, which left unanswered, and an additional one:
> > > > > > >
> > > > > > > >There is a line of work on the optimization of dictionary learning problems (unsupervised case [3] and supervised [4] to cite a few) through deep learning. I am curious how the approach in this paper compared to the cited unrolling framework in terms of performance.
> > > > > > >
> > > > > > > **Response.** We thank the reviewer that our previous response did not include a discussion on the above point. The references [3] and [4] indeed use a very interesting of unrolling an iterative algorithm (with finite horizon) into a deep neural network, which has the computational advantage of leveraging the enhanced expressive power of a deep network as well as the ease of GPU implementation (e.g., using pytorch). One of the first unrolling approaches was used to approximate the ISTA algorithm for sparse coding by a deep network. The resemblance between finite-horizon ISTA and deep network is apparent since one iteration of ISTA involves a linear transform followed by a soft-thresholding. Hence each iteration of ISTA can be represented by a single layer of a neural network.
> > > > > > >
> > > > > > > While we believe that a similar approach of unrolling our LPGD  algorithm into a deep network is very interesting and can lead to the computationally efficient approximate implementation of our algorithm, the use of low-rank projection in each iteration of our algorithm makes the connection not so immediate. We are aware that there is a recent work [5] that unrolls a single low-rank projection into a deep network. Hence at least one may be able to approximate one iteration of our algorithm into an unrolling deep network. Then we will have to stack these unrolled networks many times, so it is not entirely clear to us if the resulting architecture would still maintain the expected computational efficiency.
> > > > > > >
> > > > > > > [5] Shanmugam, Siva, and Sheetal Kalyani. "Deep learned SVT: Unrolling singular value thresholding to obtain better MSE." arXiv preprint arXiv:2105.06934 (2021).
> > > > > > >
> > > > > > >
> > > > > > >
> > > > > > > >I appreciate it if the authors comment on the deployment of their method on GPU?
> > > > > > >
> > > > > > > **Response.** Thanks for this nice comment. One way to deploy our method in a GPU-friendly implementation would be using the unrolling of low-rank projection as we described in our response to the reviewer's previous comment. We add two more possible approaches for GPU-friendly implementation.
> > > > > > >
> > > > > > > First, instead of the LPGD algorithm, one may unroll the block-coordinate-decent (BCD) type algorithm for our SDL (now SLMF) problem. Each step of this BCD-type algorithm involves solving a convex problem that is similar to (supervise) least squares problem, so it would be possible to implement each step of BCD in a few layers of the unrolled network. This would lead to a simple unrolling approximation of our algorithm. Theoretically, BCD lacks convergence to global optimum in constrat to our LPGD. Empirically we observe that LPGD has much faster convergence than BCD in many instances (see Fig. 1c in the 1-page supplementary PDF in our initial rebuttal). However, it would be very interesting to see if an unrolled BCD algorithm can yield computational benefits over LPGD.
> > > > > > >
> > > > > > > Second, in one of our ongoing works, we developed a simple implementation of solving the bi-convex constrained optimization problem for nonnegative matrix factorization into training certain two-layer neural networks. This construction does not unroll NMF, but one can iterate the forward- and back-propagation for any number of times on the fixed network until a desired convergence has been reached. This two-layer neural network training can be done extremely efficiently on a GPU. In future work, we are planning to investigate a similar "uniform-depth" neural network implementation of SLMF problem for an efficient GPU implementation.

---

> > > > > > > > ### Comment · Reviewer_pg2E · 2023-08-20
> > > > > > > > **Recommend to Include the Unrolling and GPU discussion into the paper**
> > > > > > > >
> > > > > > > > I thank the authors for their detailed explanation. I recommend adding the above discussion on unrolling and its relation to the proposed method, and GPU to the paper. My final recommendation is to make the source code publish.
> > > > > > > >
> > > > > > > > Overall, some of my concerns are addressed on the usage of dictionary learning, the relation between their method and PCA. Given the author's engagement, the clarifications made during the discussion, and reviews from other reviewers, I have adjusted my score.

---

### Author Rebuttal · Authors · 2023-08-09

We submit an optional 1-page PDF to show the revised Figures 2 and 3 with captions. In Figure 2, we present a comprehensive comparison between our LPGD algorithm and BCD. In Figure 3, we include additional experimental details, focusing on breast cancer classification, which successfully identifies well-known oncogene and cancer-associated genes (prognostic markers). Moreover, Figure 3 now contains further benchmarks that compare our approach to convolutional neural networks and feed-forward neural networks.

---

### Decision · Program_Chairs · 2023-09-21

**Decision:**

Accept (poster)

**Comment:**

The reviewers commend the paper's approach in DL, which combines supervision with reconstruction to facilitate the learning of a low-rank matrix. The transformation of this approach into a low-rank optimization problem with a convex objective is particularly noteworthy. The inclusion of novel theoretical results is also acknowledged.

Certain reviewers initially expressed concerns about the paper's focus in relation to the extensive studies within DL, particularly those emphasizing overcomplete dictionaries with sparse regularization. However, these concerns, along with others, were effectively addressed during the authors' engaged response.

Taking into account the recommendations from the reviewers and the productive discussions during the rebuttal, I am inclined to recommend its acceptance.

For the revised version, I strongly urge the authors to fully integrate the constructive suggestions provided by the reviewers. This should encompass clarifying the distinctions between their approach and the dominant dictionary learning paradigm involving sparse regularization on overcomplete dictionaries. Additionally, revising the terminology from "supervised dictionary learning" to "supervised low-rank matrix factorization" would enhance clarity. It would be beneficial to make the source code available and to augment the paper with more comprehensive experimental results and comparisons.